# Dynamics of Stochastic Momentum Methods on Large-scale, Quadratic Models

**Courtney Paquette** [*]
Department of Mathematics and Statistics
McGill University
Montreal, Quebec H2Y 2M5
courtney.paquette@mcgill.ca

**Elliot Paquette** [†]
Department of Mathematics and Statistics
McGill University
Montreal, Quebec H2Y 2M5
elliot.paquette@mcgill.ca

## Abstract

We analyze a class of stochastic gradient algorithms with momentum on a high-dimensional random least squares problem. Our framework, inspired by random matrix theory, provides an exact (deterministic) characterization for the sequence of loss values produced by these algorithms which is expressed only in terms of the eigenvalues of the Hessian. This leads to simple expressions for nearly-optimal hyperparameters, a description of the limiting neighborhood, and average-case complexity.

As a consequence, we show that (small-batch) stochastic heavy-ball momentum with a fixed momentum parameter provides no actual performance improvement over SGD when step sizes are adjusted correctly. For contrast, in the non-strongly convex setting, it is possible to get a large improvement over SGD using momentum. By introducing hyperparameters that depend on the number of samples, we propose a new algorithm SDANA (stochastic dimension adjusted Nesterov acceleration) which obtains an asymptotically optimal average-case complexity while remaining linearly convergent in the strongly convex setting without adjusting parameters.

Methods that incorporate momentum and acceleration play an integral role in machine learning where they are often combined with stochastic gradients. Two of the most popular methods in this category are the heavy-ball method (HB) [Polyak, 1964] and Nesterov's accelerated method (NAG) [Nesterov, 2004]. These methods are known to achieve optimal convergence guarantees when employed with *exact gradients* (computed on the full training data set), but in practice, these momentum methods are typically implemented with *stochastic* gradients. In the influential work Sutskever et al. [2013], the authors demonstrated empirical advantages of augmenting stochastic gradient descent (SGD) with the momentum machinery and, as a result, momentum methods are widely used for training deep neural networks. Yet despite the popularity of these stochastic momentum methods, the theoretical understanding of these algorithms remains rather limited.

In this paper, we study the dynamics of stochastic momentum methods (with batch size 1 and constant step size) rooted in heavy-ball momentum and Nesterov's accelerated gradient algorithms on a least squares problem. Our approach uses a framework inspired by the phenomenology of random matrix theory (see Paquette et al. [2021]), which gains explanatory power when the number of samples ($n$) and features ($d$) are large. A key contribution of this work is a simple description of the exact dynamics for a class of stochastic momentum methods in the *high-dimensional limit*; we construct a smooth, deterministic function $\psi(t)$ such that $f(\boldsymbol{x}_k) \to \psi(k/n)$ as $n \to \infty$. This function $\psi$ solves

---

[*]Website courtneypaquette.github.io .

[†]Website elliotpaquette.github.io .

---
**Algorithm 1** Generic stochastic momentum method.
---
**Given**: step sizes $\Gamma_1, \Gamma_2 > 0$ and momentum parameter $\Delta(k) > 0$
**Initialize**: $\boldsymbol{x}_0 \in \mathbb{R}^d$ and $\boldsymbol{y}_0 = \boldsymbol{0}$
**for** $k \geq 1$, Select $i_k \in [n]$ uniformly and update

$$\boldsymbol{y}_k = (1 - \Delta(k))\boldsymbol{y}_{k-1} + \Gamma_1 \nabla f_{i_k}(\boldsymbol{x}_k) \qquad \text{and} \qquad \boldsymbol{x}_{k+1} = \boldsymbol{x}_k - \Gamma_2 \nabla f_{i_k}(\boldsymbol{x}_k) - \boldsymbol{y}_k \quad (0.2)$$

---

a Volterra integral equation:

$$\psi(t) = F(t) + \int_0^t \mathcal{K}_s(t)\psi(s)\,\mathrm{d}s. \tag{0.1}$$

Here $F(t)$ and $\mathcal{K}_s(t)$ are explicit, see Theorem 1 for a precise statement. This Volterra equation (0.1) gives an accurate prediction of the behavior of stochastic methods, see Figure 1. We then analyze these dynamics providing insight into step size and momentum parameter selections as well as providing both upper and lower average-case complexity (*i.e.*, the complexity of an algorithm averaged over all possible inputs) for the last iterate.

As we show in this work, both theoretically and empirically, (small batch size) SGD with heavy-ball momentum (SHB) for any fixed momentum parameter does *not* provide any acceleration over plain SGD on large-scale least square problems. We conclude under an identification of the parameters that $f(\boldsymbol{x}_k^{\mathrm{shb}}) = f(\boldsymbol{x}_k^{\mathrm{sgd}})$ for all $k$ up to errors that vanish as $n$ grows large (upper bounds of this nature have been observed before: see Kidambi et al. [2018], Sebbouh et al. [2020], Zhang et al. [2019]). Thus while SHB may provide a speed-up over SGD, it is only due to an effective increase in the learning rate, and this speed-up could be matched by appropriately adjusting the learning rate of SGD.

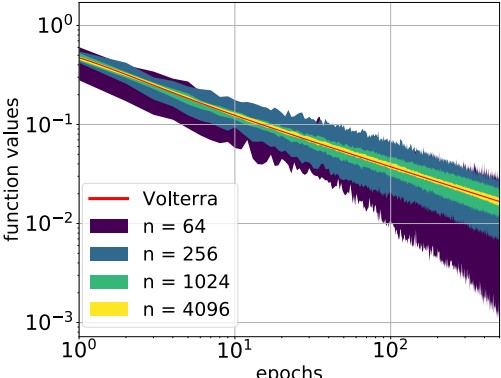

Figure 1: **Concentration of stochastic heavy-ball** (SHB) on a Gaussian random least squares problem (Sec. 2), $d = n$, an 80% confidence interval (shaded region) over 10 runs for each $n$, the parameters for SHB (Table 1) are $(\theta, \gamma) = (0.1, 0.08)$. The random least squares problem becomes non-random in the large limit and all runs of SHB converge to a deterministic function $\psi(t)$ (red) given by our Volterra equation (0.1).

The root of SHB's failure to provide meaningful acceleration is that a fixed momentum parameter is not aggressive enough when $n$ is large. We propose a new algorithm that uses a dimension-based modification of Nesterov (see Alg. 1 and Table 1). The resulting algorithm, SDANA, matches the average-case complexity of SGD when the least-squares problem is strongly convex and obtains an average-case complexity of $1/k^3$ in the convex setting.

## 1 Motivation and related work

We consider the large finite-sum setting

$$\min_{\boldsymbol{x} \in \mathbb{R}^d} \left\{ f(\boldsymbol{x}) = \frac{1}{n}\sum_{i=1}^n f_i(\boldsymbol{x}) = \frac{1}{2}\sum_{i=1}^n (\boldsymbol{a}_i \boldsymbol{x} - \boldsymbol{b}_i)^2 = \frac{1}{2}\|\boldsymbol{A}\boldsymbol{x} - \boldsymbol{b}\|^2 \right\},$$

for data matrix $\boldsymbol{A} \in \mathbb{R}^{n \times d}$ whose $i$-th row is denoted by $\boldsymbol{a}_i \in \mathbb{R}^{d \times 1}$ and target vector $\boldsymbol{b} \in \mathbb{R}^n$ (detailed in Section 2). We make the convention that the matrix $\boldsymbol{A}$ has max row norm equal to 1. Note we absorb some $n$–dependence into $\boldsymbol{A}$ and $\boldsymbol{b}$ by setting $\frac{1}{n}f_i(\boldsymbol{x}) = \frac{1}{2}(\boldsymbol{a}_i \boldsymbol{x} - b_i)^2$. We investigate a generic class of stochastic momentum algorithms (see Alg. 1 and Table 1). Particularly, we introduce a sub-class, denoted by SDA($\gamma_1, \gamma_2, \Delta$), of Alg. 1 which has parameters that are appropriately adjusted for large problems (large number of samples $n$ and large model size $d$); we refer to the *dimension* of

Table 1: **Summary of the parameters for a variety of stochastic momentum algorithms** that fit within the framework of Alg. 1, denote the normalized trace by $m \stackrel{\text{def}}{=} n^{-1} \sum_{i=1}^{n} \|a_i\|^2$. The default parameters are chosen so that its linear rate is no slower, by a factor of $4$ than the fastest possible rate for an algorithm having optimized over all step size choices.

| Methods | Alg. 1 Parameters | | | Default Parameters |
| --- | --- | --- | --- | --- |
| | $\Gamma_1$ | $\Gamma_2$ | $\Delta(k)$ | |
| Stochastic gradient descent: **SGD**$(\gamma)$ | $0$ | $\gamma$ | $1$ | $\gamma = \frac{1}{m}$, (Prop. E.4) |
| Stoch. gradient descent w/ momentum: **SHB**$(\gamma, \theta)$ | $\gamma$ | $0$ | $\theta$ | (see Fig. 2) |
| Stoch. dimension-adjusted heavy-ball: **SDAHB**$(\gamma, \theta)$ (This paper) | $\dfrac{\gamma}{n}$ | $0$ | $\dfrac{\theta}{n}$ | $\gamma = \frac{\theta}{m}$,   $\theta = 2$ (Prop. E.3) |
| Stoch. dimension-adjusted Nesterov's accel. method: **SDANA**$(\gamma_1, \gamma_2, \theta)$ (This paper) | $\dfrac{\gamma_1}{n}$ | $\gamma_2$ | $\dfrac{\theta}{k+n}$ | $\gamma_1 = \frac{1}{4m}$,   $\gamma_2 = \frac{1}{m}$, $\theta = 4$ (Cor. D.1) |

the problem as $n$. The class $\text{SDA}(\gamma_1, \gamma_2, \Delta)$ is defined by setting in Alg. 1

$$\Gamma_1 = \frac{\gamma_1}{n}, \quad \Gamma_2 = \gamma_2, \quad \text{and} \quad \Delta(k) \stackrel{\text{def}}{=} \frac{1}{n}(\log \varphi)'(\tfrac{k}{n}), \tag{1.1}$$

where $\gamma_1, \gamma_2 > 0$ are step sizes and $\varphi$ is a smooth function that represents a momentum schedule. Although we develop some theory for general $\varphi$, we are principally interested in the two cases:

$$(\text{SDANA}) \ \Delta(k) = \frac{\theta}{k+n} \leftrightarrow \varphi(t) = (1+t)^\theta \quad \text{and} \quad (\text{SDAHB}) \ \Delta(k) = \frac{\theta}{n} \leftrightarrow \varphi(t) = e^{\theta t}. \tag{1.2}$$

To avoid confusion between different algorithms, we add superscripts indicating the algorithm (*e.g.*, we denote $\Gamma_2 = \gamma^{\text{sgd}}$, the step size parameter for SGD). For all these algorithms, we are interested in:

1. An expression for the (deterministic) dynamics of these algorithms when *multiple passes* on the data set are allowed. This contrasts with the "streaming" or "online" setting where at each iteration one generates an independent never-before-used data point.

2. A formula for choosing the hyperparameters and a discussion of the dependence of these hyperparameters on number of features and samples.

3. Upper and lower bounds on the average-case complexity of the *last iterate* to a neighborhood; this neighborhood disappears entirely in the overparameterized regime, while in the underparameterized regime the limiting distance to optimality concentrates in the high-dimensional limit.

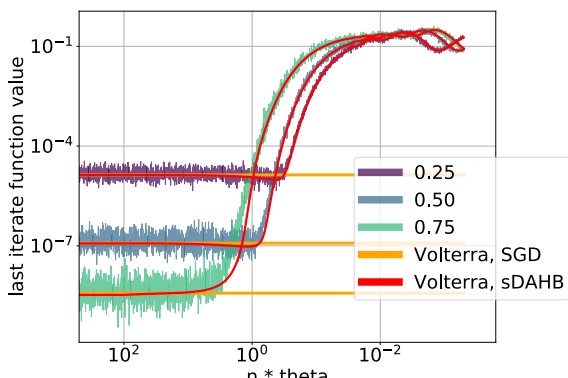

Figure 2: **Equivalence of SGD and stochastic Heavy-Ball.**, For every $\gamma^{\text{sgd}} \in \{0.25, 0.50, 0.75\}$, we select a pair of parameters $(\gamma^{\text{shb}}, \theta^{\text{shb}})$ so that $\gamma^{\text{sgd}} = \frac{\gamma^{\text{shb}}}{\theta^{\text{shb}}}$. We run SHB 3000 times with varying $\theta^{\text{shb}}$ on (2.1) with $d = 500, n = 1000$, and plot the value of the last iterate after 50 epochs. Small $\theta^{\text{shb}}$ matches SGD (orange, theory), illustrating their equivalence. With $n \cdot \theta^{\text{shb}} \approx 1$, a change is observed, giving a small improvement for some values of $\gamma^{\text{sgd}}$. Plotted against theory for SDAHB (red Volterra, see Thm. 1 and App E), which is the same algorithm as SHB after a change of parameters.

**Why divide by n? A negative result.**   Throughout the literature, there are examples for which (small batch size) stochastic momentum methods such as SHB and stochastic Nesterov's accelerated method (SNAG) achieve performances equal to (or even worse) than small batch size SGD (see

*e.g.*, Kidambi et al. [2018], Sebbouh et al. [2020], Zhang et al. [2019] for heavy-ball and Assran and Rabbat [2020], Liu and Belkin [2020], Zhang et al. [2019] for Nesterov). We also observe this phenomenon (see (2.4) and App. E.4, Thm 5), and we illustrate this in Fig. 2. The stochastic heavy-ball method for *any* fixed step size and momentum parameters (Table 1) has the *exact* same dynamics as vanilla SGD, which is to say, by setting the step size parameter in SGD to be $\gamma^{\mathrm{sgd}} = \frac{\gamma^{\mathrm{shb}}}{\theta^{\mathrm{shb}}}$, the two algorithms have the same loss values provided the number of samples is sufficiently large, *i.e.*, $f(\boldsymbol{x}_k^{\mathrm{sgd}}) = f(\boldsymbol{x}_k^{\mathrm{shb}})$.

As a consequence of this, the average-case complexity of SHB equals the last iterate complexity of SGD (This was observed in [Sebbouh et al., 2020] with an upper bound, but our result shows an exact equivalence between last iterate SGD and SHB). Although App. E.4, Thm 5 (see also (2.4)) gives an unsatisfactory answer to stochastic heavy-ball with fixed $\theta$ and small batch-size, our analysis illuminates a path forward. Particularly, *one must choose **dimension-dependent** parameters to achieve dynamics which differ from SGD.*

**Why divide by n? A positive result.** Adapting SHB for dimension, we arrive at stochastic dimension adjusted heavy ball (SDAHB). While formally equivalent to SHB, we include the dimension parameters to emphasize that any improvement in its performance for large $n$ requires it. Nonetheless, the speed-up for heavy ball is modest (see Fig. 2).

On the other hand, we show that a dimension adapted version of Nesterov acceleration, SDANA, has a large improvement in the non-strongly convex case. Moreover, with a simple parameter choice (see the default parameters in Table 1), it will perform linearly in the strongly convex case, and competitively with learning-rate-tuned SGD (or SHB), while performing orders-of-magnitude faster ($k^{-3}$ as compared to SGD $k^{-1}$) for the non-strongly convex setting (see Fig. 3 and Table 3). We believe this gives SDANA promise as an algorithm outside of the least squares context, in situations in which loss landscapes can range between alternately curved and very flat, frequently observed in neural network settings (see Ghorbani et al. [2019], Li et al. [2018], Sagun et al. [2016]).

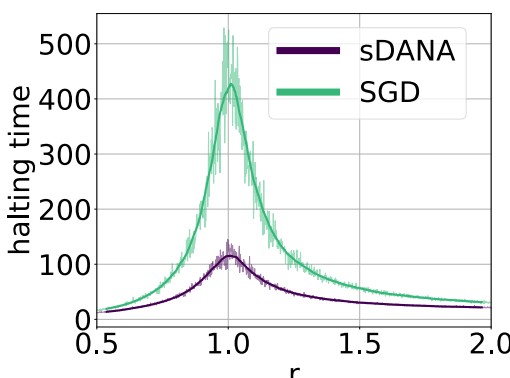

Figure 3: **Convergence of SDANA.** Halting time of SDANA vs SGD with default parameters on the Gaussian random least squares problem (2.1) with varying $d$ and $n = 1024$. When the ratio $r = d/n \to 1$ (in which case $\max\{\lambda_{\min}(\boldsymbol{A}^T\boldsymbol{A}), \lambda_{\min}(\boldsymbol{A}\boldsymbol{A}^T)\} \to 0$), SDANA requires significantly fewer iterations to reach a loss of $10^{-5}$. As the ratio $r$ moves away from 1, the performance of SDANA matches SGD.

**Related work.** Recent works have established convergence guarantees for SHB in both strongly convex and non-strongly convex setting [Flammarion and Bach, 2015, Gadat et al., 2016, Orvieto et al., 2019, Sebbouh et al., 2020, Yan et al., 2018]; the latter references having established almost sure convergence results. Specializing to the setting of minimizing quadratics, the iterates of SHB converge linearly (but not in $L^2$) under an exactness assumption [Loizou and Richtárik, 2017] while under some additional assumptions on the noise of the stochastic gradients, [Can et al., 2019, Kidambi et al., 2018] show linear convergence to a neighborhood of the solution.

Convergence results for stochastic Nesterov's accelerated method (SNAG), under both strongly convex and non-strongly setting, have also been established. The works [Assran and Rabbat, 2020, Aybat et al., 2018, Can et al., 2019, Kulunchakov and Mairal, 2019] showed that SNAG converged at the optimal accelerated rate to a neighborhood of the optimum. Under stronger assumptions, convergence to the optimum at an accelerated rate is guaranteed. Examples include the strong growth condition [Vaswani et al., 2019] and additive noise on the stochastic gradients [Laborde and Oberman, 2019].

The lack of general convergence guarantees showing acceleration for existing momentum schemes, such as heavy-ball and NAG, in the stochastic setting, has led many authors to design alternative

acceleration schemes [Allen-Zhu, 2017, Ghadimi and Lan, 2012, 2013a, Kidambi et al., 2018, Kulunchakov and Mairal, 2019, Liu and Belkin, 2020].

## 2 Random least squares problem

To formalize the analysis of a high–dimensional, typical least squares problem, we define the *random least squares problem*:

$$\underset{\boldsymbol{x} \in \mathbb{R}^d}{\arg\min} \left\{ f(\boldsymbol{x}) = \frac{1}{n} \sum_{i=1}^{n} f_i(\boldsymbol{x}) \stackrel{\text{def}}{=} \frac{1}{2} \sum_{i=1}^{n} (\boldsymbol{a}_i \boldsymbol{x} - b_i)^2 \right\}, \quad \text{with } \boldsymbol{b} \stackrel{\text{def}}{=} \boldsymbol{A}\widetilde{\boldsymbol{x}} + \boldsymbol{\eta}. \quad (2.1)$$

The data matrix $\boldsymbol{A}$ is random and we shall introduce assumptions on $\boldsymbol{A}$ as they are needed, but we suggest as a central example the *Gaussian random least squares* where each entry of $\boldsymbol{A}$ is sampled independently from a standard normal distribution with variance $\frac{1}{d}$. We always make the assumption that each row $\boldsymbol{a}_i \in \mathbb{R}^{d \times 1}$ is centered and is normalized so that $\max_i \{\mathbb{E}[\|\boldsymbol{a}_i\|^2]\} = 1$.

As for the target $\boldsymbol{b} = \boldsymbol{A}\widetilde{\boldsymbol{x}} + \boldsymbol{\eta}$, we assume it comes from a generative model corrupted by noise, where $\widetilde{\boldsymbol{x}}$ is signal and $\boldsymbol{\eta}$ is noise.

**Assumption 1** (Initialization, signal, and noise)**.** *The initial vector $\boldsymbol{x}_0 \in \mathbb{R}^d$ is chosen so that $\boldsymbol{x}_0 - \widetilde{\boldsymbol{x}}$ is independent of the matrix $\boldsymbol{A}$. The noise $\boldsymbol{\eta}$ is centered and has i.i.d. entries, independent of $\boldsymbol{A}$. The signal and noise are normalized so that*

$$\mathbb{E}\|\boldsymbol{x}_0 - \widetilde{\boldsymbol{x}}\|_2^2 = R\frac{d}{n} \quad \text{and} \quad \mathbb{E}[\|\boldsymbol{\eta}\|_2^2] = \widetilde{R}.$$

Note that deterministic $\boldsymbol{x}_0 - \widetilde{\boldsymbol{x}}$ satisfies this assumption. The vectors $\boldsymbol{x}_0 - \widetilde{\boldsymbol{x}}$ and $\boldsymbol{\eta}$ arise as a result of preserving a constant signal-to-noise ratio in the generative model. Such generative models with this scaling have been used in numerous works [Gerbelot et al., 2020, Hastie et al., 2019, Mei and Montanari, 2019].

For the data matrix $\boldsymbol{A}$ we introduce the Hessian matrix $\widetilde{\boldsymbol{H}} = \boldsymbol{A}^T\boldsymbol{A}$ and its symmetrization $\boldsymbol{H} = \boldsymbol{A}\boldsymbol{A}^T$. Let $\lambda_1 \geq \ldots \geq \lambda_n$ be the eigenvalues of the matrix $\boldsymbol{H}$. Up to appending zeros, this is the same ordered sequence of eigenvalues as those of the Hessian. Define the *empirical spectral measure* (ESM) of $\boldsymbol{H}$, $\mu_{\boldsymbol{H}}$ by the formula

$$\int g(\lambda) \mu_{\boldsymbol{H}}(\mathrm{d}\lambda) \stackrel{\text{def}}{=} \frac{1}{n} \sum_{i=1}^{n} g(\lambda_i) \quad \text{for any continuous function } g : \mathbb{R} \to \mathbb{R}. \quad (2.2)$$

This gives the interpretation for the empirical spectral measure as the distribution of an eigenvalue of $\boldsymbol{H}$ chosen uniformly at random.

**Diffusion approximation.** Our analysis will use a diffusion approximation to analyze the SDA$(\gamma_1, \gamma_2, \Delta)$ class of stochastic momentum methods (see (1.1)) on the random least squares setup (2.1). We call the approximation *homogenized SGD*:

$$\mathrm{d}\boldsymbol{X}_t \stackrel{\text{def}}{=} -\gamma_2 \, \mathrm{d}\boldsymbol{Z}_t - \frac{\gamma_1}{\varphi(t)} \int_0^t \varphi(s) \, \mathrm{d}\boldsymbol{Z}_t, \quad \text{where} \quad \mathrm{d}\boldsymbol{Z}_t \stackrel{\text{def}}{=} \nabla f(\boldsymbol{X}_t) \, \mathrm{d}t + \sqrt{\frac{2}{n} f(\boldsymbol{X}_t) \nabla^2(f)} \, \mathrm{d}\boldsymbol{B}_t, \quad (2.3)$$

and with initial conditions given by $\boldsymbol{X}_0 = \boldsymbol{x}_0$. The process $(\boldsymbol{B}_t : t \geq 0)$ is a $d$–dimensional standard Brownian motion. Here time is scaled in such a way that $t = 1$ represents one pass over the dataset or $n$ calls to the stochastic oracle. Similar SDEs have appeared frequently in the theory around SGD, see *e.g.*, Li et al. [2017, 2019], Mandt et al. [2016].

The advantage of the homogenized SGD diffusion is that we are able to give an explicit representation of the expected loss values on a least squares problem, even at finite $n$.

**Theorem 1** (Volterra dynamics at finite $n$)**.** *Let $\mathbb{E}_{\boldsymbol{H}}[\cdot]$ be the conditional expectation where $\boldsymbol{H}$ is held fixed. There are non-negative functions $F(t)$ and $\mathcal{K}_s(t)$ for $s, t \geq 0$ depending on the spectrum of $\boldsymbol{H}$ so that for all $t \geq 0$*

$$\mathbb{E}_{\boldsymbol{H}}[f(\boldsymbol{X}_t)] = F(t) + \int_0^t \mathcal{K}_s(t) \, \mathbb{E}_{\boldsymbol{H}}[f(\boldsymbol{X}_s)] \, \mathrm{d}s, \quad \text{for all} \quad t \geq 0. \quad (2.4)$$

Table 2: **Summary of the convolution kernel for the Volterra equations** (2.4) for all algorithms considered. The convolution kernel (below) for SDANA is an approximation to the true kernel. The forcing terms $G^{(\lambda)}(t)$ for SGD and SDAHB are similar to the kernel whereas the forcing term for SDANA is defined only by solving a 3rd-order ODE.

| Methods | Kernel, $K_s^{(\lambda)}(t)$ | |
|---|---|---|
| **SGD**$(\gamma)$ | $\gamma^2 \lambda^2 e^{-2\gamma\lambda(t-s)}$ | — |
| **SDAHB**$(\gamma, \theta)$ (This paper) | $\frac{2\gamma^2\lambda^2}{\omega} e^{-(t-s)\theta}(1 - \cos((t-s)\sqrt{\omega}))$ | $\omega = 4\lambda\gamma - \theta^2$ |
| **SDANA**$(\gamma_1, \gamma_2, \theta)$ $\frac{\lambda}{\omega} e^{-\lambda\gamma_2(t-s)} \left(1 - \cos((t-s)\sqrt{\lambda\omega} + \vartheta)\right)$ (This paper) | | $\tan(\vartheta) = \frac{(\omega - 2\gamma_1)\sqrt{4\gamma_1 - \omega}}{(\omega - 2\gamma_1)^2 - 2\gamma_1^2}$ $\omega = 4\gamma_1 - \gamma_2^2\lambda$ |

*The forcing function $F$ and kernel $\mathcal{K}$ are given by*

$$F(t) = \frac{1}{n} \sum_{i=1}^{n} (R\lambda_i + \widetilde{R}) G^{(\lambda_i)}(t) \quad and \quad \mathcal{K}_s(t) = \frac{1}{n} \sum_{i=1}^{n} K_s^{(\lambda_i)}(t).$$

*The functions $G^{(\lambda)}$ and $K^{(\lambda)}$ are solutions of an initial value problem with a 3-rd order ODE which depend on the hyperparameters $(\gamma_1, \gamma_2, \Delta)$ (Note, there is a 1-to-1 relationship with $\varphi$, see (1.1)).*

We refer to the supplemental materials for the explicit third–order ODE (see Theorem 4 for full details). The expression in (2.4) is a Volterra integral equation, which can be analyzed explicitly, and has a relatively simple theory, especially in the case that the kernel is of convolution type (i.e. $\mathcal{K}_s(t) = \mathcal{I}(t - s)$ for some function $\mathcal{I}$; see Table 2 for kernels. We also note that in the case of SGD$(\gamma)$ ($\varphi$ is unused), the functions $G$ and $K$ become particularly simple

$$G^{(\lambda)}(t) = e^{-2\gamma\lambda t} \quad \text{and} \quad K_s^{(\lambda)}(t) = \gamma^2 \lambda^2 e^{-2\gamma\lambda(t-s)}.$$

**Comparing homogenized SGD to the SDA class.** When $A$ is a random matrix, we can compare the diffusion (2.3) to SDA (1.1) when $n$ and $d$ are large. The argument is based on the results of Paquette et al. [2021], and we do it only in the case of SGD:

**Theorem 2** (Concentration of SGD). *Suppose that $A$ is a* left-orthogonally invariant *random matrix, meaning that for any orthogonal matrix $O \in \mathbb{R}^{n \times n}$, $OA \stackrel{law}{=} A$. Suppose further that the noise vector $\eta$ is independent of $A$ and that it satisfies*

$$\mathbb{E}\left[\|\eta\|_\infty^p\right] = \mathcal{O}(n^{\epsilon - p/2}) \quad for\ any\ \epsilon, p > 0.$$

*Fix $\gamma < 2n(\operatorname{tr} H)^{-1}$, the convergence threshold of SGD$(\gamma)$. There is an absolute constant $\varepsilon > 0$ and a constant $c(T, \lambda_H^+)$ so that with $p = \min\{d, n\}$,*

$$\Pr(\sup_{0 \le t \le T} |\mathbb{E}_H[f(X_t)] - f(x_{[nt]})| > c(T, \lambda_H^+) p^{-\varepsilon} \mid \lambda_H^+) \le p^{-\varepsilon}.$$

We expect that this theorem can be generalized, to include the entire SDA class. We also expect that the orthogonal invariance assumption can be relaxed somewhat (for example to include classes of non–Gaussian isotropic features matrices), but not entirely: the left singular vectors need to have some degree of isotropy for the result to hold. The numerical results show very good general agreement with theory and demonstrate the validity of the approximation: see Figures 1, 2, and 5 as well as Figure 6 on real data. Nonetheless, it is of great theoretical interest to establish the theorem in greater generality. We show a heuristic derivation in App. B.

## 3 Main results

In this section, and in light of the Thm. 1, we outline how to use this Volterra equation (2.4) to produce average-case analysis, nearly optimal hyperparameters, and exact expressions for the neighborhood and convergence thresholds. For additional details, see Supplementary Materials.

Table 3: **Asymptotic average-case convergence guarantees** for $\mathbb{E}_{\boldsymbol{H}}[f(\boldsymbol{X}_t)] - \frac{\widetilde{R}\dim(\ker(\boldsymbol{H}))}{2n(1-\|\mathcal{I}\|)}$ (last iterate) under default parameters (see Table 1) for the isotropic features model. The norm of the kernel is controlled by two values: the normalized trace of the matrix $m = \sum_{i=1}^n \|\boldsymbol{a}_i\|^2/n$ and the mass of the spectral measure (empirical or limiting) at 0 which we denote by $p = \dim(\ker(\boldsymbol{H}))/n$. Average-case complexity is strictly better than the worst-case complexity, in some cases by a factor $\gamma$ vs. $\gamma^2$. As in Paquette et al. [2021], the worst-case rates in non-strongly convex setting have dimension dependent constants due to the distance to the optimum $\|\boldsymbol{x}^\star - \boldsymbol{x}_0\|^2 \approx d$ which appears in the bounds. SDANA obtains an accelerated average-case rate in the non-strongly convex case over SGD while matching the average-case rate of SGD in strongly convex regime. These rates are achieved without changing hyperparameters in SDANA. For worst-case rates, see [Bottou et al., 2018, Theorem 4.6] [Ghadimi and Lan, 2013b, Theorem 2.1]; $\lambda^+$ can be replaced by the max-$\ell^2$-row-norm.

| | | Kernel, $\|\mathcal{I}\|$ | Strongly convex | Non-strongly convex |
|---|---|---|---|---|
| **SGD**$(\gamma)$ | Worst | | $\exp(-\gamma t \lambda^- + \frac{\gamma^2}{2}(\lambda^+)^2 t)$ | $(R + \widetilde{R}\cdot d)\cdot\frac{1}{t}$ |
| **SGD**$(\gamma)$ | Avg | $\frac{\gamma}{2}m$ (Eq. (C.4)) | $\exp(-\gamma t\lambda^-)$ (Lem. C.4) | $Rt^{-3/2} + \widetilde{R}t^{-1/2}$ (Eq. (C.6)) |
| **SDAHB**$(\gamma,\theta)$ | | $\frac{\gamma}{2\theta}m$ (Eq. (E.11)) | $\exp(-t\frac{\gamma\lambda^-\theta}{2\gamma\lambda^-+\theta^2})$ (Prop. E.3) | $Rt^{-3/2} + \widetilde{R}t^{-1/2}$ (Eq. (E.13)) |
| **SDANA**$(\gamma_1,\gamma_2,\theta)$ | | $\frac{\gamma_1}{2\gamma_2}(1-p) + \frac{\gamma_2}{2}m$ (Eq. (D.25)) | $\exp(-t\frac{3\gamma_1\gamma_2\lambda^-}{2\gamma_2^2\lambda^-+4\gamma_1})$ (Cor. D.1) | $Rt^{-3} + \widetilde{R}t^{-1}$ (Prop. D.3) |

### 3.1 Convolution Volterra convergence analysis: convergence threshold and neighborhood

For all algorithms considered (SDANA, SDAHB, SHB, SGD), the Volterra equation in Theorem 1 can be expressed in a simpler form, that is, as a *convolution–type Volterra equation*

$$\mathbb{E}_{\boldsymbol{H}}[f(\boldsymbol{X}_t)] = F(t) + \int_0^t \mathcal{I}(t-s)\,\mathbb{E}_{\boldsymbol{H}}[f(\boldsymbol{X}_s)]\,\mathrm{d}s \quad \text{for all} \quad t \geq 0. \tag{3.1}$$

The forcing function $F$ and the convolution kernel $\mathcal{I}$ are non-negative functions that depend on the spectrum of $\boldsymbol{H}$ and SDA parameters (see Table 2 for the kernels of various algorithms). In the case of SDANA, the kernel is in fact not a convolution Volterra equation, but it can be approximated by one so that it matches the non–convolution equation as $t \to \infty$.

First, the forcing function $F$ will in all cases be bounded, and in fact it will converge as $t \to \infty$ to a deterministic value,

$$F(t) \xrightarrow[t\to\infty]{} \frac{\widetilde{R}\mu_{\boldsymbol{H}}(\{0\})}{2} = \frac{\widetilde{R}\dim(\ker(\boldsymbol{H}))}{2n}. \tag{3.2}$$

Here $\mu_{\boldsymbol{H}}$ is the empirical spectral measure (2.2) which exists for even non-random matrices. It follows that the solution of (3.1) remains bounded if the norm $\|\mathcal{I}\| = \int_0^\infty \mathcal{I}(t)\,\mathrm{d}t$ is less than 1.

**Theorem 3** (Convergence threshold and limiting loss). *If the norm* $\|\mathcal{I}\| < 1$, *the algorithm is convergent in that*

$$\mathbb{E}_{\boldsymbol{H}}[f(\boldsymbol{X}_t)] \xrightarrow[t\to\infty]{} \frac{\widetilde{R}\dim(\ker(\boldsymbol{H}))}{2n(1-\|\mathcal{I}\|)} \qquad \textit{(limiting loss)}.$$

This theorem gives a convergence threshold for all algorithms in Table 1 based only on the norm of the kernel of the Volterra equation, which is easily computable (see Table 3).

### 3.2 Average case analysis

**Limiting spectral measures.** Average-case complexity looks at the typical behavior of an algorithm when some

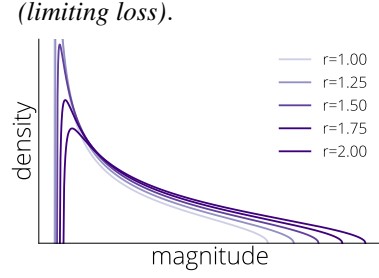

Figure 4: The *Marchenko-Pastur law*$(r)$. Varying $r = d/n$.

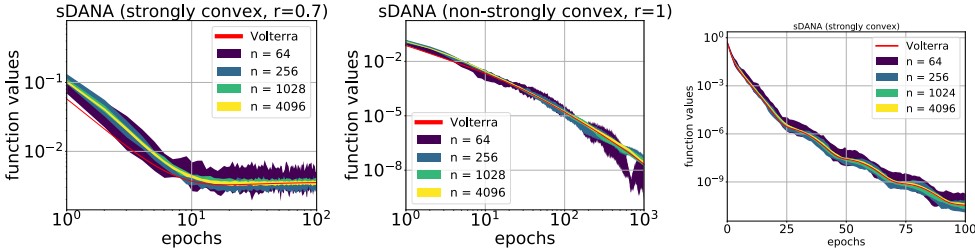

Figure 5: **Concentration of SDANA.** 80% confidence interval on 10 runs with default parameters on Gaussian random least squares problem (2.1), $d/n = r$, with noise $\widetilde{R} = 0.01$ and signal $R = 1$. The convolution-type Volterra equation (red, (3.1)) predicts the performance of SDANA and it reflects the oscillatory trajectories typically seen in momentum methods due to overshooting. Because the convolution Volterra is only an approximation to the kernel, there is always an initial mismatch between actual runs of SDANA and the Volterra solution. As $t \to \infty$, the convolution-type Volterra equation better approximates SDANA. For more details on numerical simulations see App. F.

of its inputs are chosen at random. To formulate an average case analysis that is representative of what is seen in a large scale optimization problem, we will take a limit of the empirical spectral measure as $n$ and $d$ are taken to infinity. So, we suppose that the following holds:

**Assumption 2** (Spectral limit). *Let $\boldsymbol{A}$ be an $n \times d$ matrix drawn from a family of random matrices such that the number of features, $d$, tends to infinity proportionally to the size of the data set, $n$, so that $\frac{d}{n} \to r \in (0, \infty)$; and suppose these random matrices satisfy the following.*

*1. The eigenvalue distribution of $\boldsymbol{H} = \boldsymbol{A}\boldsymbol{A}^T$ converges to a deterministic limit $\mu$ with compact support. Formally, the empirical spectral measure (ESM) converges weakly to $\mu$, in that for all bounded continuous $g : \mathbb{R} \to \mathbb{R}$*

$$\frac{1}{n} \sum_{i=1}^{n} g(\lambda_i) \xrightarrow[n \to \infty]{\text{Pr}} \int_0^\infty g(\lambda) \mu(\mathrm{d}\lambda). \tag{3.3}$$

*2. The largest eigenvalue $\lambda_{\boldsymbol{H}}^+$ of $\boldsymbol{H}$ converges to the largest element $\lambda^+$ in the support of $\mu$, i.e. $\lambda_{\boldsymbol{H}}^+ \xrightarrow[n \to \infty]{\text{Pr}} \lambda^+$.*

This assumption is typical in random matrix theory. An important example is *the isotropic features model*, which is a random $n \times d$ matrix $\boldsymbol{A}$ whose every entry is sampled from a common, mean 0, variance $\frac{1}{d}$ distribution with fourth moment $\mathcal{O}(d^{-2})$, such as a Gaussian $N(0, \frac{1}{d})$. In this case, the ESM $\mu_{\boldsymbol{H}}$ of $\boldsymbol{H} = \boldsymbol{A}\boldsymbol{A}^T$ converges to the Marchenko-Pastur law (see Figure 3.1):

$$\mathrm{d}\mu_{\mathrm{MP}}(\lambda) \stackrel{\text{def}}{=} \delta_0(\lambda) \max\{1 - r, 0\} + \frac{r\sqrt{(\lambda - \lambda^-)(\lambda^+ - \lambda)}}{2\pi\lambda} 1_{[\lambda^-, \lambda^+]}, \tag{3.4}$$
$$\text{where} \qquad \lambda^- \stackrel{\text{def}}{=} (1 - \sqrt{\tfrac{1}{r}})^2 \quad \text{and} \quad \lambda^+ \stackrel{\text{def}}{=} (1 + \sqrt{\tfrac{1}{r}})^2 \,.$$

More generally, the convergence of the spectral measure of matrices drawn from a consistent ensemble is well studied in random matrix theory, and for many random matrix ensembles the limiting spectral measure is known. In the machine learning literature, it has been shown that the spectrum of the Hessians of neural networks share characteristics with the limiting spectral distributions found in classical random matrix theory [Behrooz et al., 2019, Dauphin et al., 2014, Granziol et al., 2020, Liao et al., 2020, Martin and Mahoney, 2018, Papyan, 2018, Pennington and B., 2017, Sagun et al., 2016].

**Complexity analysis.** The forcing function and the convolution kernel both converge under Assumption 2, and the result is that

$$\lim_{n \to \infty} \mathbb{E}_{\boldsymbol{H}}[f(\boldsymbol{X}_t)] = \psi(t) \quad \text{where} \quad \psi(t) = F_\mu(t) + \int_0^\infty \mathcal{I}_\mu(t - s)\psi(s) \, \mathrm{d}s \quad \text{for all} \quad t \geq 0.$$

The forcing function and interaction kernel are given as integrals against the limit measure (such as (3.4) in the case of isotropic features) and

$$F_\mu(t) \overset{\text{def}}{=} \int_0^\infty (R\lambda + \widetilde{R})G^{(\lambda)}(t)\mu(\mathrm{d}\lambda) \quad \text{and} \quad \mathcal{I}_\mu(t) \overset{\text{def}}{=} \int_0^\infty K^{(\lambda)}(t)\mu(\mathrm{d}\lambda).$$

The kernel norm still determines the convergence properties of the Volterra equation. In particular, to have convergence of the algorithm, we need that $\|\mathcal{I}_\mu\| < 1$ and just like in the finite-$n$ case (see Lemma C.1)

$$\psi(t) \xrightarrow[t\to\infty]{} \psi(\infty) \overset{\text{def}}{=} \frac{\widetilde{R}\mu(\{0\})}{2(1 - \|\mathcal{I}_\mu\|)}.$$

To discuss average-case rates, we use the function $\psi(t)$. We consider separately the regimes when the problem is strongly convex and not. Having taken the limit, we say the problem is strongly convex if the intersection of the support of $\mu$ with $(0, \infty)$ is closed. Intuitively, this says there is a gap between 0 and the next smallest eigenvalue $\lambda^-$ of the hessian (strictly speaking it allows a vanishing fraction of the eigenvalues to approach 0).

In the non-strongly convex case the average-case complexity is relatively simple to compute. The rate of convergence of $\psi(t) - \psi(\infty) \to 0$ is only determined by the rate of of convergence of the forcing function $F_\mu(t) - F_\mu(\infty)$. This decays like $t^{-\beta}$ where $\beta$ in turn is controlled by the exponent $\alpha$ at which $\mu((0, \varepsilon]) \asymp \epsilon^\alpha$ as $\varepsilon \to 0$ (see Lemma C.2). Particular if there are more small eigenvalues, the rate is slowed. In Table 3, the rates are reported for $\alpha = \frac{1}{2}$. This is the typical behavior for random matrix distributions with a "hard-edge," such as Marchenko–Pastur with aspect ratio $r = 1$.

In the strongly convex case, $\lambda^- > 0$, the kernel $\mathcal{I}_\mu$ plays a larger role, in that it may slow down the convergence rate. In particular, if it exists, we define the *Malthusian exponent* $\lambda^*$ as the solution of

$$\int_0^\infty e^{\lambda^* t} \mathcal{I}_\mu(t) = 1.$$

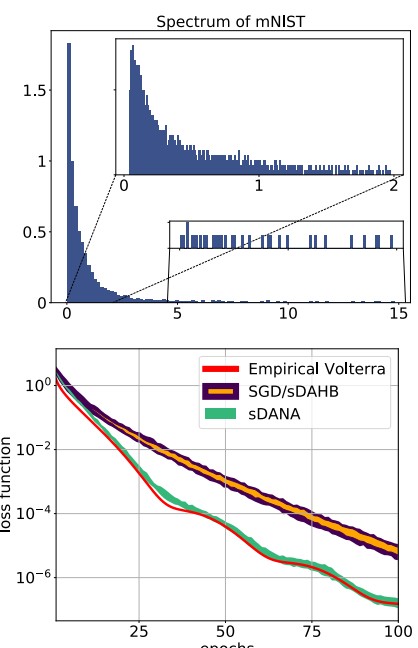

Figure 6: **SDANA & SGD vs Theory on MNIST.** MNIST ($60000 \times 28 \times 28$ images) [LeCun et al., 2010] is reshaped into 10 matrices of dimension $1000 \times 4704$, representing 1000 samples of groups of 6 digits (preconditioned to have centered rows of norm-1). First digit of each 6 is chosen to be the target $b$. Algorithms were run 10 times with default parameters (without tuning) to solve (2.1). 80%–confidence interval is displayed. Volterra (SDANA) is generated with eigenvalues from the first MNIST data matrix (top pane, $\lambda^- = 0.041$). Volterra predicts the convergent behavior of SDANA in this non-idealized setting. SDANA outperforms equivalent SGD/SDAHB. See also Appendix F.

The rate of convergence of $\psi$ to 0 (at exponential scale) will then be the slower of $F_\mu(t)$ and $e^{-\lambda^* t}$ (see Lemma C.3). In the case of SGD on Marchenko–Pastur, the exact value of $\lambda^*$ is worked out in exact form in Paquette et al. [2021]. By bounding these Malthusian exponents, we produce the rate guarantees in Table 3 (see Cor. D.1 and Prop. E.3 for the bounds in the Appendix). The **default parameters** are chosen so that its linear rate is no slower, by a factor of 4 than the fastest possible rate for an algorithm having optimized over all step size choices. This is achieved by lower bounding the Malthusian exponent at the default parameters and upper bounding the optimal rate by minimizing $F_\mu(t) - F_\mu(\infty)$ over all convergent parameters.

**Conclusions from the analysis of homogenized SGD.** The SHB algorithm is a special instance of SDAHB with parameter choices $n\theta^{\text{shb}} = \theta^{\text{sdahb}}$ and $n\gamma^{\text{shb}} = \gamma^{\text{sdahb}}$. By evaluating the kernels

for SDAHB in the large $n$ limit, it is easily seen that the homogenized SGD equations for $\boldsymbol{X}_t^{\text{shb}}$ and $\boldsymbol{X}_t^{\text{sgd}}$ satisfy

$$| \, \mathbb{E}_{\boldsymbol{H}}[f(\boldsymbol{X}_t^{\text{shb}})] - \mathbb{E}_{\boldsymbol{H}}[f(\boldsymbol{X}_t^{\text{sgd}})] | \underset{n\to\infty}{\longrightarrow} 0, \qquad (3.5)$$

see (Thm. 5). On the other hand SDAHB with default parameters is always strictly faster (for sufficiently large $\theta$) than tuned SGD, but its linear rate is never more than a factor of 2 faster than SGD (Prop. E.5). It also does not substantially improve over SGD in non-strongly convex case. In contrast, the dimension adjusted Nesterov acceleration (SDANA) greatly (and provably, using homogenized SGD) improves over SGD in non-strongly convex case (see Prop. D.3), while remaining linear (and nearly as fast as SGD) in the convex case (Cor. D.1). Furthermore, the predictions of homogenized SGD are born out even on real data (Fig. 6), which is a non-idealized setting that does not verify the assumptions we imposed for the theoretical analysis.

**Future directions.** We would like to explore the applicability of homogenized SGD to other datasets and other convex losses as well as generalizing the theoretical setting under which homogenized SGD applies (see the discussion below Thm. 2). Moreover, we would like to test and extend SDANA to non-convex problems and extend homogenized SGD to non-convex settings.

**Funding Transparency Statement.** C. Paquette's research was supported by CIFAR AI Chair, MILA. Research by E. Paquette was supported by a Discovery Grant from the Natural Science and Engineering Council (NSERC). Additional revenues related to this work: C. Paquette has part-time employment at Google Research, Brain Team, Montreal, QC.

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
