# Dynamics of Stochastic Momentum Methods on Large-scale, Quadratic Models

## Supplementary material

The appendix is organized into five sections as follows:

1. Appendix A derives the Volterra equation and proves the main result for the homogenized SGD (Theorem 1).

2. We show in Appendix B a heuristic derivation of the homogenized SGD approximation to the SDA class of algorithms on the least squares problem and we show that SGD and homogenized SGD are close under orthogonal invariance (Theorem 2).

3. We give in Appendix C a general overview of the analysis of a convolution Volterra equation of the type that arises in the SDA class.

4. Appendix D details the analysis of the homogenized SGD for SDANA, including average-case analysis and near optimal parameters.

5. Appendix E has the details showing equivalence of SDAHB with SHB as well as general average-case complexity and parameter selections.

6. Appendix F contains details on the simulations.

Unless otherwise stated, all the results hold under Assumptions 1 and 2. We include all statements from the previous sections for clarity.

**Potential societal impacts.** The results presented in this paper concern the analysis of existing methods and a new method that is a variant of an existing method. The results are theoretical and we do not anticipate any direct ethical and societal issues. We believe the results will be used by machine learning practitioners and we encourage them to use it to build a more just, prosperous world.

## A  Analysis of the Homogenized SGD evolution

### A.1  Homogenized SGD

We recall that the diffusion model is given by

$$\mathrm{d}\boldsymbol{X}_t = -\gamma_2\, \mathrm{d}\boldsymbol{Z}_t - \frac{\gamma_1}{\varphi(t)} \int_0^t \varphi(s)\, \mathrm{d}\boldsymbol{Z}_t, \quad \text{where} \quad \mathrm{d}\boldsymbol{Z}_t = \nabla f(\boldsymbol{X}_t)\, \mathrm{d}t + \sqrt{\tfrac{2}{n} f(\boldsymbol{X}_t)\nabla^2(f)}\, \mathrm{d}\boldsymbol{B}_t.$$

To connect these diffusions to SGD on the least squares problem (2.1)

$$f(\boldsymbol{x}) = \frac{1}{2}\|\boldsymbol{A}\boldsymbol{x} - \boldsymbol{b}\|^2,$$

we will use the singular value decomposition of $\boldsymbol{U}\boldsymbol{\Sigma}\boldsymbol{V}^T$ of $\boldsymbol{A}$. We order the singular values $\sigma_1 \geq \sigma_2 \geq \sigma_3 \cdots$ in decreasing order. We then let $\boldsymbol{\nu}_t = \boldsymbol{V}^T(\boldsymbol{X}_t - \widetilde{\boldsymbol{x}})$, where we recall that $\boldsymbol{b} = \boldsymbol{A}\widetilde{\boldsymbol{x}} + \boldsymbol{\eta}$. We recall that

$$\nabla f(\boldsymbol{X}_t) = \boldsymbol{A}^T(\boldsymbol{A}\boldsymbol{X}_t - \boldsymbol{b}) \quad \text{and} \quad \nabla^2 f = \boldsymbol{A}^T\boldsymbol{A}.$$

Hence, we may change the basis to write

$$\mathrm{d}(\boldsymbol{V}^T\boldsymbol{X}_t) = -\gamma_2\, \mathrm{d}(\boldsymbol{V}^T\boldsymbol{Z}_t) - \frac{\gamma_1}{\varphi(t)} \int_0^t \varphi(s)\, \mathrm{d}(\boldsymbol{V}^T\boldsymbol{Z}_t),$$

$$\mathrm{d}(\boldsymbol{V}^T\boldsymbol{Z}_t) = \boldsymbol{\Sigma}^T(\boldsymbol{\Sigma}\boldsymbol{\nu}_t - \boldsymbol{U}^T\boldsymbol{b})\, \mathrm{d}t + \sqrt{\tfrac{2}{n} f(\boldsymbol{X}_t)\boldsymbol{\Sigma}^T\boldsymbol{\Sigma}}\, \mathrm{d}(\boldsymbol{V}^T\boldsymbol{B}_t).$$

The loss values we may also represent in terms of $\boldsymbol{\nu}$

$$f(\boldsymbol{X}_t) = \frac{1}{2}\|\boldsymbol{A}\boldsymbol{X}_t - \boldsymbol{b}\|^2 = \frac{1}{2}\|\boldsymbol{\Sigma}\boldsymbol{\nu}_t - \boldsymbol{U}^T\boldsymbol{\eta}\|^2 = \frac{1}{2}\sum_{j=1}^d (\sigma_j\nu_{t,j} - (\boldsymbol{U}^T\boldsymbol{b})_j)^2.$$

We let $\mathrm{d}\boldsymbol{W}_t = \sqrt{\frac{2}{n} f(\boldsymbol{X}_t) \boldsymbol{\Sigma}^T \boldsymbol{\Sigma}} \, \mathrm{d}(\boldsymbol{V}^T \boldsymbol{B}_t)$, so that $\{W_{t,j} : t \geq 0, j \in 1, 2, \ldots, n\}$ are a family of continuous martingales with quadratic variation

$$\mathrm{d}\langle W_{t,j}, W_{t,i} \rangle = \delta_{i,j} \frac{2\sigma_j^2}{n} f(\boldsymbol{X}_t) \, \mathrm{d}t \tag{A.1}$$

for all $i$ and $j$, with $1 \leq i, j \leq n$. Finally, we conclude that

$$\mathrm{d}\nu_{t,j} = -\gamma_2 \, \mathrm{d}\xi_{t,j} - \frac{\gamma_1}{\varphi(t)} \int_0^t \varphi(s) \, \mathrm{d}\xi_{s,j} \quad \text{where} \quad \mathrm{d}\xi_{t,j} := \mathrm{d}W_{t,j} + \sigma_j^2 \big( \nu_{t,j} - \frac{(\boldsymbol{U}^T \boldsymbol{\eta})_j}{\sigma_j} \big) \, \mathrm{d}t. \tag{A.2}$$

As in (A.1), the quadratic variation of $W_{t,j}$ and $\xi_{t,j}$ is

$$\mathrm{d}\langle \xi_{t,j} \rangle = \gamma_2^{-2} \, \mathrm{d}\langle \nu_{t,j} \rangle = \mathrm{d}\langle W_{t,j} \rangle = \frac{2\sigma_j^2 f(\boldsymbol{X}_t)}{n} \, \mathrm{d}t. \tag{A.3}$$

## A.2 Mean behavior of the homogenized SGD

We derive a description for the mean of the loss values $\mathbb{E}_{\boldsymbol{H}} f(\boldsymbol{X}_t)$. We define the following functions of time

$$J \stackrel{\text{def}}{=} \mathbb{E}_{\boldsymbol{H}} \bigg[ \big(\nu_{t,j} - \tfrac{(\boldsymbol{U}^T \boldsymbol{\eta})_j}{\sigma_j}\big)^2 \bigg] \quad \text{and} \quad N \stackrel{\text{def}}{=} \sigma_j^{-2} \mathbb{E}_{\boldsymbol{H}} \bigg[ \Big( \int_0^t \varphi(s) d\xi_{s,j} \Big)^2 - \int_0^t \varphi^2(s) \, \mathrm{d}\langle \xi_{s,j} \rangle \bigg]. \tag{A.4}$$

We will compute the derivatives of these expressions in time. Using Itô's rule,

$$\begin{aligned}
\mathrm{d}\big(\nu_{t,j} - \tfrac{(\boldsymbol{U}^T \boldsymbol{\eta})_j}{\sigma_j}\big)^2 &= 2\big(\nu_{t,j} - \tfrac{(\boldsymbol{U}^T \boldsymbol{\eta})_j}{\sigma_j}\big) \, \mathrm{d}\nu_{t,j} + \mathrm{d}\langle \nu_{t,j} \rangle \\
&= 2\big(\nu_{t,j} - \tfrac{(\boldsymbol{U}^T \boldsymbol{\eta})_j}{\sigma_j}\big) \Big( -\gamma_2 \, \mathrm{d}\xi_{t,j} - \frac{\gamma_1}{\varphi(t)} \int_0^t \varphi(s) \, \mathrm{d}\xi_{s,j} \Big) + \gamma_2^2 \, \mathrm{d}\langle \xi_{t,j} \rangle.
\end{aligned} \tag{A.5}$$

Since $\mathrm{d}W_{t,j}$ is a martingale increment, the expectation of the $\mathrm{d}\xi_{t,j}$ term simplifies. We may do a similar computation with $N$ and conclude that:

$$J^{(1)} = -2\gamma_2 \sigma_j^2 J - \frac{2\gamma_1}{\varphi(t)} \mathbb{E}_{\boldsymbol{H}} \bigg[ \big(\nu_{t,j} - \tfrac{(\boldsymbol{U}^T \boldsymbol{\eta})_j}{\sigma_j}\big) \int_0^t \varphi(s) \, \mathrm{d}\xi_{s,j} \bigg] + \gamma_2^2 \, \mathbb{E}_{\boldsymbol{H}} \, \mathrm{d}\langle \xi_{t,j} \rangle,$$

$$N^{(1)} = \sigma_j^{-2} \mathbb{E}_{\boldsymbol{H}} \bigg[ 2\varphi(t) \, \mathrm{d}\xi_{t,j} \int_0^t \varphi(s) \, \mathrm{d}\xi_{s,j} \bigg] = \mathbb{E}_{\boldsymbol{H}} \bigg[ 2\varphi(t) \big(\nu_{t,j} - \tfrac{(\boldsymbol{U}^T \boldsymbol{\eta})_j}{\sigma_j}\big) \int_0^t \varphi(s) \, \mathrm{d}\xi_{s,j} \bigg]$$

In summary, we may express $J$ in terms of $N$ by

$$J^{(1)} = -2\gamma_2 \sigma_j^2 J - \frac{\gamma_1}{\varphi^2(t)} N^{(1)} + \gamma_2^2 \, \mathrm{d}\langle \xi_{t,j} \rangle \quad \text{with} \quad J(0) = \mathbb{E}_{\boldsymbol{H}} \bigg[ \big(\nu_{0,j} - \tfrac{(\boldsymbol{U}^T \boldsymbol{\eta})_j}{\sigma_j}\big)^2 \bigg]. \tag{A.6}$$

Now we write a differential equation for $N$, using the product rule for stochastic calculus, and conclude

$$\begin{aligned}
\varphi \tfrac{\mathrm{d}}{\mathrm{d}t} \big(N^{(1)}/\varphi\big) = &- 2\varphi(t) \, \mathbb{E}_{\boldsymbol{H}} \bigg[ \Big( \gamma_2 \, \mathrm{d}\xi_{t,j} + \frac{\gamma_1}{\varphi(t)} \int_0^t \varphi(s) \, \mathrm{d}\xi_{s,j} \Big) \int_0^t \varphi(s) \, \mathrm{d}\xi_{s,j} \bigg] \\
&+ 2\varphi(t) \, \mathbb{E}_{\boldsymbol{H}} \bigg[ \Big( \nu_{t,j} - \tfrac{(\boldsymbol{U}^T \boldsymbol{\eta})_j}{\sigma_j} \Big) \varphi(t) \, \mathrm{d}\xi_{t,j} \bigg] + 2\varphi^2(t) \, \mathbb{E}_{\boldsymbol{H}} \langle \mathrm{d}\nu_{t,j}, \mathrm{d}\xi_{t,j} \rangle \\
= &- \gamma_2 \sigma_j^2 N^{(1)} - 2\gamma_1 \big( \sigma_j^2 N + \int_0^t \varphi^2 \, \mathrm{d}\langle \xi_{t,j} \rangle \big) + 2\varphi^2(t) \sigma_j^2 J - 2\gamma_2 \varphi^2(t) \, \mathbb{E}_{\boldsymbol{H}} \, \mathrm{d}\langle \xi_{t,j} \rangle,
\end{aligned}$$

with initial conditions $N(0) = N^{(1)}(0) = 0$. We will use

$$\widehat{J} = \varphi^2 J / \gamma_1 \quad \text{and} \quad \widehat{\psi} = 2\sigma_j^2 \varphi^2 \, \mathbb{E}_{\boldsymbol{H}} f(\boldsymbol{X}_t)/n = \varphi^2 \, \mathbb{E}_{\boldsymbol{H}} \, \mathrm{d}\langle \xi_{t,j} \rangle. \tag{A.7}$$

From these definitions we can also record, by evaluating the previous displayed equation at 0 that $N^{(2)}(0) = 2\gamma_1 \sigma_j^2 \widehat{J}(0) - 2\gamma_2 \widehat{\psi}(0)$. The $\widehat{J}$ can be expressed as

$$\widehat{J}\big(-2\Phi + 2\gamma_2 \sigma_j^2\big) + \widehat{J}^{(1)} = -N^{(1)} + \tfrac{\gamma_2^2}{\gamma_1} \widehat{\psi} \quad \text{where} \quad \Phi \stackrel{\text{def}}{=} \frac{\varphi'(t)}{\varphi(t)}.$$

We then differentiate the display equation above to produce

$$N^{(3)} + N^{(2)}\left(-\Phi + \gamma_2\sigma_j^2\right) + N^{(1)}\left(-\Phi' + 2\gamma_1\sigma_j^2\right) - 2\gamma_1\sigma_j^2\widehat{J}^{(1)} = -2\gamma_1\widehat{\psi} - 2\gamma_2\widehat{\psi}^{(1)}.$$

On substituting $\widehat{J}$, we arrive at the third-order differential equation

$$\begin{aligned}
\widehat{J}^{(3)} &+ \left(-3\Phi + 3\gamma_2\sigma_j^2\right)\widehat{J}^{(2)} + \left(-5\Phi^{(1)} + 2\Phi^2 - 4\gamma_2\sigma_j^2\Phi + 4\gamma_1\sigma_j^2 + 2\gamma_2^2\sigma_j^4\right)\widehat{J}^{(1)} \\
&+ \left(-2\Phi^{(2)} + 4\Phi\Phi^{(1)} - 4\gamma_2\sigma_j^2\Phi^{(1)} - 4\gamma_1\sigma_j^2\Phi + 4\gamma_1\gamma_2\sigma_j^4\right)\widehat{J} \\
&= \tfrac{\gamma_2^2}{\gamma_1}\widehat{\psi}^{(2)} + \left(2\gamma_2 + \left(-\Phi + \gamma_2\sigma_j^2\right)\tfrac{\gamma_2^2}{\gamma_1}\right)\widehat{\psi}^{(1)} + \left(2\gamma_1 + \left(-\Phi^{(1)} + 2\gamma_1\sigma_j^2\right)\tfrac{\gamma_2^2}{\gamma_1}\right)\widehat{\psi}.
\end{aligned} \tag{A.8}$$

The initial conditions are given by

$$\widehat{J}(0) = \gamma_1^{-1}\mathbb{E}\left[\left(\nu_{0,j} - \tfrac{(\boldsymbol{U}^T\boldsymbol{\eta})_j}{\sigma_j}\right)^2\right], \quad \widehat{J}^{(1)}(0) = \tfrac{\gamma_2^2}{\gamma_1}\widehat{\psi}(0) - \widehat{J}(0)\left(-2\Phi(0) + 2\gamma_2\sigma_j^2\right), \quad \text{and}$$

$$\widehat{J}^{(2)}(0) = \tfrac{\gamma_2^2}{\gamma_1}\widehat{\psi}^{(1)}(0) + 2\gamma_2\widehat{\psi}(0) - 2\gamma_1\sigma_j^2\widehat{J}(0) + \left(2\Phi(0) - 2\gamma_2\sigma_j^2\right)\widehat{J}^{(1)}(0) + 2\Phi^{(1)}(0)\widehat{J}(0). \tag{A.9}$$

**Two special cases for $\Delta(k,n)$.** In this section, we record the ODE for two special cases of the function $\varphi(t)$. When $\Delta(k,n) = \tfrac{\theta}{k+n}$ and thus $\varphi(t) = (1+t)^\theta$ with $\Phi(t) = \tfrac{\theta}{1+t}$, the corresponding ODE is precisely

$$\begin{aligned}
\widehat{J}^{(3)} &- \left(\tfrac{3\theta}{(1+t)} - 3\gamma_2\sigma_j^2\right)\widehat{J}^{(2)} - \left(-\tfrac{5\theta + 2\theta^2}{(1+t)^2} + \tfrac{4\gamma_2\sigma_j^2\theta}{(1+t)} - 4\gamma_1\sigma_j^2 - 2\gamma_2^2\sigma_j^4\right)\widehat{J}^{(1)} \\
&- \left(\tfrac{4\theta + 4\theta^2}{(1+t)^3} - \tfrac{4\gamma_2\sigma_j^2\theta}{(1+t)^2} + \tfrac{4\gamma_1\sigma_j^2\theta}{(1+t)} - 4\gamma_1\gamma_2\sigma_j^4\right)\widehat{J} \\
&= \tfrac{\gamma_2^2}{\gamma_1}\widehat{\psi}^{(2)} + \left(2\gamma_2 + \left(\tfrac{-\theta}{(1+t)} + \gamma_2\sigma_j^2\right)\tfrac{\gamma_2^2}{\gamma_1}\right)\widehat{\psi}^{(1)} + \left(2\gamma_1 + \left(\tfrac{\theta}{(1+t)^2} + 2\gamma_1\sigma_j^2\right)\tfrac{\gamma_2^2}{\gamma_1}\right)\widehat{\psi}.
\end{aligned} \tag{A.10}$$

and the initial conditions are given by

$$\widehat{J}(0) = \gamma_1^{-1}\mathbb{E}\left[\left(\nu_{0,j} - \tfrac{(\boldsymbol{U}^T\boldsymbol{\eta})_j}{\sigma_j}\right)^2\right], \quad \widehat{J}^{(1)}(0) = \tfrac{\gamma_2^2}{\gamma_1}\widehat{\psi}(0) - \widehat{J}(0)\left(-2\theta + 2\gamma_2\sigma_j^2\right), \quad \text{and}$$

$$\widehat{J}^{(2)}(0) = \tfrac{\gamma_2^2}{\gamma_1}\widehat{\psi}^{(1)}(0) + 2\gamma_2\widehat{\psi}(0) - 2\gamma_1\sigma_j^2\widehat{J}(0) + \left(2\theta - 2\gamma_2\sigma_j^2\right)\widehat{J}^{(1)}(0) - 2\theta\widehat{J}(0). \tag{A.11}$$

The other case is when $\Delta(k,n) = \tfrac{\theta}{n}$, or $\varphi(t) = \exp(\theta t)$. We call this the general SDAHB; one recovers SDAHB when $\gamma_1 = \gamma, \gamma_2 = 0$, and $\theta = \theta$. In this setting, the log-derivative $\Phi(t) = \alpha$ and the ODE reduces to

$$\begin{aligned}
\widehat{J}^{(3)} &+ \left(-3\theta + 3\gamma_2\sigma_j^2\right)\widehat{J}^{(2)} + \left(2\theta^2 - 4\gamma_2\sigma_j^2\theta + 4\gamma_1\sigma_j^2 + 2\gamma_2^2\sigma_j^4\right)\widehat{J}^{(1)} \\
&+ \left(-4\gamma_1\sigma_j^2\theta + 4\gamma_1\gamma_2\sigma_j^4\right)\widehat{J} \\
&= \tfrac{\gamma_2^2}{\gamma_1}\widehat{\psi}^{(2)} + \left(2\gamma_2 + \left(-\theta + \gamma_2\sigma_j^2\right)\tfrac{\gamma_2^2}{\gamma_1}\right)\widehat{\psi}^{(1)} + \left(2\gamma_1 + 2\gamma_1\sigma_j^2\tfrac{\gamma_2^2}{\gamma_1}\right)\widehat{\psi}.
\end{aligned} \tag{A.12}$$

The initial conditions are given by

$$\widehat{J}(0) = \gamma_1^{-1}\mathbb{E}\left[\left(\nu_{0,j} - \tfrac{(\boldsymbol{U}^T\boldsymbol{\eta})_j}{\sigma_j}\right)^2\right], \quad \widehat{J}^{(1)}(0) = \tfrac{\gamma_2^2}{\gamma_1}\widehat{\psi}(0) - \widehat{J}(0)\left(-2\theta + 2\gamma_2\sigma_j^2\right), \quad \text{and}$$

$$\widehat{J}^{(2)}(0) = \tfrac{\gamma_2^2}{\gamma_1}\widehat{\psi}^{(1)}(0) + 2\gamma_2\widehat{\psi}(0) - 2\gamma_1\sigma_j^2\widehat{J}(0) + \left(2\theta - 2\gamma_2\sigma_j^2\right)\widehat{J}^{(1)}(0). \tag{A.13}$$

We note that the ODE in (A.12) is constant coefficient and therefore can be solved by finding the characteristic polynomial, that is,

$$0 = \lambda^3 + (3\gamma_2\sigma_j^2 - 3\theta)\lambda^2 + (2\theta^2 - 4\gamma_2\sigma_j^2\theta + 4\gamma_1\sigma_j^2 + 2\gamma_2^2\sigma_j^4)\lambda + 4\gamma_1\gamma_2\sigma_j^4 - 4\gamma_1\sigma_j^2\theta,$$

$$0 = (\lambda + \sigma_j^2\gamma_2 - \theta)(\lambda^2 + (2\sigma_j^2\gamma_2 - 2\theta)\lambda + 4\sigma_j^2\gamma_1),$$

$$\lambda = \theta - \sigma_j^2\gamma_2 \quad \text{and} \quad \lambda = -(\sigma_j^2\gamma_2 - \theta) \pm \sqrt{(\sigma_j^2\gamma_2 - \theta)^2 - 4\sigma_j^2\gamma_1}.$$

## A.3 Inhomogeneous IVP in (A.8)

We simplify the problem in (A.8) by considering the inhomogeneous ODE

$$L[\widehat{J}] := \widehat{J}^{(3)} + p(t)\widehat{J}^{(2)} + q(t)\widehat{J}^{(1)} + r(t)\widehat{J} = C\widehat{\psi}^{(2)} + f(t)\widehat{\psi}^{(1)} + g(t)\widehat{\psi} =: R[\widehat{\psi}], \qquad \text{(A.14)}$$

where $L[\widehat{J}]$ and $R[\widehat{\psi}]$ are differential operators. Let $J_0(t)$ be the solution to the homogeneous ODE in (A.14) (*i.e.* $L[J_0] = 0$) with initial conditions given by $J_0(0) = d_0$, $J_0^{(1)}(0) = d_1$ and $J_0^{(2)}(0) = d_2$. We let $\widehat{K}_s(t)$ solve

$$\widehat{K}_s(t) = 0 \text{ for } t < s, \;\; L[\widehat{K}_s(t)] = 0 \text{ for } t \neq s, \text{ and } \widehat{K}_s(s) = c_0, \;\; \widehat{K}_s^{(1)}(s) = c_1, \;\; \widehat{K}_s^{(2)}(s) = c_2. \tag{A.15}$$

Here the initial conditions are chosen so that $L[\widehat{K}_s(t)] = R^*[\delta_s(t)]$, with $R^*$ the adjoint differential operator, *i.e.*,

$$L\Big[\int_0^\infty \widehat{K}_s(t)\widehat{\psi}(s)\,\mathrm{d}s\Big](t) = \int_0^\infty L[\widehat{K}_s(t)]\widehat{\psi}(s)\,\mathrm{d}s = \int_0^\infty R^*[\delta_s(t)]\widehat{\psi}(s)\,\mathrm{d}s = R[\widehat{\psi}](t).$$

We now just need to determine the initial conditions $c_0$, $c_1$, and $c_2$. First, we define $H_s(t)$ to be the Heaviside function with a jump at $s$ and note the following classical results for derivatives of $H_s(t)$:

$$\partial_t\big(\tfrac{(t-s)^2}{2}H_s(t)\big) = (t-s)H_s(t), \quad \partial_t((t-s)H_s(t)) = H_s(t)$$

$$\partial_t H_s(t) = \delta_s(t), \quad \partial_t^2 H_s(t) = \delta_s'(t).$$

We now define the following operator where the derivatives are taken with respect to $t$

$$R^*[\delta_s](t) \stackrel{\text{def}}{=} C\delta_s''(t) + f(t)\delta_s'(t) + g(t)\delta_s(t)$$

$$\int_0^\infty R^*[\delta_s(t)]\widehat{\psi}(s)\,\mathrm{d}s = \frac{d}{dt^2}\int_0^\infty C\delta_s(t)\widehat{\psi}(s)\,\mathrm{d}s + f(t)\frac{d}{dt}\int_0^\infty \delta_s(t)\widehat{\psi}(s)\,\mathrm{d}s + g(t)\int_0^\infty \delta_s(t)\widehat{\psi}(s)\,\mathrm{d}s$$

$$= C\widehat{\psi}^{(2)}(t) + f(t)\widehat{\psi}^{(1)}(t) + g(t)\widehat{\psi}(t) = F(t).$$

We now decompose $\widehat{K}_s(t) = \widehat{K}_s^1(t) + \widehat{K}_s^2(t) + \widehat{K}_s^3(t)$ and find initial conditions for each of these terms separately, that is, we will find

$$L[\widehat{K}_s^1(t)] = C\delta_s''(t), \quad L[\widehat{K}_s^2(t)] = f(t)\delta_s'(t), \quad \text{and} \quad L[\widehat{K}_s^3(t)] = g(t)\delta_s(t).$$

We recall for clarity that $f(t)\delta_s'(t) = f(s)\delta_s'(t) - f'(s)\delta_s(t)$. We can write $\widehat{K}_s^i = \widetilde{K}_s^i + \widetilde{H}_s^i$ where $\widetilde{K}_s^i$ is 0 at $s$ and $C^2$ and

$$\widetilde{H}_s^i \stackrel{\text{def}}{=} H_s^i(t)\big(c_0^i + c_1^i(t-s) + c_2^i \tfrac{(t-s)^2}{2}\big).$$

It follows that $L[\widetilde{H}_s^1] = C\delta_s''(t) + \{\text{continuous functions on } t \geq s\}$. To find $c_0^1, c_1^1$, and $c_2^1$, we see that

$$L[\widetilde{H}_s^1] = c_0^1\delta_s''(t) + c_1^1\delta_s'(t) + c_2^1\delta_s(t) + p(t)[c_0^1\delta_s'(t) + c_1^1\delta_s(t) + c_2^1 H_s(t)]$$

$$+ q(t)\big[c_0^1\delta_s(t) + c_1^1 H_s(t) + c_2^1\int H_s(t)\,\mathrm{d}t\big]$$

$$+ r(t)\big[c_0^1 H_s(t) + c_1^1\int H_s(t)\,\mathrm{d}t + c_2^1\int\int H_s(t)\,\mathrm{d}t\,\mathrm{d}t\big]$$

$$= c_0^1\delta_s''(t) + c_1^1\delta_s'(t) + c_2^1\delta_s(t) + c_0^1\big(p(s)\delta_s'(t) - p'(s)\delta_s(t)\big) + c_1^1 p(s)\delta_s(t)$$

$$+ c_0^1 q(s)\delta_s(t) + \text{continuous terms}$$

As we want $L[\widetilde{H}_s^1] = C\delta_s''(t)$, then we need to solve the system

$$\begin{pmatrix} C \\ 0 \\ 0 \end{pmatrix} = \begin{pmatrix} 1 & 0 & 0 \\ p(s) & 1 & 0 \\ q(s) - p'(s) & p(s) & 1 \end{pmatrix} \begin{pmatrix} c_0^1 \\ c_1^1 \\ c_2^1 \end{pmatrix}.$$

We can know solve this system to get that

$$c_0^1 = C, \quad c_1^1 = -Cp(s), \quad \text{and} \quad c_2^1 = Cp^2(s) - C(q(s) - p'(s)). \tag{A.16}$$

Next we want to solve $L[\widetilde{H}_s^2] = f(t)\delta_s'(t) = f(s)\delta_s'(t) - f'(s)\delta_s(t)$. Using a similar argument as before, we deduce that

$$\begin{pmatrix} 0 \\ f(s) \\ -f'(s) \end{pmatrix} = \begin{pmatrix} 1 & 0 & 0 \\ p(s) & 1 & 0 \\ q(s) - p'(s) & p(s) & 1 \end{pmatrix} \begin{pmatrix} c_0^2 \\ c_1^2 \\ c_2^2 \end{pmatrix}.$$

Solving this system,

$$c_0^2 = 0, \quad c_1^2 = f(s), \quad \text{and} \quad c_2^2 = -f'(s) - p(s)f(s). \tag{A.17}$$

Lastly we want to solve $L[\widetilde{H}_s^3] = g(s)\delta_s(t)$ or equivalently,

$$\begin{pmatrix} 0 \\ 0 \\ g(s) \end{pmatrix} = \begin{pmatrix} 1 & 0 & 0 \\ p(s) & 1 & 0 \\ q(s) - p'(s) & p(s) & 1 \end{pmatrix} \begin{pmatrix} c_0^3 \\ c_1^3 \\ c_2^3 \end{pmatrix},$$

that is

$$c_0^3 = 0, \quad c_1^3 = 0, \quad \text{and} \quad c_2^3 = g(s). \tag{A.18}$$

Putting this all together, we need to solve for $\widehat{K}_s(t)$ such that

$$L[\widehat{K}_s(t)] = 0$$
$$\text{where} \quad \widehat{K}_s(s) = C, \quad \widehat{K}_s'(s) = f(s) - Cp(s), \tag{A.19}$$
$$\text{and} \quad \widehat{K}_s''(s) = Cp^2(s) - C(q(s) - p'(s)) - f'(s) - p(s)f(s) + g(s)$$

**Proposition A.1** (Kernel representation, general). *Consider the inhomogeneous ODE in* (A.8). *Let $\widehat{K}_s(t)$ and $\widehat{J}_0(t)$ solve the homogeneous ODE in* (A.8)*, that is,*

1. $L[\widehat{K}_s(t)] = 0$ where $\widehat{K}_s(s) = \frac{\gamma_2^2}{\gamma_1}$, $\widehat{K}_s'(s) = 2\gamma_2 + \frac{2\gamma_2^2}{\gamma_1}\left(\Phi(s) - \gamma_2\sigma_j^2\right)$, and
   $\widehat{K}_s''(s) = 2(\gamma_1 + 3\gamma_2\Phi(s) - 4\gamma_2^2\sigma_j^2) + \frac{2\gamma_2^2}{\gamma_1}\left[\Phi^{(1)}(s) + 2\Phi^2(s) - 4\gamma_2\sigma_j^2\Phi(s) + 2\gamma_2^2\sigma_j^4\right].$

2. $L[\widehat{J}_0(t)] = 0$ where $\widehat{J}_0(0) = \frac{1}{\gamma_1}\mathbb{E}\left[\left(\nu_{0,j} - \frac{(U^T\eta)_j}{\sigma_j}\right)^2\right]$, $\widehat{J}_0^{(1)}(0) = (2\Phi(0) - 2\gamma_2\sigma_j^2)\widehat{J}_0(0)$
   and $\widehat{J}_0^{(2)}(0) = \left((2\Phi(0) - 2\gamma_2\sigma_j^2)^2 - 2\gamma_1\sigma_j^2 + 2\Phi^{(1)}(0)\right)\widehat{J}_0(0).$

*Then the solution to the inhomogeneous ODE in* (A.8) *is given by*

$$\widehat{J}(t) = \widehat{J}_0(t) + \int_0^t \widehat{K}_s(t)\widehat{\psi}(s)\,\mathrm{d}s.$$

*Proof.* This is a direct application of (A.19) with coefficients defined by (A.14) to the ODE in (A.8). $\square$

This leads immediately to a general representation of the kernel and forcing terms for homogenized SGD, which we summarize in the following theorem.

**Theorem 4.** *The homogenized SGD diffusion loss values satisfy*

$$\mathbb{E}_H[f(X_t)] = F(t) + \int_0^t \mathcal{K}_s(t)\,\mathbb{E}_H[f(X_s)]\,\mathrm{d}s \quad \text{for all} \quad t \geq 0.$$

*The forcing function $F$ and the kernel $\mathcal{K}$ are given by*

$$F(t) = \frac{1}{n}\sum_{i=1}^n (R\lambda_i + \widetilde{R})G^{(\lambda_i)}(t) \quad \text{and} \quad \mathcal{K}_s(t) = \frac{1}{n}\sum_{i=1}^n K_s^{(\lambda_i)}(t).$$

The function $G^{(\lambda)}(t)$ and $K_s^{(\lambda)}(t)$ are solutions of a differential equation, where if $\lambda = 0$ then $G^{(\lambda)}(t) = 1$ and $K_s^{(\lambda)}(t) = 0$. Define the differential operator

$$L^{(\lambda)}[\widehat{J}] = \widehat{J}^{(3)} + \left(-3\Phi + 3\gamma_2\lambda\right)\widehat{J}^{(2)} + \left(-5\Phi^{(1)} + 2\Phi^2 - 4\gamma_2\lambda\Phi + 4\gamma_1\lambda + 2\gamma_2^2\lambda^2\right)\widehat{J}^{(1)}$$
$$+ \left(-2\Phi^{(2)} + 4\Phi\Phi^{(1)} - 4\gamma_2\lambda\Phi^{(1)} - 4\gamma_1\lambda\Phi + 4\gamma_1\gamma_2\lambda^2\right)\widehat{J}.$$

*Then the interaction kernel is given by*

$$K_s^{(\lambda)}(t) = \frac{\lambda^2\varphi^2(s)\widehat{K}_s(t)}{\varphi^2(t)} \quad \text{where} \quad L^{(\lambda)}[\widehat{K}_s](t) = 0, \ t \geq s, \ \widehat{K}_s(t) = 0, \ t < s, \quad \text{and}$$
$$\widehat{K}_s(s) = \gamma_2^2, \quad \widehat{K}_s'(s) = 2\gamma_2\gamma_1 + 2\gamma_2^2\left(\Phi(s) - \gamma_2\lambda\right), \quad \text{and}$$
$$\widehat{K}_s''(s) = 2\gamma_1(\gamma_1 + 3\gamma_2\Phi(s) - 4\gamma_2^2\lambda) + 2\gamma_2^2\left[\Phi^{(1)}(s) + 2\Phi^2(s) - 4\gamma_2\lambda\Phi(s) + 2\gamma_2^2\lambda^2\right].$$

*The forcing kernel is given by*

$$G^{(\lambda)}(t) = \frac{\widehat{J}_0(t)}{2\varphi^2(t)} \quad \text{where} \quad L^{(\lambda)}[\widehat{J}_0] = 0, \quad t \geq 0, \quad \text{and}$$

$$\widehat{J}_0(0) = 1, \ \widehat{J}_0^{(1)}(0) = (2\Phi(0) - 2\gamma_2\lambda), \ \text{and} \ \widehat{J}_0^{(2)}(0) = \left((2\Phi(0) - 2\gamma_2\lambda)^2 - 2\gamma_1\lambda + 2\Phi^{(1)}(0)\right).$$

*Proof.* Using the results derived so far, we now formulate the autonomous Volterra equation for the loss under homogenized SGD $\mathbb{E}\, f(\boldsymbol{X}_t)$. We recall that for the least squares problem we have taking expectation (conditioning on the singular values $\boldsymbol{\Sigma}$)

$$\mathbb{E}_{\boldsymbol{H}}[f(\boldsymbol{X}_t)] = \frac{1}{2}\,\mathbb{E}_{\boldsymbol{H}}\,\|\boldsymbol{\Sigma}\boldsymbol{\nu}_t - (\boldsymbol{U}^t\boldsymbol{\eta})\|^2 = \frac{1}{2}\sum_{j=1}^n \sigma_j^2\,\mathbb{E}_{\boldsymbol{H}}\left(\nu_{t,j} - \frac{(\boldsymbol{U}^t\boldsymbol{\eta})_j}{\sigma_j}\right)^2 + \frac{1}{2}\sum_{j=d}^n \mathbb{E}_{\boldsymbol{H}}(\boldsymbol{U}^T\boldsymbol{\eta})_j^2,$$

where the second sum is empty when $n < d$. Recall that $J = J^{(\sigma_j^2)}$ (A.4) gives the expectation of $\mathbb{E}_{\boldsymbol{H}}\left(\nu_{t,j} - \frac{(\boldsymbol{U}^t\boldsymbol{\eta})_j}{\sigma_j}\right)^2$ and hence

$$\mathbb{E}_{\boldsymbol{H}}\, f(\boldsymbol{X}_t) = \frac{1}{2}\sum_{j=1}^n \sigma_j^2 J^{(\sigma_j^2)}(t) + \frac{\widetilde{R}\min\{n - d, 0\}}{2n}.$$

Using (A.7)

$$\widehat{J} = \varphi^2 J/\gamma_1 \quad \text{and} \quad \widehat{\psi} = 2\sigma_j^2\varphi^2\,\mathbb{E}\, f(\boldsymbol{X}_t)/n.$$

The term $n\sigma_j^2\widehat{J}$ has as initial conditions $\sigma_j^2 R + \widetilde{R}$ (when $\sigma_j^2 = 0$, the process $\widehat{J}$ is constant). We conclude that

$$\mathbb{E}_{\boldsymbol{H}}\, f(\boldsymbol{X}_t) = \frac{1}{n}\sum_{j=1}^n \frac{\gamma_1 n\sigma_j^2\widehat{J}^{(\sigma_j^2)}}{2\varphi^2(t)}.$$

From Proposition A.1,

$$\mathbb{E}_{\boldsymbol{H}}\, f(\boldsymbol{X}_t) = \frac{1}{n}\sum_{j=1}^n \frac{\gamma_1 n\sigma_j^2\widehat{J}_0^{(\sigma_j^2)}(t)}{2\varphi^2(t)} + \int_0^t \frac{1}{n}\sum_{j=1}^n \frac{\gamma_1\sigma_j^4\varphi^2(s)\widehat{K}_s^{(\sigma_j^2)}(t)\cdot\mathbb{E}_{\boldsymbol{H}}\, f(\boldsymbol{X}_s)}{\varphi^2(t)}\,\mathrm{d}s.$$

After defining $G^{(\lambda)}$ and $K^{(\lambda)}$ as in the statement of the Theorem, this completes the proof.

$\square$

We now give an explicit expressions for the kernel in two specific cases.

**Corollary A.1** (Kernel representation, SDANA). *Consider the inhomogeneous ODE in* (A.10). *Define the differential operators with* $\lambda \overset{def}{=} \sigma_j^2$

1. $L[\widehat{K}_s](t) = 0, \ t \geq s, \ \widehat{K}_s(t) = 0, \ t < s$
   where $\widehat{K}_s(s) = \gamma_2^2, \quad \widehat{K}_s'(s) = 2\gamma_2\gamma_1 + 2\gamma_2^2\left(\frac{\theta}{1+s} - \gamma_2\lambda\right)$, and
   $\widehat{K}_s''(s) = \gamma_2^2\left[\frac{4\theta^2 - 2\theta}{(1+s)^2} - \frac{8\theta\gamma_2\lambda}{1+s} + 4\lambda^2\gamma_2^2\right] + \gamma_1(2\gamma_1 - 8\gamma_2^2\lambda + \frac{6\theta\gamma_2}{1+s})$.

2. $L[\widehat{J}_0(t)] = 0$

 where $\widehat{J}_0(0) = 1$, $\quad \widehat{J}_0^{(1)}(0) = 2\theta - 2\gamma_2\lambda$, $\quad$ and $\quad \widehat{J}_0^{(2)}(0) = (2\theta - 2\gamma_2\lambda)^2 - 2\gamma_1\sigma_j^2 - 2\theta$.

*Then one has that*

$$K_s^{(\lambda)}(t) = \frac{\lambda^2(1+s)^{2\theta}\widehat{K}_s(t)}{(1+t)^{2\theta}} \quad and \quad G^{(\lambda)}(t) = \frac{\widehat{J}_0(t)}{2(1+t)^{2\theta}}.$$

**Corollary A.2** (Kernel representation, general SDAHB). *Consider the inhomogeneous ODE in* (A.12). *Define the differential operators with* $\lambda \overset{def}{=} \sigma_j^2$

1. $L[\widehat{K}_s](t) = 0$, $t \geq s$, $\widehat{K}_s(t) = 0$, $t < s$

 where $\quad \widehat{K}_s(s) = \gamma_2^2$, $\quad \widehat{K}_s'(s) = 2\gamma_2\gamma_1 + 2\gamma_2^2(\theta - \gamma_2\lambda)$, $\quad$ and
 $\widehat{K}_s''(s) = 2\gamma_1(\gamma_1 + 3\gamma_2\theta - 4\gamma_2^2\lambda) + 2\gamma_2^2[2\theta^2 - 4\gamma_2\lambda\theta + 2\gamma_2^2\lambda^2]$.

2. $L[\widehat{J}_0(t)] = 0$

 where $\widehat{J}_0(0) = 1$, $\widehat{J}_0^{(1)}(0) = 2\theta - 2\gamma_2\lambda$, and $\widehat{J}_0^{(2)}(0) = (2\theta - 2\gamma_2\lambda)^2 - 2\gamma_1\lambda$.

*Then one has that*

$$K_s^{(\lambda)}(t) = \lambda^2 e^{2\theta(s-t)}\widehat{K}_s(t) \quad and \quad G^{(\lambda)}(t) = \frac{e^{-2\theta t}\widehat{J}_0(t)}{2}.$$

# B Relating Homogenized SGD to SGD on the random least squares problem

## B.1 Heuristic reduction

In this section, we give a nonrigorous derivation of the homogeneous sGD which holds in general. In the next section, we give a proof of Theorem 2 which applies in the case of $\gamma_1 = 0$ using the results from Paquette et al. [2021].

We are considering the SDA class of algorithms (1.1) which, for $\boldsymbol{x}_1 \in \mathbb{R}^d$ and $\boldsymbol{y}_0 = 0$,

$$\boldsymbol{y}_k = (1 - \Delta(k))\boldsymbol{y}_{k-1} + \frac{\gamma_1}{n}\nabla f_{i_k}(\boldsymbol{x}_k) \quad and \quad \boldsymbol{x}_{k+1} = \boldsymbol{x}_k - \gamma_2\nabla f_{i_k}(\boldsymbol{x}_k) - \boldsymbol{y}_k, \tag{B.1}$$

Here $\gamma_1, \gamma_2 > 0$ are step sizes and $\Delta$ is a function of the iteration $k$ and number of samples $n$ such that

$$\Delta(k) \overset{def}{=} \Delta(k,n) \overset{def}{=} \tfrac{1}{n}(\log\varphi)'(\tfrac{k}{n}).$$

Recall that our two motivating cases are SDANA for which $\Delta(k,n) = \frac{\theta}{k+n}$ and SDAHB for which $\Delta(k,n) = \frac{\theta}{n}$.

Recall that we consider the normalized least squares problem

$$f(\boldsymbol{x}) = \frac{1}{2}\|\boldsymbol{A}\boldsymbol{x} - \boldsymbol{b}\|^2,$$

and we use the singular value decomposition of $\boldsymbol{U}\boldsymbol{\Sigma}\boldsymbol{V}^T$ of $\boldsymbol{A}$, with singular values $\sigma_1 \geq \sigma_2 \geq \sigma_3\cdots$ in decreasing order. We then let $\boldsymbol{\nu}_t = \boldsymbol{V}^T(\boldsymbol{X}_t - \widetilde{\boldsymbol{x}})$. We recall that

$$\nabla f(\boldsymbol{X}_t) = \boldsymbol{A}^T(\boldsymbol{A}\boldsymbol{X}_t - \boldsymbol{b}) \quad and \quad \nabla^2 f = \boldsymbol{A}^T\boldsymbol{A}.$$

Hence, we may change the basis to write

$$\mathrm{d}(\boldsymbol{V}^T\boldsymbol{X}_t) = -\gamma_2\,\mathrm{d}(\boldsymbol{V}^T\boldsymbol{Z}_t) - \frac{\gamma_1}{\varphi(t)}\int_0^t \varphi(s)\,\mathrm{d}(\boldsymbol{V}^T\boldsymbol{Z}_t),$$

$$\mathrm{d}(\boldsymbol{V}^T\boldsymbol{Z}_t) = \boldsymbol{\Sigma}^T(\boldsymbol{\Sigma}\boldsymbol{\nu}_t - \boldsymbol{U}^T\boldsymbol{b})\,\mathrm{d}t + \sqrt{\tfrac{2}{n}f(\boldsymbol{X}_t)\boldsymbol{\Sigma}^T\boldsymbol{\Sigma}}\,\mathrm{d}(\boldsymbol{V}^T\boldsymbol{B}_t).$$

The loss values we may also represent in terms of $\boldsymbol{\nu}$

$$f(\boldsymbol{X}_t) = \frac{1}{2}\|\boldsymbol{A}\boldsymbol{X}_t - \boldsymbol{b}\|^2 = \frac{1}{2}\|\boldsymbol{\Sigma}\boldsymbol{\nu}_t - \boldsymbol{U}^T\boldsymbol{\eta}\|^2 = \frac{1}{2}\sum_{j=1}^n(\sigma_j\nu_{t,j} - (\boldsymbol{U}^T\boldsymbol{b})_j)^2.$$

We let $\widehat{\boldsymbol{w}}_k = \frac{n}{\gamma_1} \boldsymbol{V}^T \boldsymbol{y}_k$ and $\widehat{\boldsymbol{\nu}}_k = \boldsymbol{V}^T(\boldsymbol{x}_k - \widetilde{\boldsymbol{x}})$ (with $\boldsymbol{w}_0 = \boldsymbol{0}$ and $\widehat{\boldsymbol{\nu}}_1 \in \mathbb{R}^d$, $k \geq 1$) so that for a random rank-1 coordinate projection matrix $\boldsymbol{P}_k$

$$\widehat{\boldsymbol{w}}_k = \big(1 - \Delta(k, n)\big)\widehat{\boldsymbol{w}}_{k-1} + \boldsymbol{\Sigma}^T \boldsymbol{U}^T \boldsymbol{P}_k(\boldsymbol{U}\boldsymbol{\Sigma}\widehat{\boldsymbol{\nu}}_k - \boldsymbol{\eta}), \quad \text{and} \tag{B.2}$$

$$\widehat{\boldsymbol{\nu}}_{k+1} = \widehat{\boldsymbol{\nu}}_k - \gamma_2 \boldsymbol{\Sigma}^T \boldsymbol{U}^T \boldsymbol{P}_k(\boldsymbol{U}\boldsymbol{\Sigma}\widehat{\boldsymbol{\nu}}_k - \boldsymbol{\eta}) - \tfrac{\gamma_1}{n}\widehat{\boldsymbol{w}}_k. \tag{B.3}$$

By unraveling the recurrence for $\widehat{\boldsymbol{w}}$, a simple computation shows that

$$\widehat{\boldsymbol{w}}_k = \sum_{\ell=1}^{k} \prod_{i=\ell}^{k-1} \big[1 - \Delta(i, n)\big] \boldsymbol{\Sigma}^T \boldsymbol{U}^T \boldsymbol{P}_\ell(\boldsymbol{U}\boldsymbol{\Sigma}\widehat{\boldsymbol{\nu}}_\ell - \boldsymbol{\eta}). \tag{B.4}$$

We now create a continuous time version of the vector $\widehat{\boldsymbol{w}}_k$ so that $t$ and $s$ correspond to one pass over the data set. In doing so, we can approximate the product $\prod_{i=\ell}^{k-1} 1 - \Delta(i, n)$ by first taking logarithms and then approximating the sum with a Riemann integral. If we let $\ell = ns$ and $k = nt$,

$$\prod_{i=\ell}^{k-1} \big[1 - \Delta(i, n)\big] = \prod_{i=ns}^{nt} \big[1 - \Delta(i, n)\big] = \exp\left(\sum_{i=ns}^{nt} \log\big(1 - \Delta(i, n)\big)\right)$$

$$\approx \exp\left(-\sum_{i=ns}^{nt} \Delta(i, n)\right) = \exp\left(-\frac{1}{n}\sum_{i=ns}^{nt} (\log\varphi)'(\tfrac{i}{n})\right)$$

$$\approx \exp\left(-\int_s^t (\log\varphi)'(u) \; \mathrm{d}u\right) = \frac{\varphi(s)}{\varphi(t)}.$$

We are trying to isolate the martingale term in $\widehat{\boldsymbol{w}}_k$ so we need to find the mean behavior of $\widehat{\boldsymbol{w}}$. As such,

$$\widehat{w}_{k,j} = \sum_{\ell=1}^{k} \prod_{i=\ell}^{k-1} \big[1 - \Delta(i, n)\big]\big(\mathrm{e}_j^T \boldsymbol{\Sigma}^T \boldsymbol{U}^T \boldsymbol{P}_\ell(\boldsymbol{U}\boldsymbol{\Sigma}\boldsymbol{\nu}_\ell - \boldsymbol{\eta}) - \big(\tfrac{\sigma_j^2}{n}\widehat{\nu}_{\ell,j} - \tfrac{\sigma_j}{n}(\boldsymbol{U}^T\boldsymbol{\eta})_j\big)\big)$$

$$+ \sum_{\ell=1}^{k} \prod_{i=\ell}^{k-1} \big[1 - \Delta(i, n)\big]\big(\tfrac{\sigma_j^2}{n}\widehat{\nu}_{\ell,j} - \tfrac{\sigma_j}{n}(\boldsymbol{U}^T\boldsymbol{\eta})_j\big).$$

Define the martingale increment $\Delta\widehat{M}_{\ell,j} \stackrel{\text{def}}{=} \mathrm{e}_j^T \boldsymbol{\Sigma}^T \boldsymbol{U}^T \boldsymbol{P}_\ell(\boldsymbol{U}\boldsymbol{\Sigma}\boldsymbol{\nu}_\ell - \boldsymbol{\eta}) - \big(\tfrac{\sigma_j^2}{n}\widehat{\nu}_{\ell,j} - \tfrac{\sigma_j}{n}(\boldsymbol{U}^T\boldsymbol{\eta})_j\big)$. Then

$$\widehat{w}_{k,j} = \sum_{\ell=1}^{k} \prod_{i=\ell}^{k-1} \big[1 - \Delta(i, n)\big]\Delta\widehat{M}_{\ell,j} + \sum_{\ell=1}^{k} \prod_{i=\ell}^{k-1} \big[1 - \Delta(i, n)\big]\big(\tfrac{\sigma_j^2}{n}\widehat{\nu}_{\ell,j} - \tfrac{\sigma_j}{n}(\boldsymbol{U}^T\boldsymbol{\eta})_j\big)$$

We now pass to the continuous time by letting $k \sim nt$. So we define a continuous time, purely discontinuous martingale $M_t$ with jumps at times $\mathbb{N}/n$ which are given by

$$(\Delta M)_{\ell/n,j} \stackrel{\text{def}}{=} \mathrm{e}_j^T \boldsymbol{\Sigma}^T \boldsymbol{U}^T \boldsymbol{P}_\ell(\boldsymbol{U}\boldsymbol{\Sigma}\boldsymbol{\nu}_{\ell/n} - \boldsymbol{\eta}) - n^{-1}\big(\sigma_j^2 \nu_{\ell/n,j} - \sigma_j(\boldsymbol{U}^T\boldsymbol{\eta})_j\big)$$

In terms of this martingale, we define càdlàg processes $w_{t,j}$ and $\nu_{t,j}$ as approximations for $\widehat{\boldsymbol{w}}_{k,j}$ and $\widehat{\boldsymbol{\nu}}_{k,j}$. For $\boldsymbol{w}$ this is given by

$$w_{t,j} = \frac{1}{\varphi(t)}\int_0^t \varphi(s)\big(\mathrm{d}M_{s,j} + \big(\sigma_j^2 \nu_{s,j} - \sigma_j(\boldsymbol{U}^T\boldsymbol{\eta})_j\big) \, \mathrm{d}s\big).$$

As for $\boldsymbol{\nu}$, we must compute also compute the change in $\widehat{\nu}_{k,j}$:

$$\widehat{\nu}_{k+1,j} - \widehat{\nu}_{k,j} = -\mathrm{e}_j^T \gamma_2 \boldsymbol{\Sigma}^T \boldsymbol{U}^T \boldsymbol{P}_k(\boldsymbol{U}\boldsymbol{\Sigma}\widehat{\boldsymbol{\nu}}_k - \boldsymbol{\eta}) - \frac{\gamma_1}{n}\widehat{w}_{k,j}.$$

Again on scaling time to be like $k \sim nt$, we arrive at a continuous time stochastic evolution

$$\mathrm{d}\nu_t = -\gamma_2\big(\mathrm{d}M_{t,j} + \sigma_j^2 \nu_{t,j} - \sigma_j(\boldsymbol{U}^T\boldsymbol{\eta})_j\big) - \frac{\gamma_1}{\varphi(t)}\int_0^t \varphi(s)\big(\mathrm{d}M_{s,j} + \big(\sigma_j^2 \nu_{s,j} - \sigma_j(\boldsymbol{U}^T\boldsymbol{\eta})_j\big) \, \mathrm{d}s\big).$$

Thus this is exactly the homogenized SGD (A.2), but with the martingales $\{W_{t,j}\}$ replaced by $\{M_{t,j}\}$.

The martingales $\{M_{t,j}\}$ are purely discontinuous. Their predictable quadratic variations are given by (ignoring errors induced by smoothing the indexing)

$$\mathrm{d}\langle M_{t,j}, M_{t,i}\rangle = \sigma_j^2 \sum_{\ell=1}^{n} U_{\ell,j} U_{\ell,i} \big((\boldsymbol{U}\boldsymbol{\Sigma}\boldsymbol{\nu}_t - \boldsymbol{\eta})_\ell\big)^2 - n^{-1}\big(\sigma_j^2 \nu_{t,j} - \sigma_j(\boldsymbol{U}^T\boldsymbol{\eta})_j\big)\big(\sigma_i^2 \nu_{t,i} - \sigma_i(\boldsymbol{U}^T\boldsymbol{\eta})_i\big).$$

The latter term is too small to recover and so disappears in the large-$n$ limit. Note that in the first sum, if $(\boldsymbol{U}\boldsymbol{\Sigma}\boldsymbol{\nu}_t - \boldsymbol{\eta})_\ell$ could be decoupled from $U_{\ell,j}U_{\ell,i}$ and if $(U_{\ell,j}^2 : 1 \leq \ell \leq n)$ is sufficiently delocalized, then we would arrive at

$$\mathrm{d}\langle M_{t,j}, M_{t,i}\rangle \approx \delta_{i,j}\sigma_j^2 \sum_{\ell=1}^{n} U_{\ell,j}^2\big((\boldsymbol{U}\boldsymbol{\Sigma}\boldsymbol{\nu}_t - \boldsymbol{\eta})_\ell\big)^2 \approx \delta_{i,j}\frac{2\sigma_j^2}{n}f(\boldsymbol{X}_t),$$

from the fact that $U_{\ell,j}^2 \approx \frac{1}{n}$ on average. This is the homogenized SGD. The main input is sufficiently strong input information on the eigenvector matrix $\boldsymbol{U}$. In Paquette et al. [2021], it is assumed that this is independent of the spectra and Haar orthogonally distributed. We expect it remains true under weaker assumptions, but note some type of eigenvector assumption is needed. If for example $\boldsymbol{A}$ is diagonal, the resulting coordinate processes decouple entirely, as opposed to interacting through the loss values $\psi$.

## B.2    Proof of correspondence for SGD

We give a proof of Theorem 2, or rather show how Paquette et al. [2021] (which contains the argument) may be adapted to show the statement claimed. The starting point is an embedding of the discrete problem into continuous time. That is we create a homogeneous Poisson process $N_t$ with rate $n$ (so that in one unit of time, in expectation $n$ Poisson points arrive). We then introduce the notation

$$\psi_\varepsilon(t) \stackrel{\text{def}}{=} f(\boldsymbol{x}_{N_t}) = \frac{1}{2}\|\boldsymbol{A}(\boldsymbol{x}_{N_t} - \widetilde{\boldsymbol{x}}) - \boldsymbol{\eta}\|^2 = \frac{1}{2}\|\boldsymbol{\Sigma}\boldsymbol{\nu}_t - \boldsymbol{U}^T\boldsymbol{\eta}\|^2.$$

$$= \frac{1}{2}\sum_{j=1}^{d}\sigma_j^2\nu_{t,j}^2 - \sum_{j=1}^{n\wedge d}\sigma_j\nu_{t,j}(\boldsymbol{U}^T\boldsymbol{\eta})_j + \frac{1}{2}\|\boldsymbol{\eta}\|^2. \tag{B.5}$$

By partially integrating the equation, we can rewrite this equation (see the derivation [Paquette et al., 2021, Equation (41)] through [Paquette et al., 2021, Lemma 21] – note we are using batchsize $\beta = 1$).

$$\psi_\varepsilon(t) = \frac{1}{2}\sum_{j=1}^{n}\sigma_j^2\left(e^{-2t\gamma\sigma_j^2}\nu_{0,j}^2 + \int_0^t e^{-2(t-s)\gamma\sigma_j^2}\gamma^2\frac{2\sigma_j^2\psi_\varepsilon(s)}{n}\,\mathrm{d}s\right)$$

$$+ \frac{1}{2}\sum_{j=1}^{n}\int_0^t e^{-2(t-s)\gamma\sigma_j^2}\gamma 2\sigma_j^3\nu_{s,j}(\boldsymbol{U}^T\boldsymbol{\eta})_j\,\mathrm{d}s + \frac{1}{2}\|\boldsymbol{\eta}\|^2 - \sum_{j=1}^{n\wedge d}\sigma_j\nu_{t,j}(\boldsymbol{U}^T\boldsymbol{\eta})_j \tag{B.6}$$

$$+ \varepsilon_{\text{KL}}^{(n)}(t) + \varepsilon_{\text{M}}^{(n)}(t)$$

The two error terms $\varepsilon_{\text{KL}}^{(n)}(t)$ and $\varepsilon_{\text{M}}^{(n)}(t)$ are (first) due to the eigenvectors not being perfectly delocalized and (second) due to the randomness of SGD. Note that we can write this in terms of the notation for Theorem 4 by

$$\psi_\varepsilon(t) = F_{\boldsymbol{H}}(t) + \int_0^t \mathcal{K}_s(t)\psi_\varepsilon(s)\,\mathrm{d}s + \varepsilon_{\text{KL}}^{(n)}(t) + \varepsilon_{\text{M}}^{(n)}(t) + \varepsilon_{\text{xtra}}^{(n)}(t), \quad \text{where}$$

$$\varepsilon_{\text{xtra}}^{(n)}(t) = \frac{1}{2}\sum_{j=1}^{n}\sigma_j^2\left(e^{-2t\gamma\sigma_j^2}\big((\nu_{0,j} - (\boldsymbol{U}^T\boldsymbol{\eta})_j/\sigma_j)^2 - R/n - \widetilde{R}/(n\sigma_j^2)\big)\right)$$

$$+ \frac{1}{2}\sum_{j=1}^{n}\int_0^t e^{-2(t-s)\gamma\sigma_j^2}\gamma 2\sigma_j^3\nu_{s,j}(\boldsymbol{U}^T\boldsymbol{\eta})_j\,\mathrm{d}s \tag{B.7}$$

The main errors are controlled directly using the results of Paquette et al. [2021].

**Proposition B.1.** *There is an $\delta > 0$ so that for any $T > 0$ there is a constant $C(T, \lambda_{\boldsymbol{H}}^+) > 0$ so that*

$$\Pr[\sup_{0\leq t\leq T}\big\{|\varepsilon_{\text{KL}}^{(n)}(t)| + |\varepsilon_{\text{M}}^{(n)}(t)|\big\} \geq C(T)n^{-\delta} \mid \boldsymbol{\Sigma}] \leq n^{-\delta}.$$

*Proof.* In short this the combination of [Paquette et al., 2021, Lemma 13], [Paquette et al., 2021, Lemma 14], [Paquette et al., 2021, Proposition 16]. We note that the event that dominates the probability is the application of [Paquette et al., 2021, Lemma 13], which is simply Markov's inequality applied to the loss. □

The extra errors are controlled by [Paquette et al., 2021, Proposition 19] and by concentration of the initial conditions. Thus by (B.7), we have that the true loss of SGD satisfies an approximate Volterra equation. Using the stability of Volterra equations with respect to perturbation [Paquette et al., 2021, Proposition 11], we can conclude that $\psi_\epsilon$ is close to a solution of the Volterra equation with $\varepsilon = 0$, with the claimed probability.

## C   Analysis of convolution Volterra equations: convergence and rates

In what follows, we give an analysis of a class of convolution Volterra equations that appear naturally in the SDA context: our analysis will give convergence guarantees, convergence rates and limiting losses (in the underparameterized context. Ultimately, for all of SGD, SDAHB and SDANA, we will have the task of describing the evolution of the training loss $L(t) = \mathbb{E}_{\boldsymbol{H}}[f(X_t)]$ which satisfies

$$L(t) = F(t) + \int_0^t \mathcal{I}(t-s)L(s), \qquad\qquad \text{for all } t \geq 0,$$
$$F(t) = \int_0^\infty (R\lambda + \widetilde{R})G^{(\lambda)}(t)\mu(d\lambda), \quad \text{and} \quad \mathcal{I}(t) = \int_0^\infty K^{(\lambda)}(t)\mu(d\lambda), \quad \text{for all } t \geq 0.$$
(C.1)

We refer to $F$ as the forcing function and $\mathcal{I}$ as the convolution kernel. The measure $\mu$ is the limiting spectral measure of the Hessian problem (some parts of the analysis also hold with $\mu$ the actual empirical spectral measure of the problem). In all cases, operate under the following assumptions.

**Assumption 3.** *The functions $(\lambda, t) \mapsto G^{(\lambda)}(t)$ and $(\lambda, t) \mapsto K^{(\lambda)}(t)$ are non-negative, continuous and bounded on bounded sets of $\lambda$. Assume further that for each $\lambda > 0$, $K$ and $G$ tend to $0$ as $t \to \infty$. At $\lambda = 0$, on the other hand $K^{(\lambda)} \equiv 0$ and $G^{(\lambda)} \equiv 1$.*

In this section, we shall also do the analysis for SGD. Much of SGD analysis appeared already in Paquette et al. [2021], but it serves as an instructive and simple example. Recall that for the case of SGD, we have that

$$G_{\text{sgd}}^{(\lambda)}(t) = e^{-2\gamma\lambda t} \quad \text{and} \quad K_{\text{sgd}}^{(\lambda)}(t) = \gamma^2\lambda^2 e^{-2\gamma\lambda t}$$
(C.2)

The actual convergence analysis of $L$ in this setup is relatively simple. As a consequence of dominated convergence, we have $F$ converges as $t \to \infty$, and in fact

$$\lim_{t \to \infty} F(t) = \mu(\{0\})\widetilde{R}.$$

The important input to ensure convergence is that the norm of the convolution kernel is controlled.

$$\|\mathcal{I}\| = \int_0^\infty \mathcal{I}(t)dt = \int_0^\infty \int_0^\infty K^{(\lambda)}(t)dt\mu(d\lambda)$$
(C.3)

Thus by dominated convergence, we have the following:

**Lemma C.1.** *Suppose Assumption 3 and suppose $\|\mathcal{I}\| < 1$. Then*

$$\lim_{t \to \infty} L(t) = \frac{\mu(\{0\})\widetilde{R}}{1 - \|\mathcal{I}\|}.$$

For SGD, in particular that means (using (C.3))

$$\|\mathcal{I}_{\text{sgd}}\| = \int_0^\infty \int_0^\infty K_{\text{sgd}}^{(\lambda)}(t)dt\mu(d\lambda) = \int_0^\infty \frac{\gamma\lambda}{2}\mu(d\lambda) = \frac{\gamma\text{tr}(\mu)}{2},$$
(C.4)

where $\text{tr}(\mu)$ is the limiting normalized trace of the Hessian, i.e. the first moment of the measure $\mu$.

**Rates (heavy-tailed case)** The rate analysis is divided into two cases, according to the behavior of the forcing function $F$. If $F$ converges exponentially quickly to its limit (which occurs in our applications when the spectrum of $\mu$ is separated from 0), then the forcing function converges exponentially. On the other hand, if $\mu$ has a density in a neighborhood of 0, then the rate is subexponential, and we will suppose further that $F$ and $\mathcal{I}$ are both tending slowly to 0.

**Assumption 4.** *The function $F$ dominates $\mathcal{I}$ and $\mathcal{I}/\|\mathcal{I}\|$ defines a subexponential distribution: that is*

$$F(t)/\mathcal{I}(t) \to \infty \quad and \quad \frac{\int_T^\infty \int_0^t \mathcal{I}(t-s)\mathcal{I}(s)dsdt}{\int_T^\infty \mathcal{I}(s)ds} \to 2\|\mathcal{I}\|.$$

A simple sufficient condition for the latter of the two conditions is that $\mathcal{I}(t)t^\beta \to c$ for some $c > 0, \beta > 0$ as $t \to \infty$. In this case, the rate of convergence of $L$ to its limit is

**Lemma C.2.** *Suppose Assumptions 3 and 4 and suppose $\|\mathcal{I}\| < 1$. Then*

$$L(t) - \frac{\mu(\{0\})\widetilde{R}}{1-\|\mathcal{I}\|} \sim_{t\to\infty} \frac{F(t) - \mu(\{0\})\widetilde{R}}{1-\|\mathcal{I}\|}.$$

In other words, the rate is completely dominated by whichever rate $F$ takes.

*Proof.* If we subtract the limiting behavior from $L$, we have

$$L(t) - \frac{\mu(\{0\})\widetilde{R}}{1-\|\mathcal{I}\|} = F(t) - \mu(\{0\})\widetilde{R} + \int_0^t \mathcal{I}(t-s)\left(L(s) - \frac{\mu(\{0\})\widetilde{R}}{1-\|\mathcal{I}\|}\right)ds$$

Thus if we set

$$\widehat{L} = L - \frac{\mu(\{0\})\widetilde{R}}{1-\|\mathcal{I}\|} \quad and \quad \widehat{F} = F - \mu(\{0\})\widetilde{R},$$

we conclude that

$$\widehat{L}(t) = \widehat{F}(t) + \int_0^t \mathcal{I}(t-s)\widehat{L}(s)ds.$$

This is a defective renewal equation, and moreover it is a defective renewal equation in which $\mathcal{I}/\|\mathcal{I}\|$ defines a *subexponential distribution*. Thus from [Asmussen, 2003, (7.8)], we conclude the claim. $\square$

To apply this to SGD, we suppose that

$$\mu((0,\epsilon]) \underset{\epsilon\to 0}{\sim} \ell\epsilon^\alpha. \tag{C.5}$$

In this case, we conclude that

$$\mathcal{I}_{\text{sgd}}(t) \underset{t\to\infty}{\sim} \frac{\Gamma(2+\alpha)\alpha\ell\gamma^2}{(2\gamma t)^{2+\alpha}} \quad and \quad F_{\text{sgd}}(t) - \mu(\{0\})\widetilde{R} \underset{t\to\infty}{\sim} \frac{R\Gamma(1+\alpha)\alpha\ell}{(2\gamma t)^{1+\alpha}} + \frac{\widetilde{R}\Gamma(\alpha)\alpha\ell}{(2\gamma t)^\alpha}.$$

Thus we conclude the rate for the loss of SGD (using (C.2) and Lemma C.2

$$\mathbb{E}_{\boldsymbol{H}}[f(X_t)] - \frac{\mu(\{0\})\widetilde{R}}{1 - \frac{\gamma\text{tr}(\mu)}{2}} \underset{t\to\infty}{\sim} \frac{1}{1 - \frac{\gamma\text{tr}(\mu)}{2}}\left(\frac{R\Gamma(1+\alpha)\alpha\ell}{(2\gamma t)^{1+\alpha}} + \frac{\widetilde{R}\Gamma(\alpha)\alpha\ell}{(2\gamma t)^\alpha}\right). \tag{C.6}$$

**Rates (exponential case)** We now consider the case $F$ and $\mathcal{I}$ tend to 0 exponentially, as is the case when $\mu$ has support $\{0\} \cup [\lambda_-, \lambda_+]$ for positive $\lambda_-$. We enforce these assumptions by assuming that both $G^{(\lambda)}$ and $K^{(\lambda)}$ behave well in a neigbhorhood of the spectral edge.

**Assumption 5.** *The support of $\mu$ is contained in $\{0\} \cup [\lambda_-, \lambda_+]$ for some $\lambda_- > 0$, and $\lambda_-$ is in the support. The kernels satisfy for some positive strictly increasing function $f$ on some $[\lambda_-, \lambda_- + \delta]$*

$$G^{(\lambda)}(t) = e^{-(f(\lambda)+o(1))t} \quad and \quad K^{(\lambda)}(t) = e^{-(f(\lambda)+o(1))t}$$

*as $t \to \infty$. Both $G^{(\lambda)}(t)$ and $K^{(\lambda)}(t)$ are bounded by $e^{-(f(\lambda)+o(1))t}$ for larger $\lambda$ as $t \to \infty$.*

To estimate the rate, we need to introduce the Laplace transform of the kernel

$$\mathcal{F}(x) \stackrel{\text{def}}{=} \int_0^\infty e^{xt}\mathcal{I}(t)dt = \int_0^\infty \int_0^\infty e^{xt}K^{(\lambda)}(t)dt\mu(dx) \tag{C.7}$$

We define the Malthusian exponent $\lambda^*$, if it exists, as the solution of

$$\mathcal{F}(\lambda^*) = 1. \tag{C.8}$$

The Malthusian exponent gives the right behavior, on exponential scale for the rate of convergence, when it exists. Otherwise it is simply $e^{-f(\lambda_-)t}$:

**Lemma C.3.** *Suppose Assumptions 5 and 3 and suppose $\|\mathcal{I}\| < 1$. Then*

$$L(t) - \frac{\mu(\{0\})\widetilde{R}}{1 - \|\mathcal{I}\|} = \begin{cases} e^{-(\lambda^* + o(1))t}, & \text{if } \lambda^* \text{ exists}, \\ e^{-(f(\lambda_-) + o(1))t}, & \text{otherwise}, \end{cases}$$

*as $t \to \infty$. Furthermore $\lambda^* \le f(\lambda_-)$ if it exists.*

*Proof.* First, if the Malthusian exponent exists, we observe it must be positive, since by assumption

$$\mathcal{F}(0) = \int_0^\infty \mathcal{I}(t)dt = \|\mathcal{I}\| < 1,$$

and the function $x \mapsto \mathcal{F}(x)$ is increasing (and continuously differentiable on $[0, \lambda_-)$). Furthermore, the Laplace transform $\mathcal{F}(x) = \infty$ for any $x > f(\lambda_-)$ as $\lambda_-$ is in the support of $\mu$ and $f$ is increasing. Hence $\lambda^*$ if it exists is in $(0, f(\lambda_-)]$.

It follows that if the Malthusian exponent does not exist, then $\mathcal{F}(f(\lambda_-)) < 1$. In that case we can transform (C.1) by taking

$$\widehat{L}(t) = e^{f(\lambda_-)t}\left(L(t) - \frac{\mu(\{0\})\widetilde{R}}{1 - \|\mathcal{I}\|}\right) \quad \text{and} \quad \widehat{L}(t) = e^{f(\lambda_-)t}\left(F(t) - \mu(\{0\})\widetilde{R}\right),$$

which therefore satisfies

$$\widehat{L}(t) = \widehat{F}(t) + \int_0^t e^{f(\lambda_-)s}\mathcal{I}(s)\widehat{L}(t-s)ds.$$

The function $\widehat{F}(t)$ grows subexponentially in $t$ by hypothesis, and since $e^{f(\lambda_-)s}\mathcal{I}(s)$ has norm less than 1 (its norm being $\mathcal{F}(f(\lambda_-))$) it follows that

$$\widehat{L}(t) = \widehat{F}(t) + \int_0^t R(s)\widehat{F}(t-s)ds,$$

for the L$^1$-resolvent kernel of $e^{f(\lambda_-)s}\mathcal{I}(s)$, which is the infinite series of convolution powers of this kernel. Hence $\widehat{L}(t)$ grows at most subexponentially and at least as fast as $\widehat{F}$, from which we conclude that $L(t)$ behaves like $e^{-(f(\lambda_-) + o(1))t}$.

If $\mathcal{F}(f(\lambda_-)) \ge 1$, we instead have a nontrivial Malthusian exponent inside of $(0, f(\lambda_-)]$. We therefore have after making the transformation

$$\widehat{L}(t) = e^{\lambda^* t}\left(L(t) - \frac{\mu(\{0\})\widetilde{R}}{1 - \|\mathcal{I}\|}\right) \quad \text{and} \quad \widehat{L}(t) = e^{\lambda^* t}\left(F(t) - \mu(\{0\})\widetilde{R}\right),$$

that $\widehat{L}(t)$ solves Blackwell's renewal equation (see [Asmussen, 2003, Theorem 4.7]. If $\mathcal{F}(f(\lambda_-)) > 1$, then the Laplace transform $\mathcal{F}$ is differentiable at $\lambda^*$, and so the renewals have finite mean $\mu = \mathcal{F}(\lambda^*)'$. In particular it follows that $\widehat{L}(t)$ actually converges to $\frac{1}{\mu}\int_0^\infty \widehat{F}(t)dt$, which is finite by the exponential growth condition.

In the critical case $\mathcal{F}(f(\lambda_-)) = 1$, we observe that

$$\max_{[0,t]} \widehat{L}(u) \le \max_{[0,t]} \widehat{F}(u) + \left(\max_{[0,t]} \widehat{L}(u)\right)\int_0^t e^{f(\lambda_-)s}\mathcal{I}(s)ds,$$

thus rearranging, and using that $\int_0^t e^{f(\lambda_-)s}\mathcal{I}(s)ds \to 1$ as $t \to \infty$ we conclude

$$\max_{[0,t]} \widehat{L}(u) \le \frac{\left(\max_{[0,t]} \widehat{F}(u)\right)}{\int_t^\infty e^{f(\lambda_-)s}\mathcal{I}(s)ds},$$

which therefore grows subexponentially. $\qquad\square$

For SGD, the $K$ and $G$ (recall (C.2)) satisfy Assumption 5 trivially with $f(\lambda) = 2\gamma\lambda$. Hence, to esetimate the rate, by which we mean

$$\text{sgd-rate}(\gamma) \stackrel{\text{def}}{=} \liminf_{t\to\infty} \frac{-\log\big(\mathbb{E}_{\boldsymbol{H}}[f(\boldsymbol{X}_t)] - (\mathbb{E}_{\boldsymbol{H}}[f(\boldsymbol{X}_\infty)])\big)}{t},$$

the only task is to estimate the Malthusian exponent.

**Lemma C.4.** *At the default parameter for SGD, $\gamma = \frac{1}{\text{tr}(\mu)}$, the rate is at least $\frac{\lambda_-}{\text{tr}(\mu)}$. The maximum rate over all $\gamma$ is at most $4\frac{\lambda_-}{\text{tr}(\mu)}$.*

*Proof.* Note that the Laplace transform is given by

$$\mathcal{F}_{\text{sgd}}(x) = \int_0^\infty e^{xt}\mathcal{I}_{\text{sgd}}(t)dt = \int_0^\infty \frac{\gamma^2\lambda^2}{2\gamma\lambda - x}\mu(dx).$$

Note that if we choose the default parameter $\gamma = \frac{1}{\text{tr}(\mu)}$, then at $x = \frac{\lambda_-}{\text{tr}(\mu)}$ we have

$$\mathcal{F}_{\text{sgd}}\big(\tfrac{\lambda_-}{\text{tr}(\mu)}\big) = \frac{1}{\text{tr}(\mu)}\int_0^\infty \frac{\lambda^2}{2\lambda - \lambda_-}\mu(dx) \leq \frac{1}{\text{tr}(\mu)}\int_0^\infty \frac{\lambda^2}{\lambda}\mu(dx) \leq 1.$$

It follows that we have shown that the rate at the default parameter is at least $\frac{\lambda_-}{\text{tr}(\mu)}$.

To get an upper bound over all step sizes, note that from (C.4), the largest $\gamma$ we can take is determined by

$$\frac{\gamma\text{tr}(\mu)}{2} = \|\mathcal{I}_{\text{sgd}}\| < 1.$$

Further, the fastest rate we can ever attain is $e^{-2\gamma\lambda_- t}$, and hence taking the largest $\gamma$, the fastest possible rate is $e^{-\frac{4\lambda_-}{\text{tr}(\mu)}t}$. $\qquad\qquad\square$

# D  Momentum can be faster, SDANA

In this section, we consider the SDANA case in depth, developing approximations and limit behaviors for the differential equations for which one achieves acceleration in the non-strongly convex setting. Recall from the ODE for $\widehat{J}$ in (A.8) and the initial conditions (A.9),

$$\widehat{J}^{(3)} - \Big(\frac{3\theta}{(1+t)} - 3\gamma_2\sigma_j^2\Big)\widehat{J}^{(2)} - \Big(-\frac{5\theta+2\theta^2}{(1+t)^2} + \frac{4\gamma_2\sigma_j^2\theta}{(1+t)} - 4\gamma_1\sigma_j^2 - 2\gamma_2^2\sigma_j^4\Big)\widehat{J}^{(1)}$$
$$- \Big(\frac{4\theta+4\theta^2}{(1+t)^3} - \frac{4\gamma_2\sigma_j^2\theta}{(1+t)^2} + \frac{4\gamma_1\sigma_j^2\theta}{(1+t)} - 4\gamma_1\gamma_2\sigma_j^4\Big)\widehat{J} \tag{D.1}$$
$$= \frac{\gamma_2^2}{\gamma_1}\widehat{\psi}^{(2)} + \Big(2\gamma_2 + \big(\frac{-\theta}{(1+t)} + \gamma_2\sigma_j^2\big)\frac{\gamma_2^2}{\gamma_1}\Big)\widehat{\psi}^{(1)} + \Big(2\gamma_1 + \big(\frac{\theta}{(1+t)^2} + 2\gamma_1\sigma_j^2\big)\frac{\gamma_2^2}{\gamma_1}\Big)\widehat{\psi},$$

where the initial conditions are given by

$$\widehat{J}(0) = \gamma_1^{-1}\mathbb{E}\left[\Big(\nu_{0,j} - \frac{(\boldsymbol{U}^T\boldsymbol{\eta})_j}{\sigma_j}\Big)^2\right], \quad \widehat{J}^{(1)}(0) = \frac{\gamma_2^2}{\gamma_1}\widehat{\psi}(0) - \widehat{J}(0)\big(-2\theta + 2\gamma_2\sigma_j^2\big), \quad \text{and}$$
$$\widehat{J}^{(2)}(0) = \frac{\gamma_2^2}{\gamma_1}\widehat{\psi}^{(1)}(0) + 2\gamma_2\widehat{\psi}(0) - 2\gamma_1\sigma_j^2\widehat{J}(0) + (2\theta - 2\gamma_2\sigma_j^2)\widehat{J}^{(1)}(0) - 2\theta\widehat{J}(0). \tag{D.2}$$

**Remark 1.** *It is possible to represent the solutions to the homogeneous ODE (D.1) by*

$$\widehat{J}(t) = c_1 \underbrace{(1+t)^\theta e^{-\gamma_2\sigma_j^2 t}\big(\text{WhittakerM}(A,B,C)\big)^2}_{y_1} + c_2 \underbrace{(1+t)^\theta e^{-\gamma_2\sigma_j^2 t}\big(\text{WhittakerW}(A,B,C)\big)^2}_{y_2}$$
$$+ c_3 \underbrace{(1+t)^\theta e^{-\gamma_2\sigma_j^2 t}\big(\text{WhittakerM}(A,B,C)\cdot\text{WhittakerW}(A,B,C)\big)}_{y_3}$$

$$where \quad A = \frac{\sigma_j\gamma_2\theta}{2\sqrt{\sigma_j^2\gamma_2^2 - 4\gamma_1}}, \quad B = \frac{\theta-1}{2}, \quad and \quad C = \sigma_j(1+t)\sqrt{\sigma_j^2\gamma_2^2 - 4\gamma_1}.$$

$$\tag{D.3}$$

*For multiple reasons, working with this representation appears to add complications: we need uniform asymptotic expansions as $\sigma$ tends to 0. We also need estimates for the fundamental solutions with parameters in a neighborhood of the turning point $\sigma_j^2\gamma_2^2 - 4\gamma_1 \approx 0$.*

## D.1 The fundamental solutions of the scaled ODE

To give uniform estimates as $\sigma_j \to 0$, we will scale time $t$ by $\sigma_j$ and in doing so, we define a scaled differential equation for $\widetilde{J}(t) = \widehat{J}(t/\sigma_j)$. We develop properties of the fundamental solutions of the homogeneous version of the equation (D.1), given by

$$
\widetilde{L}[\widetilde{J}] := \widetilde{J}^{(3)} - \left( \frac{3\theta}{\sigma_j + t} - 3\gamma_2 \sigma_j \right) \widetilde{J}^{(2)} - \left( -\frac{5\theta + 2\theta^2}{(\sigma_j + t)^2} + \frac{4\gamma_2 \sigma_j \theta}{\sigma_j + t} - 4\gamma_1 - 2\gamma_2^2 \sigma_j^2 \right) \widetilde{J}'
$$
$$
- \left( \frac{4\theta + 4\theta^2}{(\sigma_j + t)^3} - \frac{4\gamma_2 \sigma_j \theta}{(\sigma_j + t)^2} + \frac{4\gamma_1 \theta}{\sigma_j + t} - 4\gamma_1 \gamma_2 \sigma_j \right) \widetilde{J} = 0.
\tag{D.4}
$$

One can, in principle, derive an exact solution for this ODE using Whittaker functions; the resulting solution is quite cumbersome. As such we develop families of local solutions in a neighborhood of 0 and in neighborhood of $\infty$.

The Wronskian of this differential equation will be needed multiple times. Due to Abel's identity, the Wronskian of any three fundamental solutions of (D.4) is (for any $t, s \in \mathbb{R}$)

$$
\frac{\mathscr{W}(t)}{\mathscr{W}(s)} = \frac{(\sigma_j + t)^{3\theta}}{(\sigma_j + s)^{3\theta}} e^{-3\gamma_2 \sigma_j (t-s)}.
\tag{D.5}
$$

**The neighborhood of infinity.** The approach we take is to derive a local series solution for large $t$ as seen in [Coddington and Levinson, 1955, Chapter 5]. We observe that the coefficients in the linear ODE (D.4) are analytic in a neighborhood of $\infty$. As such, there exists a *formal* solution to this ODE [Coddington and Levinson, 1955, Chapter 5, Theorem 2.1] given by

$$
\widetilde{J}(t) = e^{\lambda t}(\sigma_j + t)^\rho P(t) = e^{\lambda t}(\sigma_j + t)^\rho \left( c_0 + \frac{c_1}{\sigma_j + 1} + \frac{c_2}{(\sigma_j + t)^2} + \cdots \right),
$$

where $\lambda, \rho$ are constants and $P(t)$ is an analytic function in a neighborhood of $\infty$. This formal series solution asymptotically agrees with the actual solution [Coddington and Levinson, 1955, Chapter 5, Theorem 4.1], and in fact are convergent solutions for all $t \in (0, \infty)$. We now derive the constants $\lambda$ and $\rho$ by simply plugging in our guess for the solution and deriving equations for $\lambda$ and $\rho$. To make this computationally tractable, we will compute derivatives in terms of $\widetilde{J}'/\widetilde{J}$, that is,

$$
\frac{\widetilde{J}'}{\widetilde{J}} = \lambda + \frac{\rho}{\sigma_j + t} - \frac{c_1}{(\sigma_j + t)^2} + \mathcal{O}(t^{-3}), \quad \left( \frac{\widetilde{J}'}{\widetilde{J}} \right)' = \frac{\widetilde{J}''}{\widetilde{J}} - \left( \frac{\widetilde{J}'}{\widetilde{J}} \right)^2 = -\frac{\rho}{(\sigma_j + t)^2} + \mathcal{O}(t^{-3})
$$
$$
\text{and} \quad \left( \frac{\widetilde{J}'}{\widetilde{J}} \right)'' = \frac{\widetilde{J}'''}{\widetilde{J}} - 3\frac{\widetilde{J}''\widetilde{J}'}{\widetilde{J}^2} + 2\left( \frac{\widetilde{J}'}{\widetilde{J}} \right)^3 = \mathcal{O}(t^{-3}).
$$

In particular, after some simple computations, we get the following expressions

$$
\frac{\widetilde{J}'}{\widetilde{J}} = \lambda + \frac{\rho}{\sigma_j + t} - \frac{c_1}{(\sigma_j + t)^2} + \mathcal{O}(t^{-3})
$$
$$
\frac{\widetilde{J}''}{\widetilde{J}} = \lambda^2 + \frac{2\lambda\rho}{\sigma_j + t} + \frac{\rho^2 - 2\lambda c_1 - \rho}{(\sigma_j + t)^2} + \mathcal{O}(t^{-3})
$$
$$
\frac{\widetilde{J}'''}{\widetilde{J}} = \lambda^3 + \frac{3\lambda^2 \rho}{\sigma_j + t} + \frac{3\lambda\rho^2 - 3\lambda^2 c_1 - 3\lambda\rho}{(\sigma_j + t)^2} + \mathcal{O}(t^{-3}).
$$

Finally we have all the pieces to get the expressions for the coefficients $\lambda$ and $\rho$ by using (D.4)

$$
0 = \lambda^3 + 3\gamma_2 \sigma_j \lambda^2 + (4\gamma_1 + 2\gamma_2^2 \sigma_j^2)\lambda + 4\gamma_1 \gamma_2 \sigma_j = (\lambda + \sigma_j \gamma_2)(\lambda^2 + 2\sigma_j \gamma_2 \lambda + 4\gamma_1)
$$
$$
\text{and} \quad 0 = (\sigma + t)^{-1}\left[ 3\theta\lambda^2 + 4\gamma_2 \sigma_j^2 \theta\lambda + 4\gamma_1 \sigma_j^2 \theta - \rho(3\lambda^2 + 6\gamma_2 \sigma_j^2 \lambda + 4\gamma_1 \sigma_j^2 + 2\gamma_2^2 \sigma_j^4) \right].
\tag{D.6}
$$

From solving the cubic equation, we get that $\lambda = -\sigma_j \gamma_2, -\sigma_j \gamma_2 \pm \sqrt{\sigma_j^2 \gamma_2^2 - 4\gamma_1}$. For each $\lambda$, we determine the corresponding $\rho$, that is,

$$
\lambda = -\sigma_j \gamma_2 \quad \Rightarrow \quad \rho = \theta
$$
$$
\lambda = -\sigma_j \gamma_2 \pm \sqrt{\sigma_j^2 \gamma_2^2 - 4\gamma_1} \quad \Rightarrow \quad \rho = \theta\left( 1 \mp \frac{\sigma_j \gamma_2}{\sqrt{\sigma_j^2 \gamma_2^2 - 4\gamma_1}} \right).
$$

As a result, the three fundamental solutions are

$$j_1(t) = e^{-\sigma_j\gamma_2 t}(\sigma_j + t)^\theta \left(1 + \frac{c_1}{\sigma_j+t} + \mathcal{O}(t^{-2})\right)$$

$$j_2(t) = e^{-\sigma_j\gamma_2 t}(\sigma_j + t)^\theta \exp\left(\sqrt{\sigma_j^2\gamma_2^2 - 4\gamma_1}\left[t - \log(\sigma_j + t)\frac{\theta\sigma_j\gamma_2}{\sigma_j^2\gamma_2^2 - 4\gamma_1}\right]\right)\left(1 + \frac{c_1}{\sigma_j+t} + \mathcal{O}(t^{-2})\right)$$

$$j_3(t) = e^{-\sigma_j\gamma_2 t}(\sigma_j + t)^\theta \exp\left(-\sqrt{\sigma_j^2\gamma_2^2 - 4\gamma_1}\left[t - \log(\sigma_j + t)\frac{\theta\sigma_j\gamma_2}{\sigma_j^2\gamma_2^2 - 4\gamma_1}\right]\right)\left(1 + \frac{c_1}{\sigma_j+t} + \mathcal{O}(t^{-2})\right).$$

$$\text{(D.7)}$$

**The neighborhood of zero.** We may follow the same approach in a neighborhood of $\sigma_j + t = 0$, where (D.4) has a regular singular point. The solutions are now controlled by the *indicial equation* of the differential equation, which is given by

$$\mathfrak{I}(\lambda) := \lambda(\lambda - 1)(\lambda - 2) - 3\theta\lambda(\lambda - 1) + (5\theta + 2\theta^2)\lambda - (4\theta + 4\theta^2) = 0. \quad \text{(D.8)}$$

This is polynomial is explicitly factorizable by

$$\mathfrak{I}(\lambda) = (\lambda - 2\theta)(\lambda - (1 + \theta))(\lambda - 2).$$

Hence when $\theta$ is not an integer, there are three fundamental solutions

$$j_1(t) = (\sigma_j + t)^{2\theta}\left(1 + \mathfrak{a}_{11}(\sigma_j + t) + \mathfrak{a}_{21}(\sigma_j + t)^2 + \cdots\right) =: (\sigma_j + t)^{2\theta}\mathfrak{a}_1(t),$$

$$j_2(t) = (\sigma_j + t)^{1+\theta}\left(1 + \mathfrak{a}_{12}(\sigma_j + t) + \mathfrak{a}_{22}(\sigma_j + t)^2 + \cdots\right) =: (\sigma_j + t)^{1+\theta}\mathfrak{a}_2(t), \quad \text{(D.9)}$$

$$j_3(t) = (\sigma_j + t)^2\left(1 + \mathfrak{a}_{13}(\sigma_j + t) + \mathfrak{a}_{23}(\sigma_j + t)^2 + \cdots\right) =: (\sigma_j + t)^2\mathfrak{a}_3(t).$$

The coefficients of these recurrences are defined by a recurrence. For the case of $\mathfrak{a}_1$, this recurrence is given by

$$\mathfrak{a}_{j1}\mathfrak{I}(2\theta + j) = \mathfrak{a}_{(j-1)1}\left\{-3\gamma_2\sigma_j(2\theta + j - 1)(2\theta + j - 2)\right\} + \mathcal{O}(j),$$

where we take coefficients $\mathfrak{a}_{k1} = 0$ for $k$ negative. The error term also depends on previous coefficients $\{\mathfrak{a}_{(j-1)1}, \mathfrak{a}_{(j-2)1}, \mathfrak{a}_{(j-3)1}\}$, and the other coefficients in (D.4). In particular, we may bound this recurrence by

$$|\mathfrak{a}_{j1}| \leq \left(\max_{0 \leq k \leq j-1} |\mathfrak{a}_{k1}|\right)\frac{3\gamma_2\sigma_j(2\theta + j - 1)(2\theta + j - 2) + M(\gamma_1, \gamma_2, \sigma_j, \theta)(2\theta + j - 2)}{\mathfrak{I}(2\theta + j)},$$

where the function $M$ is a continuous function of its parameters on all $\mathbb{R}^4$. By induction, we conclude that

$$|\mathfrak{a}_{j1}| \leq \frac{\Gamma(2\theta + 1)}{\Gamma(2\theta + 1 + j)}(3\gamma_2\sigma_j)^j(2\theta + j)^{M(\gamma_1, \gamma_2, \sigma_j, \theta)}.$$

Applying this argument to the other sequences, we conclude that:

**Lemma D.1.** *There is a continuous function $M := M(\gamma_1, \gamma_2, \sigma_j, \theta) \geq 0$ on $\mathbb{R}^4$ so that*

$$\left\|\begin{bmatrix} \mathfrak{a}_1(t) & \mathfrak{a}_2(t) & \mathfrak{a}_3(t) \\ \mathfrak{a}_1'(t) & \mathfrak{a}_2'(t) & \mathfrak{a}_3'(t) \\ \mathfrak{a}_1''(t) & \mathfrak{a}_2''(t) & \mathfrak{a}_3''(t) \end{bmatrix}\right\| \leq (2 + \sigma_j + t)^M e^{3\gamma_2\sigma_j t} \quad \textit{for all} \quad t \geq 0.$$

We will also need some estimates on the fundamental matrix built from these solutions. Define

$$\mathfrak{P}(t) = \begin{bmatrix} j_1(t) & j_2(t) & j_3(t) \\ j_1'(t) & j_2'(t) & j_3'(t) \\ j_1''(t) & j_2''(t) & j_3''(t) \end{bmatrix} \underset{t \to -\sigma_j}{\sim} \begin{bmatrix} (t + \sigma_j)^{2\theta} & (t + \sigma_j)^{1+\theta} & (t + \sigma_j)^2 \\ 2\theta(t + \sigma_j)^{2\theta-1} & (1 + \theta)(t + \sigma_j)^\theta & 2(t + \sigma_j)^1 \\ (2\theta)(2\theta + 1)(t + \sigma_j)^{2\theta-2} & (1 + \theta)\theta(t + \sigma_j)^{\theta-1} & 2 \end{bmatrix}.$$

In particular the Wronskian of $\mathfrak{P}$ satisfies

$$\det\mathfrak{P}(t) \underset{t \to -\sigma_j}{\sim} (t + \sigma_j)^{3\theta}\det\begin{bmatrix} 1 & 1 & 1 \\ 2\theta & 1 + \theta & 2 \\ (2\theta)(2\theta + 1) & (1 + \theta)\theta & 2 \end{bmatrix} =: p_\theta(t + \sigma_j)^{3\theta}.$$

We conclude from (D.5) that for any $t, \epsilon \geq -\sigma_j$,

$$\det\mathfrak{P}(t) = \det\mathfrak{P}(\epsilon)\frac{(t + \sigma_j)^{3\theta}}{(\epsilon + \sigma_j)^{3\theta}}e^{-3\gamma_2\sigma_j(t-\epsilon)} \underset{\epsilon \to 0}{\longrightarrow} (t + \sigma_j)^{3\theta}e^{-3\gamma_2\sigma_j(t+\sigma_j)}. \quad \text{(D.10)}$$

We conclude that:

**Lemma D.2.** *There is a continuous function $M := M(\gamma_1, \gamma_2, \sigma_j, \theta) \geq 0$ on $\mathbb{R}^4$ so that for all $t > -\sigma_j$*

$$\|\mathfrak{P}(t)\| \leq (2 + \sigma_j + t)^{M+2\theta} e^{3\gamma_2 \sigma_j t} \quad \text{and} \quad \|\mathfrak{P}^{-1}(t)\| \leq M \frac{(2 + \sigma_j + t)^{M+2\theta} e^{9\gamma_2 \sigma_j t}}{(\sigma_j + t)^{2\theta}}.$$

*Proof.* The first bound follows directly from Lemma D.1. The second bound follows from Cramér's rule. We note that some entries of the inverse matrix are singular at 0, but all have the form of

$$\frac{j_a^{(i-1)} j_b^{(j-1)} - j_b^{(i-1)} j_a^{(j-1)}}{\det \mathfrak{P}(t)},$$

for some $a, b, i, j \in \{1, 2, 3\}$. Hence the smallest positive power of $(t + \sigma_j)$ is achieved by taking all $a, b, i, j \in \{2, 3\}$. $\qquad\square$

**Improved bounds at the singular point.** When $\gamma_2^2 \sigma_j^2 - 4\gamma_1 \approx 0$, the solutions constructed near infinity degenerate. We may however show that the solutions constructed near 0 in fact have the correct exponential behavior at infinity. We observe that we may always represent the differential equation (D.4) by

$$(\widetilde{J} e^{\sigma_j \gamma_2 t})^{(3)} + \varepsilon_2 (\widetilde{J} e^{\sigma_j \gamma_2 t})^{(2)} + \varepsilon_1 (\widetilde{J} e^{\sigma_j \gamma_2 t})^{(1)} + \varepsilon_0 (\widetilde{J} e^{\sigma_j \gamma_2 t}) = 0, \quad \text{where}$$

$$\varepsilon_2 = -\frac{3\theta}{\sigma_j + t}, \quad \varepsilon_1 = 4\gamma_1 - \gamma_2^2 \sigma_j^2 + \frac{2\gamma_2 \sigma_j \theta}{\sigma_j + t} + \frac{2\theta^2 + 5\theta}{(\sigma_j + t)^2}, \quad \text{and} \tag{D.11}$$

$$\varepsilon_0 = -\frac{(4\gamma_1 - \gamma_2^2 \sigma_j^2)\theta}{\sigma_j + t} + \frac{(2\theta^2 - \theta)\gamma_2 \sigma_j}{(\sigma_j + t)^2} - \frac{4(\theta^2 + \theta)\gamma_2 \sigma_j}{(\sigma_j + t)^3}.$$

Hence with $Y := \widetilde{J} e^{\sigma_j \gamma_2 t}$, we can represent

$$\begin{bmatrix} Y^{(1)}(t) \\ Y^{(2)}(t) \\ Y^{(3)}(t) \end{bmatrix} = \begin{bmatrix} 0 & 1 & 0 \\ 0 & 0 & 1 \\ -\varepsilon_0(t) & -\varepsilon_1(t) & -\varepsilon_2(t) \end{bmatrix} \begin{bmatrix} Y(t) \\ Y^{(1)}(t) \\ Y^{(2)}(t) \end{bmatrix}$$

Hence if we let

$$\delta^2 := \max \left\{ |4\gamma_1 - \gamma_2^2 \sigma_j^2|, (\omega_j + t)^{-1} \right\}$$

we conclude, after conjugating

$$\begin{bmatrix} Y^{(1)}(t) \\ \delta^{-1} Y^{(2)}(t) \\ \delta^{-2} Y^{(3)}(t) \end{bmatrix} = \begin{bmatrix} 0 & \delta & 0 \\ 0 & 0 & \delta \\ -\varepsilon_0(t)\delta^{-2} & -\varepsilon_1(t)\delta^{-1} & -\varepsilon_2(t) \end{bmatrix} \begin{bmatrix} Y(t) \\ \delta^{-1} Y^{(1)}(t) \\ \delta^{-2} Y^{(2)}(t) \end{bmatrix}. \tag{D.12}$$

We define

$$N := \max\{|Y|, |\tfrac{Y'}{\delta}|, |\tfrac{Y''}{\delta^2}|\} \quad \text{and} \quad A := \max\{\delta, |\varepsilon_2| + |\varepsilon_1 \delta^{-1}| + |\varepsilon_0 \delta^{-2}|\}$$

which are $\ell^\infty$ norms of the matrix and vector that appear on the right-hand-side of (D.12). Moreover, taking the time derivative of $N$ we conclude the differential inequality

$$N'(t) \leq A(t) N(t) + \mathbf{1}[\delta^2 = (\omega_j + t)^{-1}] \frac{N(t)}{\omega_j + t}.$$

There is a continuous function $M$ of the parameters $(\gamma_1, \gamma_2, \sigma_j, \theta)$ so that $A(t)$ can be bounded by a multiple of $M\delta(t)$ for all $t \geq 1$. Applying Gronwall's inequality, it follows that for any $t \geq t_0 \geq 1$ that

$$N(t) \leq N(t_0) \exp\left( M \int_{t_0}^t \delta(s) \, ds \right). \tag{D.13}$$

We use this to conclude the fundamental matrix $\mathfrak{P}$ has reasonable decay properties for $4\gamma_1 - \gamma_2^2 \sigma_j^2$ small.

**Lemma D.3.** *Let* $\omega := 4\gamma_1 - \gamma_2^2\sigma_j^2$ *and suppose that* $|\omega| \leq 1$. *There is a continuous function* $M := M(\gamma_1, \gamma_2, \sigma_j, \theta) \geq 0$ *on* $\mathbb{R}^4$ *so that for all* $t \geq s \geq 1$

$$\|\mathfrak{P}(t)\| \leq Me^{-(\gamma_2\sigma_j - M\sqrt{|\omega|})t} \quad \text{and} \quad \|\mathfrak{P}(t)\mathfrak{P}^{-1}(s)\| \leq Me^{-(\gamma_2\sigma_j - M\sqrt{|\omega|})(t-s)}$$

Note that the first inequality extends to all $t \geq -\sigma_j$ using Lemma D.2.

*Proof.* For the first bound, we apply (D.13) to each of $j_1, j_2, j_3$ separately. As $|\omega| = \delta^2 \leq 1$ and $Y(1), Y'(1), Y''(1)$ can all be bounded using Lemma D.2 by a continuous function, we conclude for some possibly larger $M > 0$

$$Y''(t) \leq Me^{Mt\delta}.$$

By integrating this bound, we conclude, by increasing $M$ as needed that there is some $M$ so that

$$\max\{|Y(t)|, |Y'(t)|, |Y''(t)|\} \leq Me^{Mt\delta}.$$

With $Y(t) = j_a(t)e^{\sigma_j\gamma_2 t}$, for $a \in \{1, 2, 3\}$ expressing the left-hand-side of the above in terms of $j_a$ and again increasing $M$ as needed, we conclude the first claimed bound.

For the second bound, the columns of $\mathfrak{P}(t)\mathfrak{P}^{-1}(s)$ solve (D.4) and they have identity initial conditions at $s$. Hence applying (D.13) to each, we derive the desired equation in the same fashion as above. $\quad\square$

## D.2 Near infinity

Throughout this section, we work in the regime that

$$|4\gamma_1 - \gamma_2^2\sigma_j^2| > \varepsilon$$

for some positive $\varepsilon$. This regime ensures that all the roots of the indicial equation for the ODE (D.4) are distinct near $\infty$. We are interested in deriving an expression for the kernel $K_s(t)$ in Corollary A.1 when $s$ and $t$ are large. Recall the three fundamental solutions near infinity that is $j_i(t)$ (D.7). We begin by defining three different boundary conditions that will aid us in finding the kernels we are interest in, that is,

$$\begin{aligned}
\text{(Dirichlet sol., } \widetilde{\mathcal{D}}_s(t)) \quad & \widetilde{L}[\widetilde{\mathcal{D}}_s(t)] = 0 \quad \text{where} \quad \widetilde{\mathcal{D}}_s(s) = (1, 0, 0)^T \\
\text{(Neumann sol., } \widetilde{\mathcal{N}}_s(t)) \quad & \widetilde{L}[\widetilde{\mathcal{N}}_s(t)] = 0 \quad \text{where} \quad \widetilde{\mathcal{N}}_s(s) = (0, 1, 0)^T \\
\text{(2nd derivative sol., } \widetilde{\mathcal{H}}_s(t)) \quad & \widetilde{L}[\widetilde{\mathcal{H}}_s(t)] = 0 \quad \text{where} \quad \widetilde{\mathcal{H}}_s(s) = (0, 0, 1)^T
\end{aligned} \quad \text{(D.14)}$$

We will compute the asymptotics for the Dirichlet solution $\widetilde{\mathcal{D}}_s(t)$ in full details; we leave out the details for the other two solutions noting that the same approach works.

Using the fundamental solutions near infinity, we can write the Dirichlet solution as a linear combination of the fundamental solutions,

$$\widetilde{\mathcal{D}}_s(t) = c_1^D(s)j_1(t) + c_2^D(s)j_2(t) + c_3^D(s)j_3(t).$$

We need to find the coefficients $c_1, c_2, c_3$ and to do so we utilize the fundamental matrix $\Phi(s)$,

$$\Phi(s) \stackrel{\text{def}}{=} \begin{bmatrix} j_1(s) & j_2(s) & j_3(s) \\ j_1'(s) & j_2'(s) & j_3'(s) \\ j_1''(s) & j_2''(s) & j_3''(s). \end{bmatrix} \quad \text{(D.15)}$$

In particular the coefficients $c_i^D$ are found by $\Phi(s)(c_1^D, c_2^D, c_3^D)^T = (1, 0, 0)^T$. Hence, we need to compute the inverse of $\Phi(s)$ which we do by Cramer's rule. First, we need an expression for the Wroskian $\mathscr{W}(s)$ which we wrote in (D.5) as a ratio. Since we are working in a neighborhood of $t = \infty$, we can compute the Wroskian using $t = \infty$ and therefore derive an expression for the $\mathscr{W}(s)$ for any $s$. A simple calculation yields the following expression

$$\begin{aligned}
\mathscr{W}(t) = \det\Phi(t) &\sim \exp((\lambda_1 + \lambda_2 + \lambda_3)t)(\sigma_j + t)^{\rho_1+\rho_2+\rho_3} \det\left(\begin{bmatrix} 1 & 1 & 1 \\ \lambda_1 & \lambda_2 & \lambda_3 \\ \lambda_1^2 & \lambda_2^2 & \lambda_3^2 \end{bmatrix}\right) \\
&= \exp((\lambda_1 + \lambda_2 + \lambda_3)t)(\sigma_j + t)^{\rho_1+\rho_2+\rho_3}(\lambda_3 - \lambda_2)(\lambda_3 - \lambda_1)(\lambda_2 - \lambda_1)
\end{aligned}$$

where the pair $(\lambda_i, \rho_i)$ corresponds to the fundamental solution $j_i$ and the determinant of the matrix is the determinant of the Vandermonde matrix. It is clear that $\lambda_1 + \lambda_2 + \lambda_3 = -3\sigma_j\gamma_2$, $\rho_1 + \rho_2 + \rho_3 = 3\theta$, and $(\lambda_3 - \lambda_2)(\lambda_3 - \lambda_1)(\lambda_2 - \lambda_1) = 2(\sigma_j^2\gamma_2^2 - 4\gamma_1)^{3/2}$. Hence, we have that $(t) \sim 2\exp(-3\sigma_j^2\gamma_1 t)(\sigma_j + t)^{3\theta}(\sigma_j^2\gamma_2^2 - 4\gamma_1)^{3/2}$. Combining this with (D.5), we get that

$$\frac{1}{\mathscr{W}(s)} = \frac{e^{3\gamma_2\sigma_j s}(\sigma_j + s)^{-3\theta}}{2(\sigma_j^2\gamma_2^2 - 4\gamma_1)^{3/2}}. \tag{D.16}$$

Here we used that we working in the regime where the denominator is bounded away from $0$. By Cramer's rule, we have an expression for the fundamental matrix $\Phi^{-1}(s)$

$$\Phi^{-1}(s) = \frac{1}{\mathscr{W}(s)} \begin{bmatrix} j_2' j_3'' - j_2'' j_3' & j_2'' j_3 - j_2 j_3'' & j_2 j_3' - j_2' j_3 \\ j_1'' j_3' - j_1' j_3'' & j_1 j_3'' - j_1'' j_3 & j_1' j_3 - j_1 j_3' \\ j_1' j_2'' - j_2' j_1'' & j_2 j_1'' - j_1 j_2'' & j_1 j_2' - j_1' j_2 \end{bmatrix}$$

$$\sim \frac{1}{2(\sigma_j^2\gamma_2 - 4\gamma_1)^{3/2}} \begin{bmatrix} (\lambda_2\lambda_3^2 - \lambda_2^2\lambda_3)j_1^{-1} & (\lambda_2^2 - \lambda_3^2)j_1^{-1} & (\lambda_3 - \lambda_2)j_1^{-1} \\ (\lambda_3\lambda_1^2 - \lambda_3^2\lambda_1)j_2^{-1} & (\lambda_3^2 - \lambda_1^2)j_2^{-1} & (\lambda_1 - \lambda_3)j_2^{-1} \\ (\lambda_1\lambda_2^2 - \lambda_1^2\lambda_2)j_3^{-1} & (\lambda_1^2 - \lambda_2^2)j_3^{-1} & (\lambda_2 - \lambda_1)j_3^{-1} \end{bmatrix}$$

$$= \frac{1}{2(\sigma_j^2\gamma_2 - 4\gamma_1)^{3/2}} \begin{bmatrix} \lambda_2\lambda_3(\lambda_3 - \lambda_2)j_1^{-1} & (\lambda_2 - \lambda_3)(\lambda_2 + \lambda_3)j_1^{-1} & (\lambda_3 - \lambda_2)j_1^{-1} \\ \lambda_3\lambda_1(\lambda_1 - \lambda_3)j_2^{-1} & (\lambda_3 - \lambda_1)(\lambda_3 + \lambda_1)j_2^{-1} & (\lambda_1 - \lambda_3)j_2^{-1} \\ \lambda_1\lambda_2(\lambda_2 - \lambda_1)j_3^{-1} & (\lambda_1 - \lambda_2)(\lambda_1 + \lambda_2)j_3^{-1} & (\lambda_2 - \lambda_1)j_3^{-1} \end{bmatrix}.$$

We used that $j_i^{(\ell)}(s) \sim \lambda_i^\ell j_i(s)$ and we simplified the Wronskian by pulling out the appropriate terms. To simplify some of the terms, we let $\omega(\sigma) = 4\gamma_1 - \gamma_2^2\sigma^2$. We can now compute the coefficients for the various boundary solutions using that $2(\sigma_j^2\gamma_2^2 - 4\gamma_1)^{3/2} = (\lambda_3 - \lambda_2)(\lambda_3 - \lambda_1)(\lambda_2 - \lambda_1)$

$$(c_1^D(s), c_2^D(s), c_3^D(s))^T = \left( \frac{4\gamma_1}{\omega}j_1^{-1}, \frac{\sigma_j\gamma_2(\sigma_j\gamma_2 + \sqrt{-\omega})}{-2\omega}j_2^{-1}, \frac{\sigma_j\gamma_2(\sigma_j\gamma_2 - \sqrt{-\omega})}{-2\omega}j_3^{-1} \right)^T (1 + \mathcal{O}(\varepsilon))$$

$$(c_1^N(s), c_2^N(s), c_3^N(s))^T = \left( \frac{2\sigma_j\gamma_2}{\omega}j_1^{-1}, \frac{2\sigma_j\gamma_2 + \sqrt{-\omega}}{-2\omega}j_2^{-1}, \frac{2\sigma_j\gamma_2 - \sqrt{-\omega}}{-2\omega}j_3^{-1} \right)^T (1 + \mathcal{O}(\varepsilon))$$

$$(c_1^H(s), c_2^H(s), c_3^H(s))^T = \left( \frac{j_1^{-1}}{\omega}, \frac{-j_2^{-1}}{2\omega}, \frac{-j_3^{-1}}{2\omega} \right)^T (1 + \mathcal{O}(\varepsilon)).$$

We conclude the following bounds for the fundamental matrix:

**Lemma D.4.** *Let $\varepsilon > 0$ be arbitrary and suppose that $\omega = 4\gamma_1 - \gamma_2^2\sigma_j^2$ satisfies $|\omega| > \varepsilon$. There is a continuous function $M_\epsilon := M_\epsilon(\gamma_1, \gamma_2, \theta)$ so that for all $t \geq s \geq 1$,*

$$\|\Phi(t)\Phi^{-1}(s)\| \leq M_\epsilon e^{-(\gamma_j\gamma_2 - \sqrt{\max\{-\omega, 0\}})(t-s)}\frac{(\sigma_j + t)^\theta}{(\sigma_j + s)^\theta}.$$

*If on the other hand, $s \leq 1$, we instead have*

$$\|\Phi(t)\Phi^{-1}(s)\| \leq M_\epsilon e^{-(\gamma_j\gamma_2 - \sqrt{\max\{-\omega, 0\}})(t-s)}\frac{(\sigma_j + t)^\theta}{(\sigma_j + s)^{2\theta}}.$$

*Finally, we have the asymptotic representation for the fundamental solutions for $t \geq s$*

$$\widetilde{\mathcal{H}}_s(t) = \left( \frac{1 - \cos\left(\sqrt{\omega}(t-s) + \log\left(\frac{\sigma_j + t}{\sigma_j + s}\right)\frac{2\gamma_2\sigma_j\theta}{\sqrt{\omega}}\right) + o_{s,\varepsilon}(1)}{\omega} \right) \exp\left(-\sigma_j\gamma_2(t - s)\right)\frac{(\sigma_j + t)^\theta}{(\sigma_j + s)^\theta},$$

$$\widetilde{\mathcal{N}}_s(t) = 2\sigma_j\gamma_2\widetilde{\mathcal{H}}_s(t) + \left( \frac{\sin\left(\sqrt{\omega}(t-s) + \log\left(\frac{\sigma_j + t}{\sigma_j + s}\right)\frac{2\gamma_2\sigma_j\theta}{\sqrt{\omega}}\right) + o_{s,\varepsilon}(1)}{\sqrt{\omega}} \right) \exp\left(-\sigma_j\gamma_2(t - s)\right)\frac{(\sigma_j + t)^\theta}{(\sigma_j + s)^\theta},$$

$$\widetilde{\mathcal{D}}_s(t) = \sigma_j\gamma_2\widetilde{\mathcal{N}}_s(t) - (\sigma_j\gamma_2)^2\widetilde{\mathcal{H}}_s(t) + (1 + o_{s,\varepsilon}(1))\exp\left(-\sigma_j\gamma_2(t - s)\right)\frac{(\sigma_j + t)^\theta}{(\sigma_j + s)^\theta},$$

*where the error $o_{s,\varepsilon}(1)$ tends to $0$ uniformly with $s$ uniformly on compact sets of the parameter space where $|\omega| > \varepsilon$.*

*Proof.* The bounds follow from estimating above the fundamental solutions (D.7) and the Wronskian formula (D.16). By combining these with Lemma D.2, we can extend the formula to $s \leq 1$, (using

$$\Phi(t)\Phi^{-1}(s) = \Phi(t)\Phi^{-1}(1)\Phi(1)\Phi^{-1}(s) = \Phi(t)\Phi^{-1}(1)\mathfrak{P}(1)\mathfrak{P}^{-1}(s),$$

using uniqueness of the IVP). The final asymptotics follow from the display above the statement of the lemma. $\qquad\square$

**The scaling limit of the fundamental solutions as $\sigma$ tends to $0$.** We conclude with:

**Lemma D.5.** *Let $\mathfrak{d}$ be the solution of (D.4) on $[0, \infty)$ with $\sigma = 0$ that has the property that $\mathfrak{d}(t)t^{-2\theta} \to 1$ as $t \to 0$. This could also be expressed as the limit as $\sigma \to 0$ of $\mathfrak{j}_1$. For any $\varepsilon > 0$*

$$\sup_{t \in [\varepsilon, \infty)} (\sigma_j + t)^{-\theta} |\sigma^{2\theta} \cdot \widetilde{\mathcal{D}}_0(t) - \mathfrak{d}(t)| \xrightarrow[\sigma \to 0]{} 0,$$

$$\sup_{t \in [\varepsilon, \infty)} (\sigma_j + t)^{-\theta} |\sigma^{2\theta - 1} \cdot \widetilde{\mathcal{N}}_0(t) - \mathfrak{d}(t)| \xrightarrow[\sigma \to 0]{} 0,$$

$$\sup_{t \in [\varepsilon, \infty)} (\sigma_j + t)^{-\theta} |\sigma^{2\theta - 2} \cdot \widetilde{\mathcal{H}}_0(t) - \mathfrak{d}(t)| \xrightarrow[\sigma \to 0]{} 0,$$

*Proof.* We can represent $\widetilde{\mathcal{D}}_0(t), \widetilde{\mathcal{N}}_0(t), \widetilde{\mathcal{H}}_0(t)$ as the entries in the first row of

$$\Phi(t)\Phi^{-1}(0) = \Phi(t)\Phi^{-1}(1)\mathfrak{P}(1)\mathfrak{P}^{-1}(0).$$

The matrix in the middle $\Phi^{-1}(1)\mathfrak{P}(1)$ converges to a nondegenerate matrix as $\sigma \to 0$. The first row of $\Phi(t)$, given by $j_1, j_2, j_3$ each converge to solutions $a_1, a_2, a_3$ of (D.4) as $\sigma \to 0$ in the sense that for any $\varepsilon > 0$

$$\sup_{t \in [\varepsilon, \infty)} (\sigma + t)^{-\theta} |j_k(t) - a_k(t)| \xrightarrow[\sigma \to 0]{} 0.$$

The columns of $\mathfrak{P}^{-1}(0)$ behave like

$$\mathfrak{P}^{-1}(0) \underset{\sigma \to 0}{\asymp} \begin{bmatrix} \sigma^{-2\theta} & \sigma^{1-2\theta} & \sigma^{2-2\theta} \\ \sigma^{-1-\theta} & \sigma^{-\theta} & \sigma^{1-\theta} \\ \sigma^{-2} & \sigma^{-1} & 1 \end{bmatrix},$$

where we mean that the ratios of the respective entries converge to a nonzero constant as $\sigma \to 0$. To see that the limits that result are always equal to $\mathfrak{d}$, we can instead represent $\widetilde{\mathcal{D}}_0(t), \widetilde{\mathcal{N}}_0(t), \widetilde{\mathcal{H}}_0(t)$ as the first row of $\mathfrak{P}(t)\mathfrak{P}^{-1}(0)$. On taking $\sigma \to 0$, only the multiple of $\mathfrak{j}_1$ survives. $\qquad\square$

**The fundamental solutions of the unscaled ODE.** Finally, we relate the estimates we have made back to the unscaled differential differential equation (D.4) and (D.17). So we set

$$\begin{aligned} \text{(Dirichlet sol., } \mathcal{D}_s(t)) \quad &L[\mathcal{D}_s(t)] = 0 \quad \text{where} \quad \mathcal{D}_s(s) = (1, 0, 0)^T, \\ \text{(Neumann sol., } \mathcal{N}_s(t)) \quad &L[\mathcal{N}_s(t)] = 0 \quad \text{where} \quad \mathcal{N}_s(s) = (0, 1, 0)^T, \\ \text{(2nd derivative sol., } \mathcal{H}_s(t)) \quad &L[\mathcal{H}_s(t)] = 0 \quad \text{where} \quad \mathcal{H}_s(s) = (0, 0, 1)^T. \end{aligned} \tag{D.17}$$

To make the connection to (D.17), we observe that an initial value problem

$$L[f(t)] = 0 \quad \text{and} \quad \begin{bmatrix} f(t_0) \\ f'(t_0) \\ f''(t_0) \end{bmatrix} = \begin{bmatrix} c_1 \\ c_2 \\ c_3 \end{bmatrix} \quad \longleftrightarrow \quad \widetilde{L}[f(t/\sigma_j)] = 0 \text{ and } \begin{bmatrix} f(t_0/\sigma_j) \\ \partial_{t_0/\sigma_j} f(t_0/\sigma_j) \\ \partial^2_{t_0/\sigma_j} f(t_0/\sigma_j) \end{bmatrix} = \begin{bmatrix} c_1 \\ c_2/\sigma_j \\ c_3/\sigma_j^2 \end{bmatrix}.$$

Thus we have the identification

$$\mathcal{D}_s^{(\sigma_j)^2}(t) = \widetilde{\mathcal{D}}_{s\sigma_j}(t\sigma_j), \quad \mathcal{N}_s^{(\sigma_j)^2}(t) = \widetilde{\mathcal{N}}_{s\sigma_j}(t\sigma_j)/\sigma_j, \quad \mathcal{H}_s^{(\sigma_j)^2}(t) = \widetilde{\mathcal{H}}_{s\sigma_j}(t\sigma_j)/\sigma_j^2. \tag{D.18}$$

Using Lemma D.4, it is possible to give asymptotic expressions for these kernels and corresponding estimates.

We recall that we can express the terms in the Volterra equation for SDANA as

$$\mathbb{E} f(\boldsymbol{X}_t) = R h_1(t) + \widetilde{R} h_0(t) + \int_0^t \mathcal{K}_s(t) \, \mathbb{E} f(\boldsymbol{X}_s) \, \mathrm{d}s,$$

for a given spectral measure $\mu$ (especially, the empirical spectral measure or the limiting empirical spectral measure) by

$$G^{(\sigma^2)}(t) := \left( \mathcal{D}_0^{(\sigma^2)}(t) + (2\theta - 2\gamma_2\sigma^2)\mathcal{N}_0^{(\sigma^2)}(t) + ((2\theta - 2\gamma_2\sigma^2)^2 - 2\gamma_1\sigma^2 - 2\theta)\mathcal{H}_0^{(\sigma^2)}(t) \right),$$

$$h_0(t) := \frac{1}{2\varphi^2(t)} \int_0^\infty G^{(\sigma^2)}(t)\mu(\mathrm{d}\sigma^2) \quad \text{and} \quad h_1(t) := \frac{1}{2\varphi^2(t)} \int_0^\infty \sigma^2 G^{(\sigma^2)}(t)\mu(\mathrm{d}\sigma^2)$$

$$\mathcal{K}_s(t) := \frac{\varphi^2(s)}{\varphi^2(t)} \int_0^\infty \sigma^4 \left( \gamma_2^2 \mathcal{D}_s^{(\sigma^2)}(t) + \left( 2\gamma_2\gamma_1 + 2\gamma_2^2 \left( \frac{\theta}{1+s} - \gamma_2\sigma^2 \right) \right) \mathcal{N}_s^{(\sigma^2)}(t) \right)\mu(\mathrm{d}\sigma^2)$$

$$+ \frac{\varphi^2(s)}{\varphi^2(t)} \int_0^\infty \sigma^4 \left( \gamma_2^2 \left[ \frac{4\theta^2 - 2\theta}{(1+s)^2} - \frac{8\theta\gamma_2\sigma^2}{1+s} + 4\sigma^4\gamma_2^2 \right] - 8\gamma_1\gamma_2^2\sigma^2 + \frac{6\theta\gamma_2\gamma_1}{1+s} + 2\gamma_1^2 \right) \mathcal{H}_s^{(\sigma^2)}(t)\mu(\mathrm{d}\sigma^2)$$

$$\tag{D.19}$$

**Reduction to a convolution kernel.** We work under the assumption that the support of $\mu$ is contained in $[\lambda_-, \lambda_+]$.

For the kernel, we start by using the asymptotics for $\mathcal{D}, \mathcal{N}, \mathcal{H}$ in Lemma D.4, which give

$$\mathcal{H}_s(t) = \left( \frac{1 - \cos\left(\sigma\sqrt{\omega}(t-s) - \log\left(\frac{1+t}{1+s}\right)\frac{\theta\gamma_2\sigma}{\sqrt{\omega}}\right) + o_{s,\varepsilon}(1)}{\sigma^2\omega} \right) \exp\left(-\sigma^2\gamma_2(t-s)\right) \frac{(1+t)^\theta}{(1+s)^\theta},$$

$$\mathcal{N}_s(t) = 2\sigma^2\gamma_2\mathcal{H}_s(t) + \left( \frac{\sin\left(\sigma\sqrt{\omega}(t-s) - \log\left(\frac{1+t}{1+s}\right)\frac{\theta\gamma_2\sigma}{\sqrt{\omega}}\right) + o_{s,\varepsilon}(1)}{\sigma\sqrt{\omega}} \right) \exp\left(-\sigma^2\gamma_2(t-s)\right) \frac{(1+t)^\theta}{(1+s)^\theta},$$

$$\mathcal{D}_s(t) = \sigma^2\gamma_2\mathcal{N}_s(t) - \sigma^4\gamma_2^2\mathcal{H}_s(t) + (1 + o_{s,\varepsilon}(1)) \exp\left(-\sigma^2\gamma_2(t-s)\right) \frac{(1+t)^\theta}{(1+s)^\theta}.$$

$$\tag{D.20}$$

To apply these asymptotics we need to cut out a window $I_\epsilon$ of $\sigma$ for which $\omega = \omega(\sigma) = 4\gamma_1 - \gamma_2^2\sigma^2$ is small. So for an $\epsilon > 0$ let $I_\epsilon$ be those $\sigma$ for which $|\omega| < \epsilon$. If $\omega(\sigma)$ is bounded away from $0$ on the support of $\mu$ we may simply take $\epsilon = 0$ in what follows. By tracking the leading terms, we arrive at

$$\mathcal{K}_s(t) = (1 + o_s(1))\frac{2\gamma_1^2\varphi(s)}{\varphi(t)} \int_{I_\epsilon} \sigma^2 e^{-\sigma^2\gamma_2(t-s)} \left( \frac{1 - \cos\left(\vartheta(\sigma) + \sigma\sqrt{\omega}(t-s) - \log\left(\frac{1+t}{1+s}\right)\frac{\theta\gamma_2\sigma}{\sqrt{\omega}}\right)}{\omega} \right)\mu(\mathrm{d}\sigma^2)$$

$$+ \mathcal{O}\left(\epsilon e^{-(4\gamma_1\gamma_2^{-1} + M\epsilon)(t-s)}\right)$$

$$\tag{D.21}$$

where $\vartheta(\sigma)$ is a phase depending on $\gamma_1, \gamma_2, \sigma$, having $\vartheta(\sigma) \sim -2\sqrt{\gamma_1\omega(\sigma)}$ as $\omega \to 0$. The phase is defined explicitly by

$$\cos(\vartheta(\sigma)) = \frac{(\omega - 2\gamma_1)^2 - 2\gamma_1^2}{2\gamma_1^2},$$

$$\sin(\vartheta(\sigma)) = \frac{(\omega - 2\gamma_1)\sqrt{4\gamma_1 - \omega}\sqrt{\omega}}{2\gamma_1^2}.$$

$$\tag{D.22}$$

This is essentially a convolution type Volterra kernel, and so we simplify it by using an idealized kernel. Define

$$\mathcal{I}_s(t) = \mathcal{I}(t - s) = 2\gamma_1^2 \int_0^\infty \sigma^2 e^{-\sigma^2\gamma_2(t-s)} \left( \frac{1 - \cos\left(\vartheta(\sigma) + \sigma\sqrt{\omega}(t-s)\right)}{\omega} \right)\mu(\mathrm{d}\sigma^2). \quad \text{(D.23)}$$

This is a convolution kernel, which is comparable in norm to $\mathcal{K}_s(t)$. As the theory for positive convolution kernels is substantially simpler, we turn to studying the equation:

$$\Psi(t) = R\varphi(t)h_1(t) + \widetilde{R}\varphi(t)h_0(t) + \int_0^t \mathcal{I}(t - s)\Psi(s) \, \mathrm{d}s. \quad \text{(D.24)}$$

We will reduce the asymptotics of $\psi$ to those of $\Psi$.

A simple computation gives that the $L^1$ norm of the $\mathcal{I}$ is given by

$$\|\mathcal{I}\| = \int_0^\infty \mathcal{I}(t)\,dt = \int_{0+}^\infty \tfrac{\gamma_1 + \sigma^2 \gamma_2^2}{2\gamma_2}\mu(d\sigma^2), \qquad (D.25)$$

which gives a sufficient condition for neighborhood convergence. Provided that the measure $\mu$ puts no mass at the critical point, the behavior of solutions (D.24) are related to the original Volterra equation.

**Proposition D.1.** *Provided* $\|\mathcal{I}\| < 1$,

$$\mathbb{E}_{\boldsymbol{H}}\, f(\boldsymbol{X}_t) \xrightarrow[t\to\infty]{} \frac{\widetilde{R}\mu(\{0\})}{1 - \|\mathcal{I}\|}.$$

*Proof.* The forcing functions $h_k$ satisfy

$$Rh_1 + \widetilde{R}h_0 \xrightarrow[t\to\infty]{} \widetilde{R}\mu(\{0\}).$$

Moreover, for any $\epsilon > 0$ it can be decomposed into two pieces,

$$Rh_1 + \widetilde{R}h_0 = F_0 + F_\epsilon,$$

the first of which is regularly varying and the latter of which is bounded by $\epsilon$ and tends to 0 as $t \to \infty$. This comes by decomposing the eigenvalues into those separated from the critical point $\{4\gamma_1/\gamma_2^2\}$ and those in a neighborhood of it. Now it follows that solving the Volterra equation with $F_0$,

$$X_0(t) = \varphi(t)F_0(t) + \int_0^t \mathcal{I}(t-s)X_0(s)\,ds,$$

which from Lemma C.1

$$X_0(t) \underset{t\to\infty}{\sim} \frac{\varphi(t)F_0(t)}{1 - \|\mathcal{I}\|}.$$

For the second piece, we have that for

$$X_1(t) = \varphi(t)F_1(t) + \int_0^t \mathcal{I}(t-s)X_1(s)\,ds,$$

we conclude

$$X_1(t) \le \frac{\epsilon}{1 - \|\mathcal{I}\|} \quad \text{and} \quad X_1/\varphi(t) \xrightarrow[t\to\infty]{} 0.$$

Combining everything we conclude that by taking $\epsilon \to 0$

$$\Psi(t)/\varphi(t) \xrightarrow[t\to\infty]{} \frac{\widetilde{R}\mu(\{0\})}{1 - \|\mathcal{I}\|}.$$

By taking differences, we turn to bounding

$$\tfrac{\Psi(t)}{\varphi(t)} - \mathbb{E}_{\boldsymbol{H}}\, f(\boldsymbol{X}_t) = \int_0^t \mathcal{K}_s(t)\big(\tfrac{\Psi(s)}{\varphi(s)} - \mathbb{E}_{\boldsymbol{H}}\, f(\boldsymbol{X}_s)\big)\,ds + \int_0^t \big(\mathcal{I}(t-s)\tfrac{\varphi(s)}{\varphi(t)} - \mathcal{K}_s(t)\big)\tfrac{\Psi(s)}{\varphi(s)}\,ds.$$

Using that there is an $\epsilon > 0$ and a $C > 0$ so that

$$|\mathcal{I}(t-s)\tfrac{\varphi(s)}{\varphi(t)} - \mathcal{K}_s(t)| \le C(1+s)^{-1}\mathcal{I}(t-s) + \epsilon e^{-(4\gamma_1/\gamma_2 - C\epsilon)(t-s)}, \qquad (D.26)$$

we conclude that this error term tends to 0. The resolvent $R_s(t)$ of $\mathcal{K}_s(t)$ is bounded in $L^1$ using standard theory (see [Gripenberg, 1980, Theorem 3]) and by comparison to $\mathcal{I}$. Then

$$\tfrac{\Psi(t)}{\varphi(t)} - \mathbb{E}_{\boldsymbol{H}}\, f(\boldsymbol{X}_t) = \int_0^t R_x(t)\int_0^x \big(\mathcal{I}(x-s)\tfrac{\varphi(s)}{\varphi(t)} - \mathcal{K}_s(t)\big)\tfrac{\Psi(s)}{\varphi(s)}\,ds\,dx.$$

Then applying Fubini

$$\tfrac{\Psi(t)}{\varphi(t)} - \mathbb{E}_{\boldsymbol{H}}\, f(\boldsymbol{X}_t) = \int_0^t \big(\mathcal{I}(x-s)\tfrac{\varphi(s)}{\varphi(t)} - \mathcal{K}_s(t)\big)\tfrac{\Psi(s)}{\varphi(s)}\left(\int_s^t R_x(t)\,dx\right)ds.$$

Bounding the integral of $R_x(t)$ and using the bound (D.26), it follows we have that

$$\big(\tfrac{\Psi(t)}{\varphi(t)}\big)^{-1}|\tfrac{\Psi(t)}{\varphi(t)} - \mathbb{E}_{\boldsymbol{H}}\, f(\boldsymbol{X}_t)| \xrightarrow[t\to\infty]{} 0.$$

$\square$

**Average case analysis in the strongly convex case.** We suppose now that we have taken the limit of empirical spectral measures, and consider a measure $\mu$ with support $\{0\} \cup [\lambda^-, \lambda^+]$ for some $\lambda^- > 0$. We suppose that $\lambda^-$ is not at the critical point, i.e. $4\gamma_1 - \gamma_2^2\lambda^- \neq 0$. We further suppose that $\mu$ has a density with regular boundary behavior at $\lambda^-$:

$$\mu([\lambda^-, \lambda^- + \epsilon]) \underset{\epsilon \to 0}{\sim} \ell\epsilon^\alpha. \tag{D.27}$$

We need to derive the asymptotic behavior of $h_0, h_1, \mathcal{K}$. It is convenient if we remove the effect of any point mass of $\mu$ at $0$, which effects the eventual convergence of the algorithm. Set $\widetilde{h}_0 = h_0 - \mu(\{0\})$ and $\widetilde{h}_1 = h_1$. This leads to precise asymptotics of the forcing function $\widetilde{h}_k$, as from the asymptotics of $j_0, j_1, j_2$ we have (recalling $\omega(\sqrt{\lambda^-}) = 4\gamma_1 - \gamma_2^2\lambda^-$) we have for $k \in \{0, 1\}$

$$\widetilde{h}_k(t) \underset{t \to \infty}{\sim} \ell_k e^{-\gamma_2\lambda^- t}(1+t)^{-\theta} \begin{cases} 1 + c_1\cos\left(\sqrt{\lambda^-\omega}t - \log(1+t)\frac{\theta\gamma_2\sqrt{\lambda^-}}{\sqrt{\omega}} + c_2\right), & \text{if } \omega > 0, \\ e^{\lambda^-\sqrt{-\omega}t}(1+t)^{-\frac{\theta\gamma_2\sqrt{\lambda^-}}{\sqrt{-\omega}}}, & \text{if } \omega < 0. \end{cases} \tag{D.28}$$

**Malthusian exponent.** We define the Malthusian exponent $\lambda^*$, if it exists, as the solution of

$$\int_0^\infty e^{\lambda^* t}\mathcal{I}(t)\,dt = 1.$$

We observe that using (D.22) we can represent for any $\lambda < \sigma^2\gamma_2$

$$\int_0^\infty \sigma^2 e^{\lambda t - \sigma^2\gamma_2 t}\left(\frac{1 - \cos\left(\vartheta(\sigma) + \sigma\sqrt{\omega}t\right)}{\omega}\right)dt$$
$$= \left(\frac{\gamma_2\sigma^2\omega(\omega - 2\gamma_1) - (\sigma^2\gamma_2 - \lambda)((\omega - 2\gamma_1)^2 - 2\gamma_1^2)}{2\gamma_1^2((\sigma^2\gamma_2 - \lambda)^2 + \sigma^2\omega)} + \frac{1}{\sigma^2\gamma_2 - \lambda}\right)\frac{\sigma^2}{\omega}. \tag{D.29}$$

On specializing to $\lambda = 0$, we can further simplify this to

$$\int_0^\infty \sigma^2 e^{-\sigma^2\gamma_2 t}\left(\frac{1 - \cos\left(\vartheta(\sigma) + \sigma\sqrt{\omega}t\right)}{\omega}\right)dt = \frac{\gamma_1 + \sigma^2\gamma_2^2}{4\gamma_1^2\gamma_2}.$$

Returning to (D.29) and algebraically simplifying the expression, we can can write $\lambda^*$ as the solution

$$1 = \int_0^\infty e^{\lambda^* t}\mathcal{I}(t)\,dt = \int_0^\infty \sigma^4\left(\frac{\gamma_2^2(\sigma^2\gamma_2 - \lambda^*)^2 + \gamma_2(\omega - 2\gamma_1)(\sigma^2\gamma_2 - \lambda^*) + 2\gamma_1^2}{((\sigma^2\gamma_2 - \lambda^*)^2 + \sigma^2\omega)(\sigma^2\gamma_2 - \lambda^*)}\right)\mu(d\sigma^2).$$

We let $\mathcal{F}(\lambda^*)$ be the expression on the right hand side. We note that expression is necessarily increasing in $\lambda^*$ (which is clear from the expression $\mathcal{F}(\lambda) = \int_0^\infty e^{\lambda t}\mathcal{I}(t)\,dt$). Furthermore, for $\lambda > \lambda_-$, $\mathcal{F}(\lambda) = \infty$ as $\mathcal{I}(t)$ decays no slower than $e^{-\lambda_- t}$. Provided $\alpha > 1$ (recall (D.27)), then $\mathcal{F}(\lambda_-) < \infty$, and thus the existence of the Malthusian exponent $\lambda^*$ is equivalent to $\mathcal{F}(\lambda_-) \geq 1$.

**Proposition D.2.** *Suppose that (D.27) holds for some $\lambda_- > 0$ with $\omega(\lambda_-) \neq 0$ and for $\alpha > 1$. Then if $\mathcal{F}(\lambda_-) < 1$ the solution of (D.24) satisfies*

$$\frac{\Psi(t)}{\varphi(t)} - \frac{\widetilde{R}\mu(\{0\})}{1 - \|\mathcal{I}\|} \sim ce^{-\gamma_2\lambda_- t}t^{-\alpha - \theta}$$

*for some $c > 0$ or if $\mathcal{F}(\lambda_-) > 1$ then with $\lambda^*$ the unique solution of $\mathcal{F}(\lambda^*) = 1$ for some constant $c > 0$*

$$\frac{\Psi(t)}{\varphi(t)} - \frac{\widetilde{R}\mu(\{0\})}{1 - \|\mathcal{I}\|} \sim ce^{-\lambda_* t}t^{-\theta}.$$

*Proof.* This follows standard renewal theory machinery. See Asmussen [2003] or [Paquette et al., 2021, Theorem 29]. □

**Lemma D.6.** *Then for $\lambda_- > 0$ for which $\omega(\lambda_-) \neq 0$ and if $\mu(0) = 0$,*

$$\Psi(t)^{-1}|\Psi(t) - \varphi(t)\psi(t)| \underset{t \to \infty}{\longrightarrow} 0.$$

*Proof.* We start from the raw Volterra equation for $\psi$ which is given by

$$\psi(t) = Rh_1(t) + \widetilde{R}h_0(t) + \int_0^t \mathcal{K}_s(t)\psi(s)\,\mathrm{d}s.$$

Multiplying through by $\varphi(t)$, we therefore have

$$\varphi(t)\psi(t) = R\varphi(t)h_1(t) + \widetilde{R}\varphi(t)h_0(t) + \int_0^t \mathcal{K}_s(t)\tfrac{\varphi(t)}{\varphi(s)}\varphi(s)\psi(s)\,\mathrm{d}s.$$

This allows us to express the difference $\Psi(t) - \varphi(t)\psi(t)$ as

$$\Psi(t) - \varphi(t)\psi(t) = \int_0^t \mathcal{K}_s(t)\tfrac{\varphi(t)}{\varphi(s)}\big(\Psi(s) - \varphi(s)\psi(s)\big)\,\mathrm{d}s + \int_0^t \big(\mathcal{I}_s(t) - \mathcal{K}_s(t)\tfrac{\varphi(t)}{\varphi(s)}\big)\Psi(s)\,\mathrm{d}s. \quad \text{(D.30)}$$

We can dominate the kernel above and below by

$$\mathcal{K}_s(t)\tfrac{\varphi(t)}{\varphi(s)} = (1 + \mathcal{O}(s^{-1}))\mathcal{I}_s(t),$$

with the error uniform in $t \geq s$; this uses Lemma D.3 and the asymptotic representations of the fundamental solutions Lemma D.4 (see also (D.21)). Let $\lambda$ be the Malthusian exponent, if it exists, or $\lambda_-$ otherwise. The latter forcing term of (D.30) can be bounded by

$$\left| \int_0^t e^{-\lambda s}\big(\mathcal{I}_s(t) - \mathcal{K}_s(t)\tfrac{\varphi(t)}{\varphi(s)}\big)e^{\lambda s}\Psi(s)\,\mathrm{d}s \right| \leq \int_0^t C(1+s)^{-1}e^{-\lambda s}\mathcal{I}(t-s)e^{\lambda s}\Psi(s)\,\mathrm{d}s.$$

In the case that $\lambda$ is the Malthusian exponent, we have that $e^{\lambda s}\Psi(s)$ is bounded and $e^{-\lambda s}\mathcal{I}(s)$ has $L^1$–norm 1. It follows that

$$\int_0^t (1+s)^{-1}e^{-\lambda s}\mathcal{I}(t-s)e^{\lambda s}\Psi(s)\,\mathrm{d}s \lesssim e^{-\lambda t}\int_0^t (1+(t-s))^{-1}e^{-\lambda s}\mathcal{I}(s)\,\mathrm{d}s.$$

Thus by dominated convergence, we have that the forcing term satisfies

$$F(t) := e^{\lambda t}\left| \int_0^t e^{-\lambda s}\big(\mathcal{I}_s(t) - \mathcal{K}_s(t)\tfrac{\varphi(t)}{\varphi(s)}\big)e^{\lambda s}\Psi(s)\,\mathrm{d}s \right| \to 0.$$

From Gronwall's inequality [Gripenberg et al., 1990, 9.8.2] we conclude there is a non-negative resolvent kernel $r(t,s)$ so that

$$e^{\lambda t}|\Psi(t) - \varphi(t)\psi(t))| \leq \int_0^t r(t, t-s)|F(t-s)|\,\mathrm{d}s. \quad \text{(D.31)}$$

We deduce that the kernel has *bounded uniformly continuous type* (see [Gripenberg et al., 1990, Theorem 9.5.4]; see also [Gripenberg et al., 1990, Theorem 9.9.1]) and therefore satisfies

$$\lim_{h \to \infty} \sup_{t \geq 0} \int_0^{t-h} r(t, u)\,\mathrm{d}u = 0.$$

From here it follows from (D.31) that

$$e^{\lambda t}|\Psi(t) - \varphi(t)\psi(t))| \to 0$$

as $t \to \infty$, and hence from the asymptotics of $\Psi$, the same holds when dividing by $\Psi$.

For the case where $\lambda$ is not the Malthusian exponent, we must conclude a slightly stronger bound. This follows from first showing that the forcing function and the kernel $\mathcal{I}(t)$ both decay like $e^{-\lambda t}t^{-\alpha}$. By conjugating the problem by $(1+t)^\alpha$, we reduce the problem to the same strategy as used above. $\quad \square$

**Average case analysis in the non–strongly convex case.** We turn to the assumption that $\mu$ is contained in $[0, \lambda_+]$, with a possible atom at 0 and a density that is bounded away from its endpoints and that moreover $\mu$ has regular boundary behavior at 0 with

$$\mu((0, \epsilon]) \underset{\epsilon \to 0}{\sim} \ell\epsilon^\alpha. \quad \text{(D.32)}$$

In the case of Marchenko–Pastur, this $\alpha = 1/2$. We again need the behavior of $h_k$ for $k \in \{0, 1\}$. From the boundary condition at 0, we have that for $k \in \{0, 1\}$

$$\widetilde{h}_k(t) \sim \frac{\ell\alpha}{2\varphi^2(t)} \int_0^\infty (\sigma^2)^{\alpha-1}\sigma^{2k}\left(\mathcal{D}_0^{(\sigma^2)}(t) + 2\theta\mathcal{N}_0^{(\sigma^2)}(t) + 4\theta^2\mathcal{H}_0^{(\sigma^2)}(t)\right) \mathrm{d}(\sigma^2).$$

Now as we are in a neighborhood of $\sigma \approx 0$ we use the scaled solutions (D.18), due to which we can express the solution as

$$\widetilde{h}_k(t) \sim \frac{\ell\alpha}{2\varphi^2(t)} \int_0^\infty (\sigma^2)^{\alpha-1}\sigma^{2k}\left(\widetilde{\mathcal{D}}_0^{(\sigma^2)}(t\sigma) + 2\sigma^{-1}\theta\widetilde{\mathcal{N}}_0^{(\sigma^2)}(t\sigma) + 4\sigma^{-2}\theta^2\widetilde{\mathcal{H}}_0^{(\sigma^2)}(t\sigma)\right) \mathrm{d}(\sigma^2).$$

We pick a $\varepsilon > 0$ and decompose the integral according to $t\sigma > \varepsilon$ and those below. For those $\sigma$ above, we use Lemma D.5 and conclude

$$\begin{aligned}
\widetilde{h}_k(t) &\sim \frac{\ell\alpha}{2\varphi(t)} \int_{\varepsilon^{-1}t^{-1}}^\infty (\sigma^2)^{\alpha+k-1}\mathcal{O}\left(e^{-\sigma^2\gamma_2 t}\right) \mathrm{d}(\sigma^2) \\
&+ \frac{\ell\alpha}{2\varphi^2(t)} \int_{\varepsilon^2 t^{-2}}^{\varepsilon^{-1}t^{-1}} (\sigma^2)^{\alpha+k-1-\theta}\mathfrak{d}(t\sigma)\left(1 + 2\theta + 4\theta^2\right) \mathrm{d}(\sigma^2) \\
&+ \frac{\ell\alpha}{2\varphi^2(t)} \int_0^{\varepsilon^2 t^{-2}} (\sigma^2)^{\alpha+k-1}\left(1 + 2t\theta + 2t^2\theta^2 + o_\varepsilon(1)\right) \mathrm{d}(\sigma^2)
\end{aligned}$$

Both first and last integrals will be negligible. For the middle integral, we change variables with $x^2 = \sigma^2 t^2$ to get

$$\widetilde{h}_k(t) \sim \frac{\ell\alpha}{2t^{2\alpha+2k}} \int_{\varepsilon^2}^{\varepsilon^{-1}t} (x^2)^{\alpha+k-1-\theta}\mathfrak{d}(x)\left(1 + 2\theta + 4\theta^2\right) \mathrm{d}(x^2).$$

The integral is convergent when $2\alpha + 2k - \theta < 0$ as $\mathfrak{d}$ grows like $x^\theta$ as $x \to \infty$ and $\mathfrak{d}$ tends to 0 like $x^{2\theta}$ as $x \to 0$. Thus we may take $\varepsilon \to 0$ and conclude

$$\widetilde{h}_k(t) \sim \frac{\ell\alpha}{2t^{2\alpha+2k}}\left(1 + 2\theta + 4\theta^2\right) \int_0^\infty (x^2)^{\alpha+k-1-\theta}\mathfrak{d}(x) \, \mathrm{d}(x^2). \tag{D.33}$$

We again use the approximate convolution structure of the kernel, in particular the kernel $\mathcal{I}$ and the approximate Volterra equation (D.24).

**Proposition D.3.** *When $\|\mathcal{I}\| < 1$ and $\widetilde{R} = 0$ and $\theta > 2\alpha + 2$ it follows*

$$\Psi(t)/\varphi(t) \underset{t\to\infty}{\sim} \frac{R\ell\alpha}{2t^{2\alpha+2}} \frac{(1 + 2\theta + 4\theta^2)}{1 - \|\mathcal{I}\|} \int_0^\infty (x^2)^{\alpha-\theta}\mathfrak{d}(x) \, \mathrm{d}(x^2).$$

*In the case that $\widetilde{R} > 0$,*

$$\Psi(t)/\varphi(t) \underset{t\to\infty}{\sim} \frac{\widetilde{R}\ell\alpha}{2t^{2\alpha}} \frac{(1 + 2\theta + 4\theta^2)}{1 - \|\mathcal{I}\|} \int_0^\infty (x^2)^{\alpha-1-\theta}\mathfrak{d}(x) \, \mathrm{d}(x^2).$$

*Proof.* The proposition is a corollary of Lemma C.2, using $\mathcal{I}(t) \sim c(\mu, \theta)t^{-2\alpha-4}$ and (D.33). $\quad\square$

Finally, we can derive the needed bound for the original problem:

**Lemma D.7.** *When $\|\mathcal{I}\| < 1$ and (D.32) holds*

$$\psi(t) \underset{t\to\infty}{\sim} \Psi(t)/\varphi(t).$$

*Proof.* This follows the same strategy as Lemma D.6. $\quad\square$

## D.3 Rate bounds

We let $\lambda^-$ be the left endpoint of the support of $\mu$ restricted to $(0, \infty)$. If $\lambda^- = 0$, we are in the non–strongly convex case above, and the rate of convergence is polynomial for any choice of step size that is convergent. We show a step size choice that gives a good rate for all $\lambda^- > 0$ separately.

We conclude with bounds for the convolution kernel $\mathcal{I}$ which establish bounds for the rate under step size conditions which are strictly better than (D.25). We shall work under that $\gamma_1$ and $\gamma_2$ satisfy the condition

$$\int_{0+}^{\infty} \frac{\gamma_1 \Delta + \sigma^2 \gamma_2^2}{2\gamma_2} \mu(\mathrm{d}\sigma^2) \leq 1 \tag{D.34}$$

for some $\Delta > 1$, which ensures that the algorithm converges.

We recall that the Malthusian exponent is defined as the solution $\lambda^*$ of

$$1 = \mathcal{F}(\lambda^*) := \int_0^{\infty} \sigma^4 \left( \frac{\gamma_2^2(\sigma^2\gamma_2 - \lambda^*)^2 + \gamma_2(\omega - 2\gamma_1)(\sigma^2\gamma_2 - \lambda^*) + 2\gamma_1^2}{((\sigma^2\gamma_2 - \lambda^*)^2 + \sigma^2\omega)(\sigma^2\gamma_2 - \lambda^*)} \right) \mu(\mathrm{d}\sigma^2),$$

if it exists.

We shall produce a bound for $\mathcal{F}(\lambda)$ for $\lambda$ sufficiently small, namely:

**Lemma D.8.** *Suppose that $\Delta > 1$ and $\lambda^- > 0$ and that $\lambda_*$ is defined by*

$$\lambda_* = \frac{\lambda^- \gamma_2 + 2\frac{\gamma_1}{\gamma_2} - \sqrt{(\lambda^- \gamma_2 + 2\frac{\gamma_1}{\gamma_2})^2 - 8\gamma_1 \lambda^-(1 - \frac{1}{\Delta})}}{2},$$

*then for all $\sigma^2 \geq \lambda^-$,*

$$\mathrm{II} := \sigma^4 \left( \frac{\gamma_2^2(\sigma^2\gamma_2 - \lambda)^2 + \gamma_2(\omega - 2\gamma_1)(\sigma^2\gamma_2 - \lambda) + 2\gamma_1^2}{((\sigma^2\gamma_2 - \lambda)^2 + \sigma^2\omega)(\sigma^2\gamma_2 - \lambda)} \right) \leq \frac{\gamma_1 \Delta + \sigma^2 \gamma_2^2}{2\gamma_2}.$$

*Moreover, we have the bounds*

$$\lambda_* \geq \frac{2\gamma_1 \lambda^-(1 - \frac{1}{\Delta})}{\gamma_2 \lambda^- + 2\frac{\gamma_1}{\gamma_2}}.$$

This leads immediately to a rate bound:

**Corollary D.1.** *At the default parameters of SDANA $\gamma_2 = (\mathrm{tr}(\mu))^{-1}$ where $\mathrm{tr}(\mu) = \int_0^{\infty} \sigma^2 \mu(\mathrm{d}\sigma^2)$ and $\gamma_1 = \frac{\gamma_2}{4}$ the convergence rate is at least $\frac{3}{8} \min\{(\mathrm{tr}(\mu))^{-1}\lambda^-, \frac{1}{2}\}$. The fastest possible rate, in contrast, is no larger than $\min\{(\mathrm{tr}(\mu))^{-1}\lambda^-, \frac{1}{2}\}$.*

*Proof.* For the rates, we apply Lemma C.3. Lemma D.8 gives a lower bound on the Malthusian exponent, where we take $\Delta = 4$. As for the rate of the forcing function, we have that its rate is bounded by

$$\begin{cases} \gamma_2 \lambda^-, & \text{if } \omega = 4\gamma_1 - \gamma_2^2 \lambda^- > 0 \text{ or} \\ \gamma_2 \lambda^- - \sqrt{\gamma_2^2(\lambda^-)^2 - 4\gamma_1 \lambda^-}, & \text{otherwise.} \end{cases}$$

This is always bounded above by $\min\{\gamma_2 \lambda^-, 2\frac{\gamma_1}{\gamma_2}\}$. For convergence we should have

$$\|\mathcal{I}\| = \int_{0+}^{\infty} \frac{\gamma_1 + \sigma^2 \gamma_2^2}{2\gamma_2} \mu(\mathrm{d}\sigma^2) \leq 1,$$

and thus optimizing in taking $\gamma_2 \lambda^- = 2\gamma_1/\gamma_2$ and the above norm equal to 1, we conclude the fastest rate is at most $\frac{4\lambda^-}{\lambda^- + 2\mathrm{tr}(\mu)}$. This is in turn at most the claimed amount. $\square$

*Proof of Lemma.* We begin with some simplifications. The claimed bound is equivalent to

$$2\gamma_2 \sigma^4 \left(2\gamma_1 \gamma_2 - \lambda\gamma_2^2 + \frac{2\gamma_1^2}{\sigma^2\gamma_2 - \lambda}\right) \leq \left(\gamma_1 \Delta + \sigma^2 \gamma_2^2\right)\left(\lambda^2 - 2\lambda\sigma^2\gamma_2 + 4\gamma_1\sigma^2\right).$$

After cancelling terms and rearranging

$$\frac{4\gamma_1^2 \gamma_2 \sigma^4}{\sigma^2\gamma_2 - \lambda} \leq \gamma_1 \Delta\left(-2\lambda\sigma^2\gamma_2 + 4\gamma_1\sigma^2\right) + \left(\gamma_1 \Delta + \sigma^2\gamma_2^2\right)\lambda^2.$$

Hence dropping the $\lambda^2$ term and simplifying, it suffices that

$$\text{III} := \frac{2\gamma_1\sigma^2}{\left(2\frac{\gamma_1}{\gamma_2} - \lambda\right)\left(\sigma^2\gamma_2 - \lambda\right)} \le \Delta.$$

The map $x \mapsto \frac{x}{x\gamma_2 - \lambda}$ is decreasing for $x\gamma_2 > \lambda$, and hence it suffices that

$$\text{IV} := \frac{2\gamma_1\lambda^-}{\left(2\frac{\gamma_1}{\gamma_2} - \lambda\right)\left(\lambda^-\gamma_2 - \lambda\right)} \le \Delta.$$

It follows that for all $\lambda$ less than the smallest root of

$$\lambda^2 - \lambda(\lambda^-\gamma_2 + 2\tfrac{\gamma_1}{\gamma_2}) + 2\gamma_1\lambda^-(1 - \tfrac{1}{\Delta}) = 0,$$

$\text{IV} \le \Delta$. Solving for the smaller root $\lambda_*$, we have

$$\lambda \le \lambda_* = \frac{\lambda^-\gamma_2 + 2\frac{\gamma_1}{\gamma_2} - \sqrt{(\lambda^-\gamma_2 + 2\frac{\gamma_1}{\gamma_2})^2 - 8\gamma_1\lambda^-(1 - \frac{1}{\Delta})}}{2}. \tag{D.35}$$

Using concavity of the square root, we can bound $\sqrt{a + x} \le \sqrt{a} + \frac{x}{2\sqrt{a}}$ and so conclude

$$\lambda_* \ge \frac{2\gamma_1\lambda^-(1 - \frac{1}{\Delta})}{\gamma_2\lambda^- + 2\frac{\gamma_1}{\gamma_2}}.$$

$\square$

# E   The general SDAHB kernel

In this section, we analyze in detail a general version of SDAHB where we also include a $\gamma_2$ that is, we consider an algorithm SDA where $\gamma_1, \gamma_2 > 0$ and $\Delta(k, n) = \frac{\theta}{n}$. In this general setting, the log-derivative $\Phi(t) = \theta$. We recall the ODE (A.12) that describes this process (where $\sigma_j^2 = \lambda$)

$$\begin{aligned}
&\widehat{J}^{(3)} + (-3\theta + 3\gamma_2\lambda)\,\widehat{J}^{(2)} + \left(2\theta^2 - 4\gamma_2\lambda\theta + 4\gamma_1\lambda + 2\gamma_2^2\lambda^2\right)\widehat{J}^{(1)} \\
&+ \left(-4\gamma_1\lambda\theta + 4\gamma_1\gamma_2\lambda^2\right)\widehat{J} \\
&= \tfrac{\gamma_2^2}{\gamma_1}\widehat{\psi}^{(2)} + \left(2\gamma_2 + (-\theta + \gamma_2\lambda)\tfrac{\gamma_2^2}{\gamma_1}\right)\widehat{\psi}^{(1)} + \left(2\gamma_1 + 2\gamma_1\lambda\tfrac{\gamma_2^2}{\gamma_1}\right)\widehat{\psi}.
\end{aligned} \tag{E.1}$$

The initial conditions are given by

$$\begin{aligned}
&\widehat{J}(0) = \gamma_1^{-1}\mathbb{E}\left[\left(\nu_{0,j} - \tfrac{(\boldsymbol{U}^T\boldsymbol{\eta})_j}{\sigma_j}\right)^2\right], \quad \widehat{J}^{(1)}(0) = \tfrac{\gamma_2^2}{\gamma_1}\widehat{\psi}(0) - \widehat{J}(0)\left(-2\theta + 2\gamma_2\lambda\right), \quad \text{and} \\
&\widehat{J}^{(2)}(0) = \tfrac{\gamma_2^2}{\gamma_1}\widehat{\psi}^{(1)}(0) + 2\gamma_2\widehat{\psi}(0) - 2\gamma_1\lambda\widehat{J}(0) + (2\theta - 2\gamma_2\lambda)\widehat{J}^{(1)}(0).
\end{aligned} \tag{E.2}$$

We note that the ODE in (E.1) is constant coefficient and therefore can be solved by finding the characteristic polynomial, that is,

$$\begin{aligned}
0 &= \xi^3 + (3\gamma_2\lambda - 3\theta)\xi^2 + (2\theta^2 - 4\gamma_2\lambda\theta + 4\gamma_1\lambda + 2\gamma_2^2\lambda^2)\xi + 4\gamma_1\gamma_2\lambda^2 - 4\gamma_1\lambda\theta \\
0 &= (\xi + \lambda\gamma_2 - \theta)(\xi^2 + (2\lambda\gamma_2 - 2\theta)\xi + 4\lambda\gamma_1) \\
\xi &= \theta - \lambda\gamma_2 \quad \text{and} \quad \xi = -(\lambda\gamma_2 - \theta) \pm \sqrt{(\lambda\gamma_2 - \theta)^2 - 4\lambda\gamma_1}.
\end{aligned}$$

It immediately follows that the solutions to (E.1) are linear combinations of $\exp(-(\lambda\gamma_2 - \theta)t)$ and $\exp(-(\lambda\gamma_2 - \theta) \pm \sqrt{(\lambda\gamma_2 - \theta)^2 - 4\lambda_j\gamma_1})$. We now write the Dirichlet, Neumann, and 2nd-derivative solutions for which we will use to derive the kernel and the forcing term. For convenience, we denote $\omega \stackrel{\text{def}}{=} 4\lambda\gamma_1 - (\lambda\gamma_2 - \theta)^2$ and $\rho \stackrel{\text{def}}{=} \lambda\gamma_2 - \theta$. Taking derivatives, we get the following expressions for $K_s(t)$:

$$\begin{aligned}
K_s(t) &= \exp(-t\rho)\big(c_1 + c_2\exp(-t\sqrt{-\omega}) + c_3\exp(t\sqrt{-\omega})\big) \\
\tfrac{d}{dt}K_s(t) &= -\rho\exp(-t\rho)\big(c_1 + c_2\exp(-t\sqrt{-\omega}) + c_3\exp(t\sqrt{-\omega})\big) \\
&\quad + \sqrt{-\omega}\exp(-t\rho)\big(c_3\exp(t\sqrt{-\omega}) - c_2\exp(-t\sqrt{-\omega})\big) \\
\tfrac{d}{dt^2}K_s(t) &= \rho^2\exp(-t\rho)\big(c_1 + c_2\exp(-t\sqrt{-\omega}) + c_3\exp(t\sqrt{-\omega})\big) \\
&\quad + 2\rho\sqrt{-\omega}\exp(-t\rho)\big(c_2\exp(-t\sqrt{-\omega}) - c_3\exp(t\sqrt{-\omega})\big) \\
&\quad - \omega\exp(-t\rho)\big(c_2\exp(-t\sqrt{-\omega}) + c_3\exp(t\sqrt{-\omega})\big).
\end{aligned}$$

Provided that $\omega \neq 0$, we can now solve for $c_1, c_2, c_3$ for the Dirichlet, Neumann, and 2nd-derivative solutions,

$$
\begin{aligned}
\text{(Dirichlet sol., } \mathcal{D}_s(t)) & \quad L[\mathcal{D}_s(t)] = 0 \quad \text{where} \quad \mathcal{D}_s(s) = (1,0,0)^T, \\
\text{(Neumann sol., } \mathcal{N}_s(t)) & \quad L[\mathcal{N}_s(t)] = 0 \quad \text{where} \quad \mathcal{N}_s(s) = (0,1,0)^T, \\
\text{(2nd derivative sol., } \mathcal{H}_s(t)) & \quad L[\mathcal{H}_s(t)] = 0 \quad \text{where} \quad \mathcal{H}_s(s) = (0,0,1)^T.
\end{aligned}
\tag{E.3}
$$

To distinguish these solutions, we denote the coefficients by $c_i^D, c_i^N, c_i^H$ for $i = 1, 2, 3$. We begin by find the coefficients for $\mathcal{D}_s(t)$:

$$
(c_1^D, c_2^D, c_3^D) = \left( \exp(s\rho)(1 + \tfrac{\rho^2}{\omega}), \tfrac{1}{2}\exp(s(\rho + \sqrt{-\omega}))\left(\tfrac{-\rho^2}{\omega} - \tfrac{\rho}{\sqrt{-\omega}}\right), \tfrac{1}{2}\exp(s(\rho - \sqrt{-\omega}))\left(\tfrac{-\rho^2}{\omega} + \tfrac{\rho}{\sqrt{-\omega}}\right) \right)
$$

$$
(c_1^N, c_2^N, c_3^N) = \left( \exp(s\rho)\tfrac{2\rho}{\omega}, \tfrac{1}{2}\exp(s(\rho + \sqrt{-\omega}))\left(\tfrac{-2\rho}{\omega} - \tfrac{1}{\sqrt{-\omega}}\right), \tfrac{1}{2}\exp(s(\rho - \sqrt{-\omega}))\left(\tfrac{-2\rho}{\omega} + \tfrac{1}{\sqrt{-\omega}}\right) \right)
$$

$$
(c_1^H, c_2^H, c_3^H) = \left( \exp(s\rho)\tfrac{1}{\omega}, -\tfrac{1}{2\omega}\exp(s(\rho + \sqrt{-\omega})), -\tfrac{1}{2\omega}\exp(s(\rho - \sqrt{-\omega})) \right).
$$

We recall $J = \gamma_1 e^{-2\theta t} \widehat{J}$ and Corollary A.2 that

$$
J(t) = \gamma_1 e^{-2\theta t} \widehat{J}_0(t) + \gamma_1 e^{-2\theta t} \int_0^t K_s(t) \widehat{\psi}(s) \, ds.
$$

Using the coefficients in Corollary A.2, we write an expression for the forcing term

$$
\widehat{J}_0(t) = \frac{1}{2}\left(1 + \frac{\rho^2}{\omega}\right)\frac{1}{\gamma_1}\mathbb{E}\left[\left(\nu_{0,j} - \frac{(\boldsymbol{U}^T\boldsymbol{\eta})_j}{\sigma_j}\right)^2\right]e^{-t\rho}(1 + \cos(t\sqrt{\omega} + \vartheta_1)),
\tag{E.4}
$$

where the phase shift satisfies

$$
\cos(\vartheta_1) = \frac{\left(1 + \frac{\rho}{\sqrt{-\omega}}\right)^2 + \left(1 - \frac{\rho}{\sqrt{-\omega}}\right)^2}{2\left(1 + \frac{\rho^2}{\omega}\right)} = \frac{\omega - \rho^2}{\rho^2 + \omega}
$$

$$
\sin(\vartheta_1) = \frac{\left(1 - \frac{\rho}{\sqrt{-\omega}}\right)^2 - \left(1 + \frac{\rho}{\sqrt{-\omega}}\right)^2}{2\left(1 + \frac{\rho^2}{\omega}\right)}i = \frac{2\rho\sqrt{\omega}}{\rho^2 + \omega}.
$$

$$\tag{E.5}$$

We now give an expression for the kernel $K_s(t)$:

$$
K_s(t) = \frac{1}{2}\left(\frac{\gamma_2^2}{\gamma_1} + \frac{\gamma_2^2\rho^2}{\omega\gamma_1} + \frac{4}{\omega}(\gamma_1 - \gamma_2\rho)\right)e^{-(t-s)\rho}\left(1 + \cos((t-s)\sqrt{\omega} + \vartheta_2)\right)
\tag{E.6}
$$

where we have

$$
\cos(\vartheta_2) = \frac{2\left(\frac{\gamma_2^2}{4\gamma_1} - \frac{\gamma_2^2\rho}{4\omega\gamma_1} + \frac{\gamma_2\rho}{\omega} - \frac{\gamma_1}{\omega}\right)}{\frac{1}{2}\left(\frac{\gamma_2^2}{\gamma_1} + \frac{\gamma_2^2\rho^2}{\gamma_1\omega} + \frac{4}{\omega}(\gamma_1 - \gamma_2\rho)\right)} = \frac{\gamma_2^2\omega - \gamma_2^2\rho + 4\gamma_2\rho\gamma_1 - 4\gamma_1^2}{\gamma_2^2\omega + \gamma_2^2\rho^2 - 4\gamma_2\gamma_1\rho + 4\gamma_1^2}
$$

$$
\sin(\vartheta_2) = \frac{\frac{\gamma_2^2\rho}{\gamma_1\sqrt{\omega}} - \frac{2\gamma_2}{\sqrt{\omega}}}{\frac{1}{2}\left(\frac{\gamma_2^2}{\gamma_1} + \frac{\gamma_2^2\rho^2}{\gamma_1\omega} + \frac{4}{\omega}(\gamma_1 - \gamma_2\rho)\right)} = \frac{2(\gamma_2^2\rho\sqrt{\omega} - 2\gamma_2\gamma_1\sqrt{\omega})}{\gamma_2^2\omega + \gamma_2^2\rho^2 - 4\gamma_2\gamma_1\rho + 4\gamma_1^2}.
$$

$$\tag{E.7}$$

It follows that $J(t)$ is the sum of (E.4) and (E.6). We now recall that $\widehat{\psi}(s) = \frac{2\lambda\exp(2\theta s)\psi^{(n)}(s)}{n}$. Finally we arrive at the Volterra equation

$$
\psi(t) = \frac{1}{2}\int_0^\infty \lambda\gamma_1 e^{-2\theta t}\widehat{J}_0^{(\lambda)}(t)\,d\mu(\lambda) + \gamma_1\int_0^t\int_0^\infty \sigma^4 e^{-2\theta(t-s)}K_s^{(\lambda)}(t)\,d\mu(\lambda)\,\psi(s)\,ds.
$$

**Proposition E.1** (Volterra equation for general SDAHB with parameters$(\gamma_1, \gamma_2, e^{\theta t})$). *The Volterra equation for the general SDAHB with step size parameters $\gamma_1, \gamma_2 > 0$ and $\varphi(t) = e^{\theta t}$ is*

$$
G^{(\lambda)}(t) = \frac{1}{4}\left(1 + \frac{\rho^2}{\omega}\right)e^{-t(\rho + 2\theta)}(1 + \cos(t\sqrt{\omega} + \vartheta_1))
$$

$$
K_s^{(\lambda)}(t) = \frac{\lambda^2}{2}\left(\gamma_2^2 + \frac{\gamma_2^2\rho^2}{\omega} + \frac{4}{\omega}(\gamma_1^2 - \gamma_2\gamma_1\rho)\right)e^{-(t-s)(\rho + 2\theta)}\left(1 + \cos((t-s)\sqrt{\omega} + \vartheta_2)\right),
$$

$$\tag{E.8}$$

*where $\omega = 4\lambda\gamma_1 - (\lambda\gamma_2 - \theta)^2$, $\rho = \lambda\gamma_2 - \theta$, and $\vartheta_1$ and $\vartheta_2$ are defined in (E.5) and (E.7) respectively.*

**Corollary E.1** (Volterra equation for SDAHB). *The Volterra equation for SDAHB with step size parameters $\gamma_1 > 0$, $\gamma_2 = 0$, and $\varphi(t) = e^{\theta t}$ is*

$$G^{(\lambda)}(t) = \frac{1}{4}\left(1 + \frac{\theta^2}{\omega}\right)e^{-t\theta}(1 - \cos(t\sqrt{\omega} + \vartheta_1))$$

$$\text{and} \quad K_s^{(\lambda)}(t) = \frac{2\gamma_1^2\lambda^2}{\omega}e^{-(t-s)\theta}\left(1 - \cos((t-s)\sqrt{\omega})\right),$$

*where $\omega = 4\lambda\gamma_1 - \theta^2$ and $\vartheta_1$ is defined by*

$$\cos(\vartheta_1) = \frac{\theta^2 - \omega}{\theta^2 + \omega} \qquad \text{and} \qquad \sin(\vartheta_1) = \frac{2\theta\sqrt{\omega}}{\theta^2 + \omega}. \tag{E.10}$$

## E.1 Convergence analysis for SDAHB

The interaction kernel for SDAHB is therefore of convolution type, and we have

$$F(t) = \int_0^\infty G^{(\lambda)}(t)\mu(\mathrm{d}\lambda) \quad \text{and} \quad \mathcal{I}(t) = \int_0^\infty K_0^{(\lambda)}(t)\mu(\mathrm{d}\lambda).$$

The loss of homogenized SGD then satisfies

$$\mathbb{E}_{\boldsymbol{H}}\, f(\boldsymbol{X}_t) = F(t) + \int_0^t \mathcal{I}(t-s)\, \mathbb{E}_{\boldsymbol{H}}\, f(\boldsymbol{X}_s)\, \mathrm{d}s.$$

Computing the Laplace transform of this kernel for all $x$ sufficiently small,

$$\mathcal{F}(x) \stackrel{\text{def}}{=} \int_0^\infty e^{xt}\mathcal{I}(t)\, \mathrm{d}t = \int_0^\infty \frac{2\gamma_1^2\lambda^2}{(\theta - x)(x^2 - 2\theta x + 4\gamma_1\lambda)}\mu(\mathrm{d}\lambda),$$

and recall that the Malthusian exponent $\lambda^*$ is defined as the root of $\mathcal{F}(\lambda^*) = 1$, if it exists. In particular evaluating at $x = 0$, we compute the norm

$$\|\mathcal{I}\| = \int_0^\infty \frac{2\gamma_1^2\lambda^2}{\theta(4\gamma_1\lambda)}\mu(\mathrm{d}\lambda) = \frac{\gamma_1}{2\theta}\int_0^\infty \lambda\mu(\mathrm{d}\lambda) = \frac{\gamma_1}{2\theta}\mathrm{tr}(\mu). \tag{E.11}$$

## E.2 Convergence analysis for SDAHB

We now suppose we have passed to a limiting measure $\mu$ with a support $\{0\} \cup [\lambda^-, \lambda^+]$ that satisfies

$$\mu([\lambda^-, \lambda^- + \epsilon]) \underset{\epsilon \to 0}{\sim} \ell\epsilon^\alpha. \tag{E.12}$$

The forcing function satisfies, with $\omega = \omega(\lambda^-) = 4\lambda^-\gamma_1 - \theta^2$ and for some constants $c, c_1, c_2$ depending on the algorithm parameters,

$$F(t) \underset{t\to\infty}{\sim} \begin{cases} ct^{\alpha-1}e^{-t(\theta - \sqrt{\theta^2 - 4\gamma_1\lambda^-})}, & \text{if} \quad \omega < 0, \\ ct^{\alpha+1}e^{-t\theta}, & \text{if} \quad \omega = 0, \\ (c_1t^{\alpha-1} + c_2t^{\alpha-1}\cos(t\sqrt{\omega}))e^{-t\theta}, & \text{if} \quad \omega > 0. \end{cases}$$

From standard renewal theory (Lemma C.3), we have that

**Proposition E.2.** *If $\mathcal{F}(\lambda^-) < 1$ and (E.12) holds*

$$\mathbb{E}_{\boldsymbol{H}}\, f(\boldsymbol{X}_t) - \frac{\widetilde{R}\mu(\{0\})}{1 - \|\mathcal{I}\|} = F(t)e^{o(t)}$$

*or if $\mathcal{F}(\lambda_-) > 1$ then with $\lambda^*$ the unique solution of $\mathcal{F}(\lambda^*) = 1$*

$$\mathbb{E}_{\boldsymbol{H}}\, f(\boldsymbol{X}_t) - \frac{\widetilde{R}\mu(\{0\})}{1 - \|\mathcal{I}\|} = e^{-(\lambda_* + o(1))t}.$$

We note that if we take the default parameters, we come within a factor of the maximum rate.

**Proposition E.3.** *Suppose we take the default parameters for SDAHB, that is*

$$\theta = 2 \quad \text{and} \quad \gamma_1 = \frac{\theta}{\text{tr}(\mu)} = \frac{\theta}{\int \lambda\mu(\mathrm{d}\lambda)},$$

*the rate of convergence is at least*

$$\lambda_* \geq \frac{\gamma_1 \lambda^- \theta}{2\gamma_1 \lambda^- + \theta^2} = \frac{2\gamma_1 \lambda^-}{2\gamma_1 \lambda^- + 4}.$$

*The fastest possible rate is at most $\frac{4\lambda^-}{\text{tr}(\mu)}$.*

*Proof.* We just need to bound $\mathcal{F}(x) \leq 1$ for $x \leq \min\{\frac{2\lambda^- \gamma_1}{\theta}, \frac{\theta}{2}\}$ and with the parameter choices made. By monotonicity

$$\mathcal{F}(x) \leq \int_0^\infty \frac{2\gamma_1^2 \lambda\lambda^-}{(\theta - x)(x^2 - 2\theta x + 4\gamma_1 \lambda^-)}\mu(\mathrm{d}\lambda) \leq \frac{2\gamma_1 \theta\lambda^-}{(\theta - x)(x^2 - 2\theta x + 4\gamma_1 \lambda^-)}.$$

We bound further from above by dropping the $x^2$ and then solving the result quadratic, i.e. $\mathcal{F}(x) \leq 1$ if

$$x \leq \frac{4\gamma_1 \lambda^- + 2\theta^2 - \sqrt{(4\gamma_1 \lambda^- + 2\theta^2)^2 - 8\theta(2\gamma_1 \lambda^- \theta)}}{4\theta}.$$

By concavity of the square root, it suffices to have

$$x \leq \frac{\gamma_1 \lambda^- \theta}{2\gamma_1 \lambda^- + \theta^2}.$$

The rate of $F$ is at most $\min\{\frac{2\gamma_1 \lambda^-}{\theta}, \theta\}$, and so optimizing this over $\frac{\gamma_1}{2\theta}\text{tr}(\mu) \leq 1$, we arrive at $\theta = \frac{4\lambda^-}{\text{tr}(\mu)}$ □

## E.3 Average-case rates non–strongly convex

We instead suppose the support is given by $[0, \lambda^+]$ and that

$$\mu((0, \varepsilon)) \underset{\epsilon \to 0}{\sim} \ell\epsilon^\alpha.$$

The forcing function, for any $\theta > 0$ then behaves like

$$F(t) \underset{t \to \infty}{\sim} c(Rt^{-\alpha-1} + \widetilde{R}t^{-\alpha}). \tag{E.13}$$

It follows using Lemma C.2 that when $\|\mathcal{I}\| < 1$, the same rate holds for $\mathbb{E} f(\boldsymbol{X}_t)$ up to multiplication by $(1 - \|\mathcal{I}\|)^{-1}$.

## E.4 Degeneration to SGD

**Theorem 5.** *Suppose the homogenized SGD diffusions for SHB and SGD are chosen so that $\gamma^{sgd} = \frac{\gamma^{shb}}{\theta^{shb}}$. Suppose that $n \to \infty$ and that $\boldsymbol{H}$ is chosen so that $\lambda_{\boldsymbol{H}}^+$ is bounded in $n$. Then for any $t > 0$*

$$|\mathbb{E}_{\boldsymbol{H}} f(\boldsymbol{X}_t^{shb}) - \mathbb{E}_{\boldsymbol{H}} f(\boldsymbol{X}_t^{sgd})| \underset{n \to \infty}{\longrightarrow} 0.$$

*Proof.* The homogenized SGD diffusion for SHB is the same as the diffusion for SDAHB with parameters $(\theta^{sdahb}, \gamma^{sdahb}) = (n\theta^{shb}, n\gamma^{shb})$ An elementary computation shows that uniformly in compact sets of $\lambda$ and $t$, the forcing function and interaction kernel ($G^{(\lambda)}$ and $K^{(\lambda)}$) of SDAHB with these parameters satisfy

$$G^{(\lambda)}(t) \underset{n \to \infty}{\longrightarrow} e^{-2\gamma^{sgd}\lambda t} \quad \text{and} \quad K^{(\lambda)}(t) \underset{n \to \infty}{\longrightarrow} \gamma^2 \lambda^2 e^{-2\gamma^{sgd}\lambda t}.$$

Thus under the assumption that the eigenvalues of $\boldsymbol{H}$ remain bounded as $n \to \infty$, the forcing function and kernel for each of $\mathbb{E} f(\boldsymbol{X}_t^{\text{SHB}})$ and $\mathbb{E} f(\boldsymbol{X}_t^{\text{sGD}})$ differ by an error that goes to 0 as $n \to \infty$ uniformly on compact sets of time. □

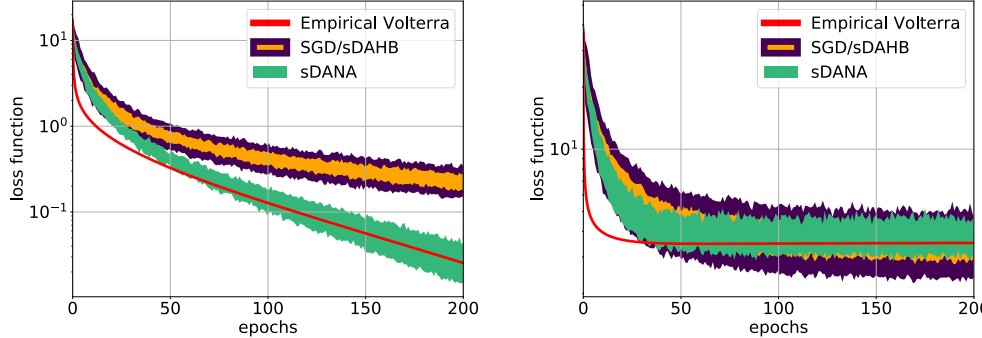

Figure 7: **SDANA & SGD vs Theory on MNIST.** MNIST ($60000 \times 28 \times 28$ images) [LeCun et al., 2010] is reshaped into 30 (left) and 60 (right) matrices of dimension $1000 \times 1568(784)$, representing 1000 samples of groups of 2 or 1 digits, respectively (preconditioned to have centered rows of norm-1). First digit of each 2 or 1 is chosen to be the target $b$. Algorithms were run 30(60) times with default parameters (without tuning) to solve (2.1). 80%–confidence interval is displayed. Volterra (SDANA) is generated with eigenvalues from the first MNIST data matrix with a ratio of signal-to-noise of 6-to-1. Volterra predicts the convergent behavior of SDANA in this non-idealized setting. SDANA outperforms equivalent SGD/SDAHB.

.

**Proposition E.4.** *For SGD, with default parameters $\gamma = \frac{1}{\int \lambda \mu(\mathrm{d}\lambda)}$, the Malthusian exponent is at least $\lambda_* \geq \gamma \lambda^-$.*

*Proof.* For SGD, the Malthusian exponent is given simply as the root of

$$1 = \mathcal{F}(x) = \int_0^\infty \frac{\gamma_1^2 \lambda^2}{(2\gamma\lambda - x)^2} \mu(\mathrm{d}\lambda) \leq \frac{\gamma \lambda^-}{2\gamma \lambda^- - x}.$$

(See Paquette et al. [2021] or send $\theta \to \infty$ with $\gamma_1 = \gamma\theta$ in SDAHB). Thus for $x = \gamma\lambda^-$ we have $\mathcal{F}(x) \leq 1$, and so $\lambda_* \geq \gamma\lambda^-$. $\qquad\square$

**Proposition E.5.** *For $\theta$ sufficiently large, and when $\mathcal{F}(\lambda^-) \leq 1$, SDAHB with parameters $(\gamma_1, \theta)$ is faster than SGD with parameters $\left(\gamma = \frac{\gamma_1}{\theta}\right)$ but never more than a factor of 2 than SGD at its default parameter.*

*Proof.* Note that for large $\theta$, with $\omega < 0$ we always have that $F(t)$ has rate

$$F(t) \sim ct^{\alpha-1}e^{-t(\theta - \sqrt{\theta^2 - 4\gamma_1\lambda^-})}.$$

The rate for $F$ satisfies

$$\theta - \sqrt{\theta^2 - 4\gamma_1\lambda^-} > \theta - \left(\theta - \frac{4\gamma_1\lambda^-}{2\theta}\right) = 2\gamma\lambda^-.$$

Moreover, the expression on the left is monotone decreasing $\theta$ until the argument of the radical becomes negative. Hence, we maximize the rate by taking the smallest admissible $\theta$, which at the convergence threshold is given by

$$\frac{\theta}{\gamma_1} = \frac{\int \lambda \mu(\mathrm{d}\lambda)}{2}.$$

Substituting this ratio into $\theta - \sqrt{\theta^2 - 4\gamma_1\lambda^-}$ to remove $\theta$ and then maximizing gives $\frac{2\lambda^-}{\int \lambda \mu(\mathrm{d}\lambda)}$, which is no more than a factor of 2 than SGD at its default parameter. $\qquad\square$

# F Numerical simulations

To illustrate our theoretical results, we report simulations using SGD, stochastic heavy-ball (SHB) [Polyak, 1964], SDAHB, SDANA (Table 1) on the least squares problem. In all simulations of the random least squares problem, the vectors $\boldsymbol{x}_0$, $\widetilde{\boldsymbol{x}}$, and $\boldsymbol{\eta}$, are sampled i.i.d. from a standard Gaussian $N(0, \frac{R}{2d}\boldsymbol{I})$, $N(0, \frac{R}{2d}\boldsymbol{I})$ and $N(0, \frac{\widetilde{R}}{n}\boldsymbol{I})$ respectively and the entries of $\boldsymbol{A}$, $A_{ij} \sim N(0, \frac{1}{d})$. Figures 1, 2, 5 are with noise; the first two have $R = \widetilde{R} = 1$ and the last is $R = 1 = 100\widetilde{R}$. Figure 3 is with noise 0.

**Volterra equation.** The forcing term $F(t)$ in (1) is solved by a Runge-Kutta method after which we applied a Chebyshev quadrature rule to approximate the integral with respect to the Marchenko-Pastur distribution. The Chebyshev quadrature is also used to derive a numerical approximation for the kernel, $\mathcal{I}(t - s)$, (3.1). Next, to generate the solution $\psi(t)$ of the Volterra equation, we implement a Picard iteration which finds a fix point to the Volterra equation by repeatedly convolving the kernel and adding the forcing term.

Despite the numerical approximations to integrals, the resulting solution to the Volterra equation ($\psi$, red lines in the plots) models the true behavior of all the stochastic algorithms analyzed in this paper remarkably well (see Fig. 1, 2, and 5). Notably, it captures the oscillatory trajectories in the momentum methods often is seen in practice due to their overshooting (see Fig. 5). We note that the Volterra equation for SDANA reliably undershoots simulations of SDANA for small time (say $t < 10$), but matches for larger times ($t > 100$). This is due in part because the convolution Volterra equation is only an approximation for SDANA that holds as time grows larger, and hence the undershoot is consistent with theory.

**Real data.** The MNIST examples (Figures 6 and 7) are shown to demonstrate that large–dimensional random matrix predictions often work for large dimensional real data. Figure 6 is strongly convex as $\lambda^- = 0.041$. This corresponds to a similar convexity structure as $r = 1.44$ in Marchenko-Pastur. Under this convexity, we do not expect SDANA to be faster than SGD/SDAHB. This is reflected in the figure as both SDANA and SGD are parallel to each other after $t > 50$. We chose to include the 6-sequential images in the main paper in order to show multiple properties of the algorithms in the same image: (1) empirical Volterra and SDANA matched and (2) SGD and SDAHB have similar dynamics. To see the behavior of the algorithms on a pure MNIST dataset, see Figure 7. As mentioned above, the Volterra equation always initially underestimates the dynamics of SDANA.