# OpenReview forum: "Dynamics of Stochastic Momentum Methods on Large-scale, Quadratic Models"
_NeurIPS.cc/2021/Conference — NeurIPS 2021 Poster_

### Official Review · Reviewer_ZKu7 · 2021-07-09

**Rating:** 6
**Confidence:** 4

**Summary:**

The paper studies the momentum SGD method (and some variants) in the high-dimensional setting. The paper, based on its analysis, proposes a new task-dependent algorithm for optimization

**Limitations And Societal Impact:**

I think the limitation can be better discussed. See above.

**Main Review:**

Understanding momentum in SGD is an important theoretical topic in deep learning, and studying this in the setting of high-dimensional regression is fairly novel.

I say "fairly novel" because I feel that the authors did not discuss the related works sufficiently to help me judge the novelty of the present work. For example, is this work the first and only work that deals with SGD in a high-dimensional setting? To my knowledge, the answer is no. The discrete-time SGD for high-dimensional regression has been analyzed in Ziyin2021, Section 4.4: arxiv.org/abs/2102.05375

Then the important question would be how does the result in this paper compare with the existing previous work. My first feeling is that the present work (in theoretical analysis) is more restrictive than Ziyin2021. For example, Ziyin2021 studies discrete-time SGD directly, while this work relies on continuous-time. Then why is this work novel? I am not suggesting that this work is not novel, but that the authors need to clearly discuss the relationship between the previous work to make its contribution stand out.

Also, one major weakness in the theory of the present work is that it relies on the assumption that the noise \eta is i.i.d. and independent of the data matrix A. However, this is not the case for SGD noise. Ziyin2021 has a very detailed study about noise here. See equation 13 of Ziyin2021; the noise eta in high-dimensional regression setting has a non-trivial noise structure, which is dependent on A and not i.i.d. I can accept more limiting assumptions than previous works, but the authors need to discuss this limitation clearly in the paper and convince the readers why the result may still be important despite this limitation. However, this assumption is given in the present paper in passing in assumption 1 without sufficient explanation. Also, I consider this a significant flaw because if the primary goal of the present work is to understand momentum in the setting of minibatch noise, then it must make sure it assumes a reasonable noise.

Also, I note an interesting difference in the result between the present paper and Ziyin2021. The present paper suggests that the step size of SGD should scale with 1/n, but Ziyin2021 Theorem 5 suggests that the step size should scale with 1/d. What causes this difference? How do I understand it?

I may recommend acceptance if the authors sufficiently address my questions.

**Time Spent Reviewing:**

2 hours.

---

> ### Author Response · Authors · 2021-08-08
> **Response to Reviewer ZKu7**
>
> We thank the reviewer for the introducing us to Ziyin2021, which we are happy to cite as there is some overlap in the topic.
>
> There are indeed some similarities between our submission and Ziyin2021:
> 1. Both look at discrete time SGD with momentum.
> 2. Both papers consider label noise, and as the reviewer points out, and Ziyin2021 does not assume the noise is independent.  While non-independent label noise is an interesting direction of generalization, there is still plenty to say about the case of independent noise, and it does have some realistic application (see Figure 6).
> 3. Both are about the high—dimensional setting.
>
> However, there are also *major* differences:
> 1. Section 4.2 of Ziyin2021 takes N to infinity while holding D fixed.  In this setting, new updates of the algorithm are independent of the past (e.g. Ziyin2021 studies streaming SGD with momentum), see Sec 4.2 paragraph after Eqn (3).  We consider the empirical loss setting where one can **reuse data**.
>
>      * These settings yield substantially different dynamics. Section 4.3 of Ziyin2021 gives a formula for the loss (without label noise) after 1 step, but the resulting formula is too complicated to have any clear consequences. To be relevant for us, one would need to compose this result repeatedly to get the $k$-th step. The loss was analyzed explicitly but only in the case of 1-dimensional SGD and only after one step of the algorithm (Appendix F.4.3 of Ziyin2021).
>
>      * We show that the loss converges to a solution of a Volterra equation under “homogenized SGD” as $n \to \infty$. One can use this Volterra equation to (1). predict the behavior of SGD with momentum, (2). analyze the average-case complexity of the algorithms and deduce faster convergence than previous worst-case complexities for the same algorithms (Table 3), and (3). produce near-optimal step-sizes with respect to optimizing complexity. These were not discussed in Ziyin2021.
>
> 2. Appendix F of Ziyin2021 deals with $N$ and $D$ large and proportional, but only in a neighborhood of the stationary point of SGD.  The results give the covariance structure of the updates started from stationarity.  While the $N$ proportional to $D$ case is our setup, our result is about the entire (non-stationary) trajectory of the loss, and the results are not directly comparable.
>
>      * We produce a continuous time approximation to SGD (and the whole sDA momentum class) called “Homogenized SGD”, Eqn (7). This approximation is proven to exactly match the loss function of the discrete time, finite-sum SGD on a random least-squares problem when N is large.  This continuous-time approximation is new.
>
> 3. We analyze a broader class of momentum based stochastic algorithms, including momentum parameters which change with dimension and/or time (e.g. stochastic Nesterov style algorithm).
>
> ---
>
> Response to specific comments:
>
> *Weakness in the theory of the present work is that it relies on the assumption that the noise $\eta$ is i.i.d. and independent of the data matrix A. However, this is not the case for SGD noise. Ziyin2021 has a very detailed study about noise here.*
>
> The noise ‘eta’ in our setup is the initial label noise; it is $\varepsilon$ in Ziyin2021.  We do not put any modeling assumptions on the ‘SGD noise’. We study actual SGD, but on a random least squares problem.
>
> *See equation 13 of Ziyin2021; the noise $\eta$ in high-dimensional regression setting has a non-trivial noise structure, which is dependent on A and not i.i.d.*
>
> The covariance referenced in equation 13 of Ziyin 2021 is not the covariance of our $\eta$.  Our ‘eta’ is the label noise.
>
> *Significant flaw if the primary goal of the present work is to understand momentum in the setting of minibatch noise, then it must make sure it assumes a reasonable noise.*
>
> Minibatch noise is an interesting problem. Our paper is about batch size 1 SGD with momentum. We again remark that our noise is on the labels. Generative models with this label noise have been used in numerous works (see e.g. [Hastie et al., 2019, Mei and Montanari, 2019]).
>
> *Also, I note an interesting difference in the result between the present paper and Ziyin2021. The present paper suggests that the step size of SGD should scale with 1/n, but Ziyin2021 Theorem 5 suggests that the step size should scale with 1/d. What causes this difference? How do I understand it?*
>
> If the problem setup is standardized, so that the row norms of the data matrix are 1, then the step size scales neither with d nor n. This is the convention used in our paper. Note that Thm 5 does not apply to our setting because Ziyin2021 assumes that $N \to \infty$ when $D$ is fixed. Also there seems to be a typo in the Ziyin2021 paper where they use both “d” and “D”, but these should be the same.

---

> > ### Comment · Reviewer_ZKu7 · 2021-08-19
> > **update**
> >
> > Thanks for the careful discussion and clarification. I would like to raise the score to 6.

---

### Official Review · Reviewer_7r4P · 2021-07-15

**Rating:** 8
**Confidence:** 2

**Summary:**

The authors study exact asymptotics of the dynamics of momentum-accelerated SGD for a least-squares regression model with a "mean-field" regression design matrix A, in the scaling limit where n,d -> infinity proportionally. The paper has the following contributions:

(1) The authors introduce a d-dimensional diffusion approximation, eq. (7), which they conjecture to describe the asymptotic behavior of a class of momentum-accelerated multi-pass SGD algorithms, under certain assumptions for the design. The conjecture is proven rigorously for (non-momentum-accelerated) SGD and orthogonally invariant designs A in Theorem 2, using a small adaptation of the argument by Paquette et al '11. A heuristic derivation of this approximation for the more general class of momentum-accelerated algorithms of interest is given in Appendix B.1, and the conjecture is supported strongly by numerical results on both Gaussian designs and regression designs derived from MNIST images.

(2) The authors prove that the expected loss value at any epoch t that is predicted by this diffusion equation admits a form given by a Volterra integral equation, Theorem 1, which depends on the eigenvalues of the Hessian A'A. Assuming the correctness of the approximation in (1), the convergence rates of various momentum acceleration schemes may then be analyzed and compared by analyzing the long-time behavior of this Volterra integral equation.

(3) The authors carry out this analysis of the Volterra integral equation for the heavy-ball and Nesterov acceleration schemes, and establish either exponential rates of convergence or polynomial rates of convergence to the limit loss, depending on whether d/n -> 1. (This limit loss is not zero in the studied regime, and is instead characterized by Theorem 3.) The results of this analysis are summarized in Table 3. This analysis is more straightforward for the heavy-ball method, where the kernel defining the Volterra equation is of convolution form. For the Nesterov-accelerated scheme, the authors show that the long-time dynamics may be approximated by a convolutional kernel (Appendix C.3) and hence analyzed via this approximation.

(4) A few practical insights from this analysis are a prescription of the scaling of the step size and momentum parameters in n that are needed to obtain a speed-up over non-accelerated SGD, and the dependence of this speed-up on the choice of acceleration scheme (heavy-ball vs. Nesterov) and on the ratio d/n.

**Main Review:**

I think this paper addresses an important question---the correct scaling-limit approximation of momentum-accelerated SGD---albeit in the context of a simple linear model. It would have been ideal if the paper made rigorous the arguments in Appendix B.1 to provide a more complete version of Theorem 2, and in the absence of this result, perhaps the authors can give a sense in Appendix B.1 on how large the technical challenges might be to establish this theorem for the full sDA class.

Even in the absence of this result, I think the proposal of the form of the Volterra dynamics and the analyses of these dynamics that the authors provide are sufficient for publication in NeurIPS. I provide a few comments on exposition below:

(1) I find Figure 2 hard to interpret---is it saying that for large n * theta, sDAHB is doing worse than SGD because it has a larger function value in the last iterate? I find the wording of the caption unclear about what the figure is showing.

(2) Precise statements of some of the main high-level take-aways of the paper---for example, the convergence rates summarized in Table 3 and the equivalence of sHB and SGD in Eq. (14)---cannot be found in the main text, and are instead delegated to the appendix. Actually, even in the appendix I cannot find where many of the claims of Table are established. (For example the analyses in the "non-strongly convex" setting in Appendices C and D have some dependence on the scaling exponent alpha, but where is alpha in Table 3?) I would like to ask the authors to formalize the discussion on page 9 and more precisely state the results that are referred to in Table 3 (i.e., what are the bounds for psi(t) under what assumptions), preferably in the main text, but otherwise in the appendix.

(3) My understanding is that the authors refer to the setting d/n -> 1 as "non-strongly-convex" and all other settings of d/n larger or smaller than 1 as "strongly convex". I don't know if this may cause confusion or is the usual nomenclature---perhaps it's more common to describe any setting d <= n as strongly convex and d > n as non-strongly-convex, which is simply based on whether the quadratic loss f(x) is strongly convex in x?

**Time Spent Reviewing:**

4

---

> ### Author Response · Authors · 2021-08-08
> **Response to Reviewer 7r4P**
>
> We thank the reviewer for their comments and careful reading of the paper.
>
> Technical challenge establishing Thm 2 to the sDA class (see also our response to Reviewer meKg): We have strong confidence that it can be done and it is an interesting random matrix theorem to establish. The techniques used for SGD in Paquette ’21 are ad-hoc and rely on martingale techniques. The type of martingales that appear in the sDA class are quite different owning to the long moving average of the stochastic gradients (i.e. momentum term). Consequently, many of the arguments will have to be re-engineered. In this paper, we wanted to establish that there were practical consequences of this analysis (e.g. step size selection and average-case analysis) and focus on drawing interesting conclusions for momentum-based stochastic algorithms.  This turned out to be quite technical, itself. So we left the other technical concentration result to future work. We also believe the NeurIPS audience would be more receptive to the implications of the analysis, while the concentration result would be better suited to a probability theory journal.
>
> Response to specific comments:
>
>  [1] Figure 2 clarification: The Figure shows that the last iterate function values for sDAHB and SGD after a fixed number of steps while varying the momentum parameter. There are three conclusions:
>
> (a). As you remarked, for very large momentum ($\theta \cdot n$ is small), sDAHB performs worse than SGD in the sense that it has a larger function value after a fixed number of iterations.
>
> (b). For small momentum (e.g. when $\theta$ is fixed and independent of $n$), sDAHB is equivalent to SGD after rescaling the step-size. Equivalence here means that the function values are exactly equal after a fixed number of iterations.
>
> (c). There is a tiny improvement of sDAHB over SGD if the step-size is small (e.g. $\gamma_{sgd} = 0.25$) and the momentum parameter $n \cdot \theta \approx 1$. To see the improvement, one can zoom in on Figure 2.
>
> If accepted, we will add clarification to the caption and we will change the step-sizes to illustrate (c) better.
>
> [2] Precise statements of main contributions: In Table 3, we will add specific pointers to the Appendix. Also in Table 3, $\alpha$ is ½ which corresponds to the isotropic features example (Sec. 3.2).
>
> Currently the results in the appendix are organized algorithm by algorithm. To find the convergence rates for one of the algorithms, one is interested in finding $f(x_t) < \varepsilon$ and since $\psi_H(t)$ is hypothesized to equal $f(x_t)$ under Homogenized SGD, we instead find where $\psi_H(t) < \varepsilon$. Here $\psi_H$ is the solution to the Volterra equation. The corresponding asymptotics for each algorithm are given in their corresponding section, and this relies on bounding the ``Malthusian exponent’’.  Again, we will add pointers to the appendix and add theorems for each rate claim.
>
> We agree that the results mentioned in the main paper (especially on page 9) are hard to find, and we will add an extended conclusions section where we organize the results and make formal claims.  Currently they are located in Appendix D, but they are formulated in a different way than is discussed in the main paper:
>
> The equivalence of SGD and SHB is due to the equivalence of their Volterra equations, Theorem 5 in Appendix D.
>
> Proposition 10 in Appendix D for the fact that SHB can only improve over SGD by a constant factor in the rate.).
>
> [3] Non-strongly convex vs. strongly convex setting: We apologize for the confusion here. When $d/n \neq 1$, we call the setting strongly convex and non-strongly convex setting when $d/n = 1$. Consider the isotropic features data generation, that is, $A_{ij}$ distributed as $N(0,1)$. The limiting eigenvalue distribution for A is the Marchenko-Pastur. When $d/n \neq 1$ and $n \to \infty$, the Marchenko-Pastur distribution has a gap between $0$ and smallest non-zero eigenvalue. This gap mimics the behavior of having a minimum eigenvalue bounded away from $0$; hence the reference to strongly convex. As in traditional worst-case analysis, if you have a gap between $0$ and the next smallest eigenvalue, one obtains a linear rate of convergence. We also see that here, provided $d/n \neq 1$. When $d/n \to 1$, this gap disappears. We have eigenvalues which converge to $0$ with $n$. As a result, the complexity more closely resembles the non-strongly convex setting. We will clarify this in our paper if accepted.

---

> > ### Comment · Reviewer_7r4P · 2021-08-27
> > **response**
> >
> > Thanks very much for this detailed response. They address adequately my comments, and (if accepted) I do encourage the authors to make the revisions to page 9 and Appendix D as discussed above.

---

### Official Review · Reviewer_meKg · 2021-07-16

**Rating:** 7
**Confidence:** 4

**Summary:**

This paper analyzes the behavior of stochastic gradient algorithms on a high dimensional least-squares problem. Under the proposed unified framework, it concludes that stochastic momentum methods with a fixed momentum parameter does not improve the convergence of SGD. The authors then propose a new algorithm with momentum parameters depending on sample size, which achieves an optimal complexity asymptotically.

**Limitations And Societal Impact:**

None.

**Main Review:**

The paper is in general well-written.

The authors propose an unified framework for stochastic gradient descent algorithms with momentum and table 1 gives a clear summary of different methods under this framework. Under the assumption of random least-squares model, they consider a diffusion approximation method (*homogenized SGD*) to anaylze the behavior of stochastic momentum algorithms. Theorem 1 describes the dynamics of  homogenized SGD and Theorem 2 links this approximation to SGD. However, as also mentioned by the authors, the high probability bound in Theorem 2 only includes the case of SGD and it would be better to consider the general cases and gives a solid theoretical support of their proposed sDANA and sDAHB algorithms.

Under the  stochastic momentum framework, the authors are able to express the result in Theorem 1 in a simple convolution–type Volterra equation and analyze the limiting behavior of these algorithms and the new algorithm sDANA obtains an accelerated average-case rate in the non-strongly convex case.

The theoretical proof in this paper requires delicate technique and I appreciate the effort authors made to solve these ODEs explicitly. My only concern is that most techniques and theoretical results are direct extension of Paquette et al. (2021). The link between SGD and the homogenized SGD is important but incomplete to the general cases, and Theorem 2 is also a direct citation from Paquette et al. (2021).

Reference mentioned above:

- C. Paquette, K. Lee, F. Pedregosa, and E. Paquette. SGD in the Large: Average-case Analysis, Asymptotics, and Stepsize Criticality. *arXiv preprint arXiv:2102.04396*, 2021.

**Time Spent Reviewing:**

2

---

> ### Author Response · Authors · 2021-08-08
> **Response to Reviewer meKg**
>
> We thank the reviewer for their comments and careful reading of the paper.
>
> Direct extensions of Paquette, 2021: We agree that the set-up is similar to Paquette et al, 2021.
>
> Most of the technical work in Paquette et al 2021 is to show concentration of the SGD training error around the Volterra equation.   The actual resulting Volterra equation for SGD, as compared to the paper under review, is quite simple and even explicitly solvable in some cases.
>
> Adding momentum greatly increases the complexity of the Volterra equation: specifically, the kernels and forcing terms are defined by 3rd-order ODEs (SGD only involved 1st-order ODEs).  This is particularly difficult in the case of Nesterov—type momentum, where the coefficients are non-constant coefficient 3rd-order ODEs.  So essentially all the technical work in this paper is devoted to (1) establishing this 3rd order ODE description assuming the Homogenized SGD assumption, and (2) analyzing the resulting Volterra equation.  For both of these tasks, there are no shared technical details, save for the fact that both involve the analysis of Volterra integral equations.
>
> In contrast, to establish the link between the whole sDA class and homogenized SGD, which is something we have left open, there would be substantial overlap between this paper and Paquette et al 2021.  We have chosen to not do this here, for the following reasons:
>
> [1] The techniques used for SGD in Paquette et al, 2021 are ad-hoc and rely on martingale techniques. The type of martingales that appear in the sDA class are quite different owning to the long moving average of the stochastic gradients (i.e. momentum term). Consequently, many of the arguments would need to be re-engineered.
>
> [2] The analysis of the homogenized SGD for the sDA class turned out to be quite technical itself, so we decided not to pursue in one paper an entirely different technical problem –moreover one which is of a completely different nature.
>
> [3] We believe the NeurIPS audience would be more receptive to the implications of the analysis and the successful numerical simulations/experiments (concentration, Fig. 1,2,5 and the MNIST data set Fig. 6), while less interested in the technical details of the concentration result.
>
> [4] The approximation of sDA by homogenized SGD, we view primarily as a random matrix theory problem, and so it is better suited project for a probability theory journal.  Hopefully, the results here justify the project of showing concentration of sDA around homogenized SGD.  This would also be a good opportunity to generalize beyond orthogonally invariant matrices, which is also the setting of Paquette et al, 2021.

---

### Official Review · Reviewer_BCD3 · 2021-07-23

**Rating:** 6
**Confidence:** 3

**Summary:**

This paper presents results on stochastic gradient methods on high dimensional least squares problems and presents a characterization of the loss of the final iterate run with a class of stochastic dimension adjusted (SDA) based momentum/Nesterov acceleration methods. The methods guarantee convergence to a neighborhood of the optimal solution where the size of the neighborhood shrinks to zero as the limit of the number of samples increases to infinity.

**Ethical Concerns:**

None.

**Limitations And Societal Impact:**

Yes.

**Main Review:**

To my knowledge, the results appear to be novel and I tend towards acceptance.

[1] Can assumptions on scale of initialization, independence of noise and noise variance (assumption 1) be relaxed? I believe most standard analysis of SGD in these settings work without relying on these assumptions (see the work of Bach and Moulines (2013) for an example).

[2] Can the authors obtain results similar to what they have presented, but with comparison to L(w^*) instead of the neighborhood of the solution? In principle, one can obtain results that converge to the solution using ideas like iterate averaging [Bach and Moulines 2013]/suffix averaging[Rakhlin et al. 2012]? Alternatively, if we wish to obtain guarantees on the final iterate, there needs to be some decay on the stepsizes [Ge et al. 2019]? I am interested as to what technical challenges exist in a path towards this result.

Bach and Moulines (2013): Non-strongly-convex smooth stochastic approximation with convergence rate O(1/n)

Rakhlin et al (2012): Making gradient descent optimal for strongly convex stochastic optimization

Ge et al (2019):  The Step Decay Schedule: A Near Optimal, Geometrically Decaying Learning Rate Schedule for Least Squares

**Time Spent Reviewing:**

3

---

> ### Author Response · Authors · 2021-08-08
> **Response to Reviewer BCD3**
>
> We thank the reviewer for their comments and careful reading of the paper.
>
> [1] Can the assumptions on scale of initialization, independence of noise and noise variance (Assumption 1) be relaxed?
>
> Currently, we cannot relax these assumptions. Since we want to compare complexity across dimensions, one needs to standardize the scale of initialization. For instance, the norm of a random initial vector $x_0$ grows like $\sqrt{d}$. Hence one would be starting farther away from the optimum as $d \to \infty$ which naturally effects the complexity. It is an interesting and open problem to consider other ways to generate the target vector $b$, e.g., incorporating non-independent noise and unbounded noise variance. We chose to model $b$ by a simple generative model. Generative models with this scaling have been used in numerous works (e.g. [Gerbelot et al., 2020, Hastie et al., 2019, Mei and Montanari, 2019]).
>
> We note that it is necessary to impose stronger assumptions than previous works such as those assumptions in Bach and Moulines (2013). This is because we want to capture the typical behavior of high-dimensional data sets. Intuitively, high-dimensions mean one has more possibilities for the inputs into an algorithm, so the input which generates the worst-case complexity can be far from typical. To capture typical complexity of an algorithm for large-scale problems, one must impose additional assumptions that can only be visible in high-dimensions (e.g. de-localization of eigenvectors).
>
> [2] Can the authors obtain results similar to what they have presented, but with comparison to L($w^*$)?
>
> Yes, one can do a similar analysis but with a decreasing step-size so that the algorithms converge. The main challenge is to modify the martingale to account for the changing step-sizes with time, but we are confident that this could be done.
>
> Note also that our analysis does give last iterate convergence to a neighborhood. For SGD and sHB, our analysis shows that last iterate average-case complexity has the same complexity as averaging the iterates.

---

> > ### Comment · Reviewer_BCD3 · 2021-08-23
> > **Thanks for the clarifications**
> >
> > Thank you for the clarifications. To me, I'd consider convergence to a neighborhood of the solution to be a shortcoming; I will retain my score, but, would recommend the authors present results that converge to the solution rather than a neighborhood of the solution, which I think is generally the norm for results in optimization.

---

> > > ### Author Response · Authors · 2021-08-23
> > > **Clarification: Convergence to neighborhood**
> > >
> > > Thank you for the comment.
> > >
> > > We wanted to clarify the neighborhood convergence.
> > >
> > > [1]. In the over parameterized setting ($d \ge n$), there is no neighborhood ($L(w^*) = 0$) and our average-case complexity is for the convergence rate of $L(w_k)-L(w^*)$.
> > >
> > > [2]. Interestingly in the under parameterized setting ($d < n$), our results show that the loss value stabilizes at a quantity, $\psi(\infty)$, as $n \to \infty$. **All** the runs of the algorithm converge to the **same value**; there is no variance in the convergence to that value. However this value may not be the minimum, $L(w^*)$. Thus, our average case complexity is computed w.r.t. to this value $\psi(\infty)$ instead of $L(w^*)$. Note from our analysis, one can compute $\psi(\infty)$. You could use this value $\psi(\infty)$ for the fixed step-size to restart the algorithm with and iterate with a smaller step-size to get the rate to $L(w^*)$.
> > >
> > > We agree with the reviewer that looking at convergence to $L(w^*)$ would be interesting future work.

---

### Decision · Program_Chairs · 2021-09-27

**Decision:**

Accept (Poster)

**Comment:**

This paper presents several interesting results concerning stochastic gradient methods on high-dimensional least-squares problems. The characterization of the role played by the momentum coefficient in terms of convergence is sharp. The new task-dependent algorithm matches the optimal complexity.